# ON PROVABLE LENGTH AND COMPOSITIONAL GENERALIZATION

## ABSTRACT

Out-of-distribution generalization capabilities of sequence-to-sequence models can be studied from the lens of two crucial forms of generalization: length generalization – the ability to generalize to longer sequences than ones seen during training, and compositional generalization: the ability to generalize to token combinations not seen during training. In this work, we provide first provable guarantees on length and compositional generalization for common sequence-to-sequence models – deep sets, transformers, state space models, and recurrent neural nets – trained to minimize the prediction error. Taking a first principles perspective, we study the realizable case, i.e., the labeling function is realizable on the architecture. We show that *simple limited capacity* versions of these different architectures achieve both length and compositional generalization. In all our results across different architectures, we find that the learned representations are linearly related to the representations generated by the true labeling function.

## 1 INTRODUCTION

Large language models (LLMs), such as the GPT models (Achiam et al., 2023) and the Llama models (Touvron et al., 2023), have led to a paradigm shift in the development of future artificial intelligence (AI) systems. The accounts of their successes (Bubeck et al., 2023; Gunasekar et al., 2023) as well as their failures, particularly in reasoning and planning (Bubeck et al., 2023; Stechly et al., 2023; Valmeekam et al., 2023), continue to rise. The successes and failures of these models have sparked a debate about whether they actually learn general algorithms or if their success is primarily due to memorization and a superficial form of generalization (Dziri et al., 2024).

A model's ability to perform well across different distribution shifts highlights its ability to learn general algorithms. For models with fixed-dimensional inputs, considerable efforts have led to methods with provable out-of-distribution (OOD) generalization guarantees (Rojas-Carulla et al., 2018; Rame et al., 2022; Chaudhuri et al., 2023; Wiedemer et al., 2023b; Eastwood et al., 2024). For sequence-to-sequence models, a large body of empirical works have investigated OOD generalization (Anil et al., 2022; Jelassi et al., 2023) but we lack efforts that study provable OOD generalization guarantees for these models. These provable guarantees provide a stepping stone towards explaining the success of the existing paradigm and also shine a light on where the existing paradigm fails.

OOD generalization capabilities of sequence-to-sequence models can be studied from the lens of two forms of generalization: length generalization – the ability to generalize to longer sequences than ones seen during training, and compositional generalization – the ability to generalize to token combinations not seen during training. While transformers (Vaswani et al., 2017) are the go-to sequence-to-sequence models for many applications, recently, alternative architectures based on state-space models, as noted by Gu et al. (2021), Orvieto et al. (2023b), and Gu & Dao (2023), have shown a lot of promise. This motivates us to study a range of natural sequence-to-sequence architectures, including deep sets (Zaheer et al., 2017), transformers, state space models (SSMs), and recurrent neural networks (RNNs). We focus on the realizable case, i.e., the labeling function is in the hypothesis class of the architecture. Further, in our theoretical analysis, we make certain simplifications to permit tractable analysis, for instance, we study RNNs with a limit on hidden state dimension. Our key contributions and insights are summarized below.

- Simple limited capacity versions of the different architectures namely deep sets, transformers, SSMs, and RNNs, provably achieve length and compositional generalization.

- In all our results across different architectures, we find that the learned representations are linearly related to the representations generated by the true labeling function, which is also termed *linear identification* (Khemakhem et al., 2020; Roeder et al., 2021).

- Through a range of experiments, we show the success in both forms of generalization, matching the predictions of the theory and even going beyond.

To the best of our knowledge, our provable guarantees for length and compositional generalization for sequence-to-sequence models are the first in the literature.

## 2 RELATED WORKS

**Length generalization**  In the field of length generalization, many important empirical insights have been synthesized over the last few years. Shaw et al. (2018) discovered the drawbacks of absolute positional embeddings and suggested relative positional embeddings as an alternative. Subsequent empirical analyses, notably by Anil et al. (2022) and Jelassi et al. (2023), explored length generalization in different settings for transformer-based models. Key findings revealed that larger model sizes don't necessarily enhance generalization, the utility of scratchpads varies, and the effectiveness of relative positional embeddings appeared task-dependent. In Kazemnejad et al. (2024), the authors did a comprehensive study of different positional embeddings and provided evidence to show that explicit use of positional encodings is perhaps not essential. In Delétang et al. (2022), the authors conducted experiments on tasks divided based on their placement in the Chomsky hierarchy and showed the importance of structured memory (stack, tape) in length generalization. In a recent work, Zhou et al. (2023) proposed RASP conjecture, which delineates the tasks where transformers excel or fall short in length generalization, emphasizing the necessity of task simplicity for the transformer and data diversity. Our work is inspired by the experimental findings of their work. While Zhou et al. (2023) provide empirical evidence for the conjecture, our work formalizes and proves simpler versions of the conjecture for a range of architectures.

On the theoretical side of length generalization, in Abbe et al. (2023), the authors showed an implicit bias of neural network training towards min-degree interpolators. This bias was used to explain the failures of length generalization on the parity task from Anil et al. (2022). In Xiao & Liu (2023), the authors leverage directed acyclic graphs (DAGs) to formulate the computation in reasoning tasks and characterize conditions under which there exist functions that permit length generalization. Our results crucially differ, we show a range of conditions under which models learned via standard expected risk minimization achieve length and compositional generalization.

**Compositional generalization**  Compositionality has long been seen as a key piece to the puzzle of human-level intelligence (Fodor & Pylyshyn, 1988; Hinton, 1990; Plate et al., 1991; Montague, 1970). Compositionality is a large umbrella term associated with several aspects (Hupkes et al., 2020). In this work, we focus on systematicity, which evaluates a model's capability to understand known parts and combine them in new contexts. The breadth of research on compositional generalization, encompassing studies like Lake & Baroni (2018); Loula et al. (2018); Gordon et al. (2019); Hupkes et al. (2020); Kim & Linzen (2020); Xu et al. (2022); Arora & Goyal (2023); Zhang et al. (2024), is too expansive to address comprehensively here, refer to these surveys (Lin et al., 2023; Sinha et al., 2024) for more detail.

In recent years, several works have taken first steps towards theoretical foundations of compositionality. We leverage the mathematical definition of compositionality from Wiedemer et al. (2023b), which focuses on generalization to the Cartesian product of the support of individual features. In Dong & Ma (2022), the authors analyze the conditions that provably guarantee generalization to the Cartesian product of the support of individual training features. Dong & Ma (2022) studied additive models, i.e., labeling function is additive over individual features. In (Wiedemer et al., 2023a), the authors focus on a more general model class than Dong & Ma (2022), where the labeling function is of the form $f(x_1, \cdots, x_n) = C(\psi_1(x_1), \cdots, \psi_n(x_n))$. However, to guarantee compositional generalization, Wiedemer et al. (2023b) require that the learner needs to know the exact function $C$

that is used to generate the data. In our work, we do not make such an assumption, our data generation is dictated by the architecture in question, e.g., RNN, and we constrain the dimension of its hidden state. Lachapelle et al. (2023); Brady et al. (2023) extend these precursor results from Dong & Ma (2022) from the supervised setting to the unsupervised setting. In particular, Lachapelle et al. (2023); Brady et al. (2023) are inspired by the success of object-centric models and show additive decoder based autoencoders achieve compositional generalization.

## 3 PROVABLE LENGTH AND COMPOSITIONAL GENERALIZATION

We are given a dataset comprising of a sequence of inputs $\{x_1, \cdots, x_t\}$ and a corresponding sequence of labels $\{y_1, \cdots, y_t\}$, where each $x_i \in \mathbb{R}^n$ and $y_i \in \mathbb{R}^m$. Observe that this formulation includes both standard downstream tasks such as arithmetic tasks, e.g., $y_i = \sum_{j=1}^{i} x_j, y_i = \Pi_{j=1}^{i} x_j$ etc., as well as next-token prediction task, where $\{y_1, \cdots, y_t\} = \{x_2, \cdots, x_{t+1}\}$. We denote a sequence $\{s_1, \cdots, s_t\}$ as $s_{\leq t}$, $X_k$ is random variable for token at $k^{th}$ position and its realization is $x_k$. Consider a sequence $\{x_j\}_{j=1}^{\infty}$, which is sampled from $\mathbb{P}_X$, and a subsequence of this sequence $x_{\leq t} = \{x_j\}_{j=1}^{t}$, whose distribution is denoted as $\mathbb{P}_{X_{\leq t}}$. The label $y_t = f(x_{\leq t})$, where $f$ is the labeling function. The tuple of base distribution and the labeling function is denoted as $\mathcal{P} = \left\{\mathbb{P}_X, f\right\}$ and the tuple of base distribution up to length $t$ is denoted as $\mathcal{P}(t) = \left\{\mathbb{P}_{X_{\leq t}}, f\right\}$. The support of $k^{th}$ token $X_k$ in the sequence sampled from $\mathbb{P}_X$ is denoted $\mathsf{supp}(X_k)$. Given training sequences of length $T$ from $\mathcal{P}(T)$, we are tasked to learn a model from the dataset that takes a sequence $x_{\leq t}$ as input and predicts the true label $y_t$ as well as possible. If the model succeeds to predict well on sequences that are longer than maximum training length $T$, then it is said to achieve length generalization (a more formal definition follows later). Further, if the model succeeds to predict well on sequences comprising of combination of tokens that are never seen under training distribution, then it is said to achieve compositional generalization (a more formal definition follows later.). We study both these generalization forms next.

**Learning via expected risk minimization** Consider a map $h$ that accepts sequences of $n$-dimensional inputs to generate a $m$-dimensional output. We measure the loss of predictions of $h$, i.e., $h(x_{\leq t})$, against true labels as $\ell\big(h(x_{\leq t}), y_t\big)$, where $y_t$ is the true label for sequence $x_{\leq t}$. In what follows, we use the $\ell_2$ loss. Given sequences sampled from $\mathcal{P}(T)$, the expected risk across all time instances up to maximum length $T$ is defined as $R(h; T) := \sum_{t=1}^{T} \mathbb{E}\big[\ell\big(h(x_{\leq t}), y_t\big)\big]$. The learner aims to find an $h^*$ that solves

$$h^* \in \underset{h \in \mathcal{H}}{\arg\min}\, R(h; T), \tag{1}$$

where $\mathcal{H}$ is the hypothesis class of models. We seek to understand the properties of solutions to equation 1 through the lens of following questions.

> When do common sequence-to-sequence models $\mathcal{H}$ succeed at length & compositional generalization and when do they fail?

**Definition 1.** *Consider the setting where a model is trained on sequences $(x_{\leq t}, y_{\leq t})$ of length up to $T$ drawn from $\mathcal{P}(T)$. If the model achieves zero error on sequences $(x_{\leq t}, y_{\leq t})$ of length up to $\tilde{T}$ drawn from $\mathcal{P}(\tilde{T}), \forall\, \tilde{T} \geq 1$, then it achieves length generalization w.r.t. $\mathcal{P}$.*

In the above definition of length generalization, we simply ask if the model generalizes to longer sequences. We drop the phrase w.r.t $\mathcal{P}$ hereafter to avoid repetition. We now define a test distribution that evaluates compositional generalization capabilities. We consider sequences of fixed length $T$. Define a uniform distribution $\mathbb{Q}_{X_{\leq T}}$ such that the support of $\mathbb{Q}_{X_{\leq T}}$ equals the Cartesian product of the support of each token $X_k$ from $\mathbb{P}_X$, we write this joint support as $\Pi_{j=1}^{T}\mathsf{supp}(X_j)$. In this case as well, the labeling function continues to be $f$. Hence, we obtain the tuple $\mathcal{Q}(T) = \{\mathbb{Q}_{X_{\leq T}}, f\}$.

**Definition 2.** *Consider the setting where a model is trained on sequences $(x_{\leq t}, y_{\leq t})$ of length up to $T$ drawn from $\mathcal{P}(T)$. If the model achieves zero error on sequences $(x_{\leq t}, y_{\leq t})$ of length up to $T$ drawn from $\mathcal{Q}(T)$, then it achieves compositional generalization.*

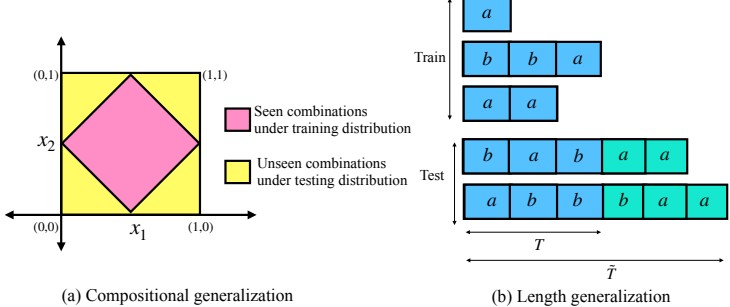

(a) Compositional generalization          (b) Length generalization

Figure 1: Illustrating support of train vs test distribution for (a) compositional generalization and (b) length generalization.

This definition of compositionality above is based on Wiedemer et al. (2023b); Brady et al. (2023). In this definition, we ask if the model generalizes to new combinations of seen tokens.

*Illustrative example* We teach the model multiplication on sequences of length $2$, where each $x_j$ is a scalar, $y_i = \Pi_{j=1}^i x_j$. Say the support of the entire sequence drawn from $\mathbb{P}_X$ is $\left\{ x \mid \|(x_1, x_2) - \frac{1}{2}\mathbf{1}\|_1 \leq \frac{1}{2},\ x_k \in [0,1],\ \forall k \geq 3 \right\}$. The support of training distribution $\mathbb{P}_{X_{\leq 2}}$ is $\left\{ x \mid \|(x_1, x_2) - \frac{1}{2}\mathbf{1}\|_1 \leq \frac{1}{2} \right\}$ shown in the pink region in Figure 1a. In Figure 1a, we illustrate compositional generalization, the model is trained on pink region and asked to generalize to the yellow region. Further, if the model continues to correctly multiply on longer sequence lengths in $\mathbb{P}_{X_{\leq \tilde{T}}}$ for $\tilde{T} \geq T$, then it achieves length generalization shown in Figure 1b.

**A preview of the technical challenges** Both notions of compositional generalization and length generalization introduced above involve testing on distributions whose support is not contained in the training distributions. The long line of work on distribution shifts (Sugiyama et al., 2007; David et al., 2010; Ben-David & Urner, 2014; Rojas-Carulla et al., 2018; Arjovsky et al., 2019; Ahuja et al., 2021) assume the support of test is contained in the support of train distribution. In recent years, there has been development of theory for distribution shifts under support mismatch Dong & Ma (2022); Abbe et al. (2023); Wiedemer et al. (2023b); Netanyahu et al. (2023); Shen & Meinshausen (2023). Our work is closer to the latter line of work but it comes with its own *technical challenges*, which involve building new proofs different from the above line of work, as we study a new family of models, i.e., sequence-to-sequence models, and a new form of generalization, i.e., length generalization.

**RASP conjecture** Zhou et al. (2023) propose a conjecture backed by empirical evidence, which delineates the conditions that suffice for length generalization for transformers. The conjecture places three requirements – a) realizability: the task of interest is realizable on the transformer, b) simplicity: the task can be expressed as a short program in RASP-L language, c) diversity: the training data is sufficiently diverse such that there is no shorter program that achieves in-distribution generalization but not OOD generalization. We leverage assumptions similar to a) and b). We assume realizability, which means labeling function $f$ is in the hypothesis class $\mathcal{H}$. As to simplicity, we consider hypothesis class $\mathcal{H}$ with limited capacity, e.g., we study one block transformer, or RNNs with a limit on hidden state dimension. We emphasize that the third assumption c) on diversity from Zhou et al. (2023) is quite strong. In our setting, we do not invoke it and instead, we require that the support of test distribution is not larger than the Cartesian product of the marginal distribution of the tokens. We now move to proving simplified versions of this conjecture for different architectures.

### 3.1 DEEP SETS

Deep sets are a natural first choice of architecture to study here. These take sets as inputs and thus handle inputs of arbitrary lengths. These were introduced in Zaheer et al. (2017). Informally stated, Zaheer et al. (2017) show that a large family of permutation-invariant functions can be decomposed as $\rho\left(\sum_{x \in \mathcal{X}} \phi(x)\right)$. Consider the examples of the sum operator or the multiplication operator, which take $\{x_1, x_2, \cdots, x_k\}$ as input, and return the sum $y = \sum_{j=1}^k x_j$ or the product $y = \Pi_{j=1}^k x_j$. These

operations are permutation invariant and can be expressed using the decomposition above. For the sum operator $\rho$ and $\phi$ are identity and for the multiplication operator $\rho = \exp$ and $\phi = \log$. Consider another example from language. We construct a bag of words sentiment classifier, where $\{x_1, x_2, \cdots, x_i\}$ is the set of words that appear in the sentence, $\phi(x_j)$ is the feature embedding for word $j$. $\sum_{j \leq i} \phi(x_j)$ is the representation of the entire sentence which is passed to the final layer $\rho$ that generates the sentiment label. In what follows, we aim to understand when such a classifier generalizes to sentences beyond training lengths and to new sentences comprised of unseen word combinations.

**Assumption 1.** *Each function in the hypothesis class $\mathcal{H}$ takes a sequence $\{x_1, \cdots, x_i\}$ as input and outputs $h(x_1, \cdots, x_i) = \omega\Big(\sum_{j \leq i} \psi(x_j)\Big)$, where $\omega$ is a single layer perceptron with a continuously differentiable bijective activation (e.g., sigmoid) and $\psi$ is a map that is differentiable.*

A simple mathematical example of a function from the above family when $\psi(x_j) = [x_j, x_j^2, x_j^3]$ is a polynomial map of degree 3 and each $x_j$ is a scalar $- \sigma\big(a \sum_{j \leq i} x_j + b \sum_{j \leq i} x_j^2 + c \sum_{j \leq i} x_j^3\big)$. In the assumption that follows, we assume that the support of the sequences is regular closed in the standard topology, i.e., the set is equal to the closure of its interior.

**Assumption 2.** *The joint support $\mathsf{supp}(X_{\leq i})$ is a regular closed set for all $i \leq T$.*

In most of our results in the main body, we invoke Assumption 2. This assumption is satisfied in many cases if the tokens are continuous random variables but it is not satisfied for discrete random variables. In the Appendix, we extend several of our key results to discrete tokens.

**Linear identification** Each architecture that we study in this work relies on a hidden representation that is passed on to a non-linearity to generate the label. Under the realizability condition for deep sets, the labeling function takes the form $f(\mathcal{X}) = \rho(\sum_{x \in \mathcal{X}} \phi(x))$, where $\phi(x)$ is the hidden representation. If the learned deep set is denoted by $\omega(\sum_{x \in \mathcal{X}} \psi(x))$, then the learned hidden representation is $\psi(x)$. If $\psi(x) = A\phi(x)$, then the learned representation is said to *linearly identify* the true data generating representation $\phi(x)$. We borrow this definition from the representation identification literature (Khemakhem et al., 2020; Roeder et al., 2021).

**Theorem 1.** *If $\mathcal{H}$ follows Assumption 1, the realizability condition holds, i.e., $f \in \mathcal{H}$, $\mathsf{supp}(X_j) = [0, 1]^n$, $\forall j \geq 1$, and the regular closedness condition in Assumption 2 holds, then the model trained to minimize the risk in equation 1 with $\ell_2$ loss generalizes to all sequences in the hypercube $[0, 1]^{nt}$, $\forall t \geq 1$ and thus achieves length and compositional generalization.*

The detailed proof is in Section C.1. In the above result, we require the support of the marginal distribution of each token to be $[0, 1]^n$. The support of $T$ token length sequence under the joint training distribution can still be a much smaller subset of $[0, 1]^{nT}$, as illustrated in Figure 1a (and Figure 4 in the Appendix). Despite this the model generalizes to all sequences in $[0, 1]^{nt}$ for all $t$. An important insight from the proof is that the hidden representation learned by the model is a linear transform of the true hidden representation, i.e., it achieves linear identification $\psi = A\phi$ (Further details are in Corollary 1).

**Extensions of Theorem 1** In Theorem 8, we extend Theorem 1 to $\omega$ from $C^1$-diffeomorphisms[1]. As a by product, we obtain length & compositional generalization for multiplication operator. In Theorem 7, we extend the above result to discrete tokens. Further, most results in this work translate to settings where we do not observe labels at all lengths from 1 to $T$ (further discussion in Appendix).

**High capacity deep sets** In the above results, we operated with some constraints on the deep sets. In Theorem 1, we used limited capacity $\omega$ that are represented via a single layer perceptron. In Theorem 8, we used $\omega$ that are represented via $C^1$-diffeomorphisms, which implies the output dimension of $\psi$ equals label dimension $m$ and cannot be larger. What happens when we work with deep sets with arbitrary capacity, i.e., no constraints on $\omega$ and $\psi$? These models then express a large family of permutation invariant maps (Zaheer et al., 2017). Suppose $\mathcal{H}$ is the class of all permutation invariant maps and the labeling function $f \in \mathcal{H}$. Consider a map $h$ such that $h = f$ for all sequences of length up to $T$, and $h = f + c$ otherwise. Observe that $h$ is permutation invariant and also belongs

---

[1]$C^1$-diffeomorphism - a continuously differentiable map that has a continuously differentiable inverse.

to $\mathcal{H}$. $h$ achieves zero generalization error on training sequences of length $T$ but a non-zero error on longer sequences. Thus in the setting of high capacity deep sets, there exist solutions to equation 1, which do not achieve length generalization. We can construct the same argument for compositional generalization as well and say $h = f$ on the training distribution (pink region) in Figure 1a and $h = f + c$ on the testing distribution (yellow region) in Figure 1a. In order to show successful generalization (length or compositional) in Theorem 1, we require all solutions to risk minimization in equation 1 to match the predictions of true labeling function on data beyond the support of the training distribution. In order to show that high capacity models are not guaranteed to succeed, we focused on showing that there exists a solution to equation 1 that does not generalize beyond the support of training distribution. A more nuanced argument for failure should show that there exist solutions reachable via gradient descent that do not generalize. We leave a rigorous theoretical exploration of this to future work. However, we conduct experiments with high capacity models in the Appendix (Section D.3) to illustrate failures in high capacity regime.

## 3.2 TRANSFORMERS

Ever since their introduction in Vaswani et al. (2017), transformers have revolutionized all domains of AI. In this section, we seek to understand length generalization for these models. Transformer architectures are represented as alternating layers of attention and position-wise non-linearity. We drop layer norms for tractability. Following similar notation as previous section, we denote position-wise non-linearity as $\rho$ and attention layer as $\phi$. We obtain the simplest form of causal transformer model as $\rho\Big( \sum_{j=1}^{i} \frac{1}{i} \cdot \phi(x_i, x_j) \Big)$. This decomposition captures linear attention, ReLU attention, sigmoid attention, ReLU squared attention, which were studied previously in Wortsman et al. (2023); Hua et al. (2022); Shen et al. (2023) and found to be quite effective in several settings. This decomposition does not capture softmax-based attention and developing provable length generalization guarantees for the same is an exciting future work. Other works (Bai et al., 2023) also replaced softmax with other non-linear attention for a more tractable analysis. We illustrate the sigmoid-based transformer from Wortsman et al. (2023) below. Let $W_q \in \mathbb{R}^{k \times n}$, $W_k \in \mathbb{R}^{k \times n}$, and $W_v \in \mathbb{R}^{k \times n}$ be the query, key and value matrices. $\rho$ is parametrized via a multi-layer perceptron denoted as MLP.

$$q_i = W_q x_i, \ k_j = W_k x_j, v_j = W_v x_j, \phi(x_i, x_j) = \sigma\left( \frac{q_i^\top k_j}{\sqrt{d}} \right) v_j, \ \mathsf{MLP}\Big( \sum_{j=1}^{i} \frac{1}{i} \cdot \phi(x_i, x_j) \Big). \quad (2)$$

In the above feedforward computation, the output of attention for the current query is computed and sent to the MLP to generate the label.

**Assumption 3.** *Each function in the hypothesis class $\mathcal{H}$ takes a sequence $\{x_1, \cdots, x_i\}$ as input and outputs $h(x_1, \cdots, x_i) = \omega\Big( \sum_{j \leq i} \frac{1}{i} \cdot \psi(x_i, x_j) \Big)$, where $\omega$ is a single layer perceptron with continuously differentiable bijective activation (e.g., sigmoid) and $\psi$ is a map that is differentiable.*

We denote the joint support of two tokens $X_i, X_j$ as $\mathsf{supp}(X_i, X_j)$.

**Theorem 2.** *If $\mathcal{H}$ follows Assumption 3, the realizability condition holds, i.e., $f \in \mathcal{H}$, $\mathsf{supp}(X_i, X_j) = [0, 1]^{2n}$, $\forall i \neq j$ and the regular closedness condition in Assumption 2 holds, then the model trained to minimize the risk in equation 1 (with $T \geq 2$) with $\ell_2$ loss generalizes to all sequences in the hypercube $[0, 1]^{nt}$, $\forall t \geq 1$ and thus achieves length and compositional generalization.*

Similar to Theorem 1, we observe linear identification here too, i.e., learned attention representation denoted $\psi$ is a linear transform of the true attention representation denoted $\phi$, i.e., $\psi(x_i, x_j) = C\phi(x_i, x_j)$, (details in Section C.2, see Corollary 2). We now extend Theorem 2 from single layer perceptron $\omega$ to $C^1$-diffeomorphism. We also extend Theorem 2 to discrete tokens in Theorem 10.

**Assumption 4.** *Each function in $\mathcal{H}$ takes $\{x_1, \cdots, x_i\}$ as input and outputs $h(x_1, \cdots, x_i) = \omega(\sum_{j=1}^{i-1} \frac{1}{i-1} \cdot \psi(x_i, x_j))$, where $\omega$ is a $C^1$-diffeomorphism, $\omega(0) = 0$.*

The reader would notice that the summation is up to $i - 1$ and hence it computes attention scores w.r.t all other terms in the context except $x_i$. We conjecture that the theorem that we present next extends to the more general case where summation includes the $i^{th}$ term.

**Assumption 5.** *The joint support $\mathsf{supp}(X_{\leq i})$ is a regular closed set for all $i \leq T$. The support of all pairs of tokens is equal, i.e., $\mathsf{supp}(X_i, X_j) = [0,1]^{2n}$, where $i \neq j, i \geq 1, j \geq 1$. The support of $[\phi(X_1, X_2), \phi(X_1, X_3)]$ is $\mathbb{R}^{2m}$, where $\phi$ is the embedding function for the labeling function $\rho(\sum_{j \leq i} \phi(x_i, x_j))$.*

**Theorem 3.** *If $\mathcal{H}$ follows Assumption 4, the realizability condition holds, i.e., $f \in \mathcal{H}$, and a further assumption on the support (Assumption 5) holds, then the model trained to minimize the risk in equation 1 (with $T \geq 3$) with $\ell_2$ loss generalizes to all sequences in $[0,1]^{nt}, \forall t \geq 1$ and thus achieves length and compositional generalization.*

**Multiple attention heads and positional encoding**   While the discussion in this section used a single attention head $\phi$, the results extend to multiple attention heads as shown in Section C.2. The model of transformers discussed so far uses the current query and compares it to keys from the past, it does not distinguish the keys based on their positions. For many arithmetic tasks such as computing the median, maximum etc., the positions of keys do not matter but for other downstream tasks such as sentiment classification, the position of the words can be important. In Section C.2, we adapt the architecture to incorporate relative positional encodings and show how some of the results extend. We modify the model as $\rho(\sum_{j=1}^{i} \frac{1}{i} \cdot \phi_{i-j}(x_i, x_j))$, where $\phi_{i-j}(x_i, x_j)$ computes the query key inner product while taking the relative position $i - j$ into account. We show that if $\phi_{i-j} = 0$ for $i - j > T_{\max}$, i.e., two tokens sufficiently far apart do not impact the data generation, then length generalization and compositional generalization are achieved.

**High capacity transformers**   In the above results, we operated with constraints on transformers, which limit their capacity. Similar to the setting of deep sets, observe that Assumption 3 constrains $\omega$ to single layer perceptron, Assumption 4 constraints $\omega$ to $C^1$-diffeomorphisms. What happens if we work with transformers with no constraint on $\omega$ and $\psi$? If $\psi(x, y) = \psi(\tilde{x}, y), \forall x \neq \tilde{x}$, then the decomposition for the causal transformer $\omega\left(\sum_{j=1}^{i} \frac{1}{i} \cdot \psi(x_i, x_j)\right)$ becomes $\omega\left(\sum_{j=1}^{i} \frac{1}{i} \cdot \psi(x_j)\right)$, which is very similar to deep sets. In such a case, we can adapt arguments similar to that of arbitrary capacity deep sets and argue that there exist solutions to equation 1 that do not achieve length and compositional generalization. We now move to state-space models and RNNs.

### 3.3   STATE SPACE MODELS

In recent years, state space models Gu et al. (2021); Orvieto et al. (2023b) have emerged as a promising competitor to transformers. In (Orvieto et al., 2023a;b), the authors used the lens of linear recurrent layer followed by position-wise non-linearities as the main building block to understand these models. We illustrate the dynamics of these models to show the generation of $x_{\leq t}$ and $y_{\leq t}$ next. Given the current input $x_t$, we combine it linearly with the hidden state from the past to obtain the current hidden state. The hidden state is input to $\rho$, which generates the label as follows

$$
\begin{aligned}
&h_1 = Bx_1; \quad h_2 = \Lambda h_1 + Bx_2; \quad \cdots, h_t = \Lambda h_{t-1} + Bx_t, \\
&y_1 = \rho(h_1); \; y_2 = \rho(h_2); \; \cdots\cdots, \quad\quad y_t = \rho(h_t),
\end{aligned} \tag{3}
$$

where $h_t \in \mathbb{R}^k$ is hidden state at point $t$, $\Lambda \in \mathbb{R}^{k \times k}, B \in \mathbb{R}^{k \times n}$ and $\rho : \mathbb{R}^k \to \mathbb{R}^m$. We can succinctly write $h_t = \sum_{j=0}^{t-1} \Lambda^j B x_{t-j}$.

**Assumption 6.** *Each function in the hypothesis class $\mathcal{H}$ takes a sequence $\{x_1, \cdots, x_i\}$ as input and outputs $h(x_1, \cdots, x_i) = \omega\left(\sum_{j=0}^{i-1} \Lambda^j B x_{i-j}\right)$, where $\omega : \mathbb{R}^k \to \mathbb{R}^m$ is a $C^1$-diffeomorphism, $B$ and $\Lambda$ are square invertible. As a result, $m = k = n$.*

**Assumption 7.** *The joint support $\mathsf{supp}(X_{\leq i})$ is a regular closed set for all $i \leq T$. The support of $X_1$ is $\mathbb{R}^n$. For some length $2 \leq i \leq T$ an there exists in sequences $x_{\leq i}$ such that their concatenation forms a $in \times in$ matrix of rank $in$.*

**Theorem 4.** *If $\mathcal{H}$ follows Assumption 6, and the realizability condition holds, i.e., $f \in \mathcal{H}$, and a further condition on the support, i.e., Assumption 7, holds, then the model trained to minimize the risk in equation 1 with $\ell_2$ loss $(T \geq 2)$ achieves length and compositional generalization.*

The proof is provided in Section C.3. Similar to previous theorems, the hidden state estimated by the learned model, $\tilde{h}_t$, and the true hidden state, $h_t$, bear a linear relationship (Corollary 4), i.e., linear identification is achieved. We extend Theorem 4 to discrete tokens in Theorem 12.

**High capacity SSMs** In the above result, we operated with certain constraints on SSMs, i.e., the input dimension, output dimension, and the hidden state dimension are equal. These constraints limit their capacity. What happens if we put no constraints on $\Lambda$, $B$ and $\omega$? Orvieto et al. (2023a) showed that SSMs with appropriately large $\Lambda$ and $B$ matrices can approximate a sequence-to-sequence mapping up to some length with arbitrary precision. Consider the true labeling function $f$ and another function $h$, which is equal to $f$ for all sequences of length up to $T$ and $f + c$ for larger lengths. If we use such arbitrary capacity SSMs as our hypothesis class, then this hypothesis class contains both $f$ and $h$. As a result, $h$ is a solution to equation 1 and it does not achieve length generalization. We can extend the same argument to compositional generalization as well.

### 3.4 VANILLA RECURRENT NEURAL NETWORKS

Standard RNNs have a non-linear recurrence unlike the linear recurrence studied in the previous section. We use the same notation as the previous section and only add an activation for non-linear recurrence. We illustrate the dynamics to show the generation of $x_{\leq t}$ and $y_{\leq t}$ below.

$$
\begin{aligned}
h_1 &= \sigma(Bx_1); \quad h_2 = \sigma(\Lambda h_1 + Bx_2); \quad \cdots, h_T = \sigma(\Lambda h_{T-1} + Bx_T) \\
y_1 &= \rho(h_1); \qquad y_2 = \rho(h_2); \quad \cdots\cdots\cdots, \qquad y_T = \rho(h_T),
\end{aligned}
\tag{4}
$$

**Assumption 8.** *Each function in the hypothesis class $\mathcal{H}$ is a vanilla RNN of the form equation 4, where the position-wise non-linearity is a single layer perceptron $\sigma \circ A$, and $\Lambda$, $B$ govern the hidden state dynamics (equation 4). $A, \Lambda, B$ are square invertible matrices, and $\sigma$ is the sigmoid activation.*

**Theorem 5.** *If $\mathcal{H}$ follows Assumption 8, and the realizability condition holds, i.e., $f \in \mathcal{H}$ and regular closedness condition in Assumption 2 holds, then the model trained to minimize the risk in equation 1 with $\ell_2$ loss (with $T \geq 2$) achieves length and compositional generalization.*

The hidden state estimated by the learned model, i.e., $\tilde{h}_t$, and the true hidden state $h_t$, bear a linear relationship (See Corollary 5 in Section C.4 for details), where the linear relationship is a permutation map. We extend Theorem 5 to discrete tokens in Theorem 13.

**High capacity RNNs** In our result above, similar to previous sections we showed that limited capacity RNNs can achieve length and compositional generalization. How about RNNs with arbitrary capacity, i.e., no constraint on $\Lambda$, $B$ and $\rho$? These systems can approximate sequence-to-sequence models to arbitrary precision (Sontag, 1992; Gühring et al., 2020). Hence, we can use the same argument as previous sections to argue that if $\mathcal{H}$ corresponds to RNNs with arbitrary capacity, then there exist solutions to equation 1 that do not achieve length and compositional generalization.

**Remark on proofs** Finally, we would like the reader to appreciate that our proofs follow different strategies in comparison to Wiedemer et al. (2023b); Dong & Ma (2022), due to the fact that we cater to sequence-to-sequence models. Consider the proofs in Wiedemer et al. (2023a), which reduce the solutions of equation 1 to solutions of set of ordinary differential equations, which under their assumptions are unique. That leads to exact identification in contrast to linear identification.

### 3.5 FINITE HYPOTHESIS CLASS

In the discussion so far, we have focused on different hypothesis class $\mathcal{H}$ of infinite size. In this section, we focus on finite hypothesis class, i.e., the set $\mathcal{H}$ has a finite size. We can construct such a finite hypothesis class for any architecture by restricting the parameter vectors (weights, biases etc.) to assume a finite set of values. Each possible parameter configuration denotes one distinct element in $\mathcal{H}$. Unlike the previous sections, we do not impose any futher restrictions on $\mathcal{H}$ other than the finite size. This allows us to consider arbitrary sequence to sequence models – RNNs, deep sets, transformers (e.g., with hard-coded positional encodings as in (Vaswani et al., 2017)) without restrictions on the depth and width as seen in the previous sections.

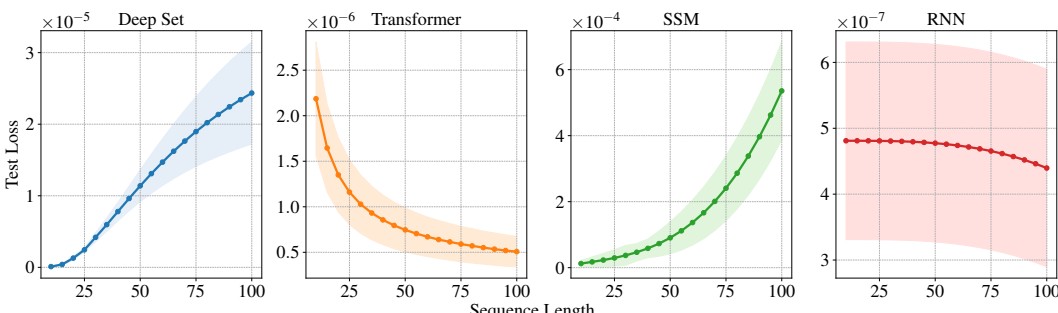

Figure 2: Length generalization: Test $\ell_2$ loss on sequences of different lengths. The models are trained only on sequences of length up to $T = 10$. All models achieve small error values $\approx 10^{-4} - 10^{-7}$ at all sequence lengths and thus length generalize. Since the error values are already quite small, the increasing or decreasing trends are not numerically significant.

**Theorem 6.** *If $\mathcal{H}$ is a finite hypothesis class, the realizability condition holds, i.e., $f \in \mathcal{H}$, then $\exists\, T_0 < \infty$ such that the model trained to minimize the risk in equation 1 with $\ell_2$ loss and $T > T_0$ achieves length generalization.*

The above theorem states that for a finite hypothesis class, length generalization is provably achieved provided the training length is sufficiently large. Observe that the above theorem only focuses on length generalization and does not apply to compositional generalization. In the above result, the value of the threshold on $T$, i.e., $T_0$, can be very large, and future work should consider quantifying bounds on $T_0$. In previous sections, where we had more structural restrictions on $\mathcal{H}$, the threshold on $T$ was two.

## 4 EXPERIMENTS

We present the empirical evaluation of compositional and length generalization capabilities of the architectures from the previous section. All the experiments are carried out in the realizable case where $f \in \mathcal{H}$, i.e., depending on the architecture in question, we use a random instance of the architecture to generate the labels. We train a model $h$ from the same architecture class to minimize the $\ell_2$ loss between $h$ and $f$. Under different scenarios, we ask if $h$ achieves length generalization and compositional generalization. We also seek to understand the relationship between the hidden representations of $h$ and hidden representations of $f$.

### 4.1 LENGTH GENERALIZATION

We sample sequences $x_{\leq t}$ of varying length with a maximum length of $T = 10$. Each token $x_i \sim \text{Uniform}[0, 1]^n$, where $n = 20$. The sequences are then fed to the labeling $f$, which comes from the hypothesis class of the architecture, to generate the labels. We minimize the empirical risk version of equation 1 over the same hypothesis class with $\ell_2$ loss. For evaluation, we present the $\ell_2$ loss on the test datasets. We also evaluate $R^2$ of linear regression between the learned hidden representations denoted $\psi(x_i)$ and true hidden representations $\phi(x_i)$ for all $x_i \in x_{<t}$ from the test dataset sequences. This metric is often used to evaluate the claims of linear identification (Khemakhem et al., 2020), i.e., the higher this value, the closer the linear relationship. We present results averaged over five seeds for models with *two* hidden layer MLPs for $\rho$ ($\phi$ is two hidden layer MLP for deep sets). Figure 2 shows a very small test loss of models on increasing sequence lengths when only trained with sequences of up to length $T = 10$, which is in agreement with Theorem 1-5. Further, in Figure 3, we show an exemplar sequence from test set and how the trained transformer tracks it. Table 1 shows the average of $R^2$ score of $\psi(x_i), \phi(x_i)$ across different positions $i$ at test time. These results demonstrate a linear relationship between learned and true hidden representations, which agrees with our theoretical claims. In Section D, we show that when realizability condition does not hold, i.e., $f \notin \mathcal{H}$, then length generalization is not achieved. We also present *additional experiments with discrete tokens, failures in the high capacity settings*, and other experimental details in Section D.

| Model | $R^2$ ($t = 20$) | $R^2$ ($t = 100$) |
|---|---|---|
| Deep set | $0.97 \pm 0.01$ | $0.97 \pm 0.01$ |
| Transformer | $0.99 \pm 0.01$ | $0.99 \pm 0.01$ |
| SSM | $0.99 \pm 0.01$ | $0.99 \pm 0.01$ |
| RNN | $0.99 \pm 0.01$ | $0.99 \pm 0.01$ |

| Model | Test Loss $\times 10^6$ | $R^2$ |
|---|---|---|
| Deep set | $0.08 \pm 0.02$ | $0.96 \pm 0.01$ |
| Transformer | $3.06 \pm 1.11$ | $1.00 \pm 0.00$ |
| SSM | $5.92 \pm 2.47$ | $1.00 \pm 0.00$ |
| RNN | $0.35 \pm 0.17$ | $0.96 \pm 0.01$ |

Table 1: Average test $R^2$ of true and learned hidden representations $\psi(x_i), \phi(x_i)$ across all positions $i$ at various lengths unseen during training. A strong linear relationship is observed for all models across lengths.

Table 2: Compositional generalization: Test $\ell_2$ loss and $R^2$ score for models with *two* hidden layers on sequences of length $T = 10$. A strong linear relationship is observed for all models for new sequences made of unseen token combinations.

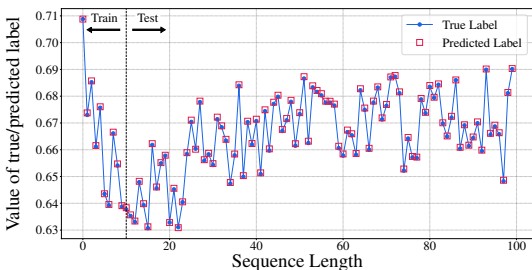

Figure 3: A transformer model with softmax attention with *two* hidden layer MLP for $\omega$ trained on sequences of length up to $T = 10$ length generalizes to sequences of length up to 100.

## 4.2 COMPOSITIONAL GENERALIZATION

For compositional generalization, we generate data following the illustration in Figure 1a. During training, we sample each component $k$ of a token from Uniform$[0, 1]$ and accept the sampled sequences that satisfy the following for all components $i$: $-0.5 \leq \sum_{j=1}^{T}(x_j^k - 0.5) \leq 0.5 \ \ \forall k$, where $x_j^k$ is the $k^{th}$ component of token $j$. During testing, we sample $x_{\leq t}$ from the complementary set of the training set, i.e., corners of hypercube $[0, 1]^{nt}$. We present the $\ell_2$ loss on the test dataset, as well as the mean $R^2$, where the results are averaged over 5 seeds. The rest of the details are the same as the previous section, i.e., $T = 10$, $n = 20$, $\rho$ is a two hidden layer MLP ($\phi$ is also a two hidden layer MLP for deep sets). Table 2 shows the test $\ell_2$ loss and $R^2$ scores for linear identification.

## 5 DISCUSSION AND LIMITATIONS

Our work is a step towards theoretical foundations of successes and failures of length and compositional generalization in sequence-to-sequence models. We prove simplified versions of the recently proposed RASP conjecture under weaker data diversity assumptions. In our analysis, we make certain simplifications, e.g., on the architectures considered, which motivates some of the important conjectures for future work. The main conjectures go as follows – a) **Conjecture 1:** Theorem 2 and 3 currently incorporate different non-linear attentions but not the softmax attention. We believe these guarantees on transformers extend to softmax attention given the experimental evidence in Section 4. b) **Conjecture 2:** Theorem 2 and 3 use one block of attention and one block of non-linearity. We believe that it is possible to extend these results to more expressive $\mathcal{H}$, e.g., with more alternating blocks. c) **Conjecture 3:** Our results focus on the generalization properties of all the possible solutions to risk minimization equation 1. However, in practice the optimization procedure may be biased towards a subset of those. Does accounting for the bias of optimization procedure give way to explaining the success of generalization in even higher capacity architectures?

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

APPENDIX

CONTENTS

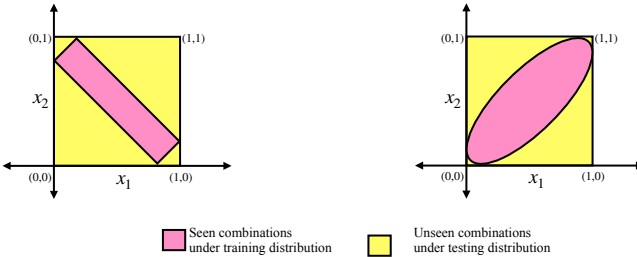

Figure 4: Illustration of observed support and its Cartesian product. These examples illustrate the support of the observed training data distribution can be much smaller than the Cartesian product of the support of the individual tokens.

## A    ILLUSTRATION OF THE TEST SUPPORT FOR COMPOSITIONAL GENERALIZATION

The notion of compositional generalization we study requires us to evaluate the model on the Cartesian product of the support of individual token distributions. In Figure 4, we give some additional examples besides the ones shown in Figure 1a to illustrate the difference between the Cartesian product set and the observed support. These examples illustrate the support of the observed training data distribution can be much smaller than the Cartesian product of the support of the individual tokens.

## B    SUPPLEMENT ON RELATED WORKS

We briefly discuss some other relevant works here, which could not be mentioned in the main body due to space constraints. In Schug et al. (2023), the authors exploit compositionality in the context of meta learning, where each task parameter is specified via a linear combination of some basis module parameters. They construct an approach that achieves provable compositional guarantees and outperforms meta-learning approaches such as MAML and ANIL. In a concurrent work Hou et al. (2024) propose an interesting scratch pad strategy inspired from the operation of Turing machines. They call this strategy Turing programs. The scratch pad emulates the operation of a Turing machine. The authors argue that there exist short RASP program ($O(n)$ length) that can simulate the operation of a Turing machine for sufficiently long number of steps ($O(\exp(n))$). Our current framework does not incorporate scratchpad strategies into it, and it is a promising future work to investigate provable length generalization guarantees with scratchpad.

## C    PROOFS

In all the results that follow, we work with standard topology in $\mathbb{R}^{nt}$, where $n$ is dimension of each token and $t$ is the sequence length. We remind the reader of the definition of a regular closed set – if a set is equal to the closure of its interior, then it is said to be a regular closed set. In all the results that follow, we either work with continuous random variables for which the Radon-Nikodym derivative of $X_{\leq t}$ is absolutely continuous w.r.t Lebesgue measure $\forall t$ or we work with discrete random variables for which the Radon-Nikodym derivative of $X_{\leq t}$ is absolutely continuous w.r.t counting measure $\forall t$.

**Lemma 1.** *Let $\mathcal{X} \subseteq \mathbb{R}^n$. If $f : \mathcal{X} \to \mathbb{R}^m$ and $g : \mathcal{X} \to \mathbb{R}^m$ are continuously differentiable functions that satisfy $f(x) = g(x)$ almost everywhere in $\mathcal{X}$, where $\mathcal{X}$ is a regular non-empty closed set, then $f(x) = g(x), \forall x \in \mathcal{X}$ and $\nabla f(x) = \nabla g(x), \forall x \in \mathcal{X}$, where $\nabla$ is the Jacobian w.r.t $x$.*

*Proof.* Let us consider the interior of $\mathcal{X}$ and denote it as $\mathcal{X}^{\text{int}}$. We first argue that the two functions $f$ and $g$ are equal at all points in the interior. Suppose there exists a point $x \in \mathcal{X}^{\text{int}}$ at which $f(x) \neq g(x)$. Consider a ball centered at $x$ of radius $r$ denoted as $B(x, r) \subset \mathcal{X}^{\text{int}}$ (such a ball exists as this point is in the interior of $\mathcal{X}$.). We argue that there exists at least one point $x_1$ in this ball at which $f(x_1) = g(x_1)$. If this were not the case, then the equality will not hold on the entire ball, which would contradict the condition that the equality $f(x) = g(x)$ can only be violated on a set of measure zero. Note this condition holds true for all $r > 0$. Suppose the distance of $x_1$ from $x$ is $r_1 \leq r$. Consider another ball with radius $r_2 < r_1$ and let $x_2 \in B(x, r_2)$ where the equality holds. By repeating this argument, we can construct a sequence $\{x_k\}_{k\in\mathbb{N}}$ that converges to $x$, where $\mathbb{N}$ is the set of natural numbers. On this sequence, the following conditions hold.

$$f(x_k) = g(x_k), \forall k \in \mathbb{N} \tag{5}$$

Further, from the continuity of $f$ and $g$ it follows that

$$\lim_{k\to\infty} f(x_k) = f(x), \lim_{k\to\infty} g(x_k) = g(x) \tag{6}$$

Combining the above two conditions, we get that $f(x) = g(x)$. This leads to a contradiction since we assumed that $f(x) \neq g(x)$. Thus there can be no such $x$ in the interior at which $f(x) \neq g(x)$. From this it follows that $f(x) = g(x)$ for all $x \in \mathcal{X}^{\text{int}}$. Now let us consider the closure of $\mathcal{X}^{\text{int}}$, which is $\mathcal{X}$ itself since it is a regular closed set. Every point $x \in \mathcal{X}$ in the closure can be expressed as limit of points in $\mathcal{X}^{\text{int}}$. Consider an $x \in \mathcal{X}$ and from the definition of regular closed set it follows that $\lim_{k\to\infty} x_k = x$, where $x_k \in \mathcal{X}^{\text{int}}$. We already know from the fact that $f$ and $g$ are equal in the interior

$$f(x_k) = g(x_k), \forall k \in \mathbb{N} \tag{7}$$

From the continuity of $f$ and $g$ it follows

$$\lim_{k\to\infty} f(x_k) = f(x), \lim_{k\to\infty} g(x_k) = g(x) \tag{8}$$

Combining the above two we get that $f(x) = g(x)$ for all $x \in \mathcal{X}$. After this we can use Lemma 6 from (Lachapelle et al., 2023) to conclude that $\nabla f(x) = \nabla g(x), \forall x \in \mathcal{X}$. We repeat their proof here for completeness. For all points in the interior of $\mathcal{X}$, it follows that $\nabla f(x) = \nabla g(x), \forall x \in \mathcal{X}^{\text{int}}$.

Now consider any point $x \in \mathcal{X}$. Since $\mathcal{X}$ is a regular closed set, $\lim_{k\to\infty} x_k = x$. Since each $x_k$ is in the interior of $\mathcal{X}$ it follows that

$$\nabla f(x_k) = \nabla g(x_k), \forall k \in \mathbb{N} \tag{9}$$

From the continuity of $\nabla f$ and $\nabla g$ it follows that

$$\lim_{k\to\infty} \nabla f(x_k) = \nabla f(x), \lim_{k\to\infty} \nabla g(x_k) = \nabla g(x) \tag{10}$$

Combining the above conditions, we get that $\nabla f(x) = \nabla g(x)$. This completes the proof.

$\square$

## C.1 DEEP SETS

In this section, we provide the proofs for length and compositional generalization for deep sets. We first provide the proof for Theorem 1, followed by Corollary 1, where we establish linear identification. We then present the discrete tokens counterpart to Theorem 1 in Theorem 7. In the next part of this section, we extend Theorem 1 with $\omega$ from $C^1$-diffeomorphism in Theorem 8.

We restate the theorems from the main body for convenience of the reader. In what follows, we remind the reader that we denote the labeling function $f(\mathcal{X}) = \rho(\sum_{x\in\mathcal{X}} \phi(x))$ and the function learned is denoted as $h(\mathcal{X}) = \omega(\sum_{x\in\mathcal{X}} \psi(x))$.

**Theorem 1.** *If $\mathcal{H}$ follows Assumption 1, the realizability condition holds, i.e., $f \in \mathcal{H}$, $\text{supp}(X_j) = [0,1]^n$, $\forall j \geq 1$, and the regular closedness condition in Assumption 2 holds, then the model trained to minimize the risk in equation 1 with $\ell_2$ loss generalizes to all sequences in the hypercube $[0,1]^{nt}$, $\forall t \geq 1$ and thus achieves length and compositional generalization.*

*Proof.* Consider any $h$ that solves equation 1. Since $\ell$ is $\ell_2$ loss and realizability condition holds, $f$ is one of the optimal solutions to equation 1. For all $x_{\leq T} \in \mathsf{supp}(X_{\leq T})$ except over a set of measure zero the following condition holds

$$h(x_{\leq T}) = f(x_{\leq T}). \tag{11}$$

The above follows from the fact that $h$ solves equation 1, i.e., $\mathbb{E}[\|h - f\|^2] = 0$ and from Theorem 1.6.6. (Ash & Doléans-Dade, 2000). Since $\mathsf{supp}(X_{\leq T})$ is regular closed, $f, h$ are both continuously differentiable, we can use Lemma 1, it follows that the above equality holds for all $x_{\leq T} \in \mathsf{supp}(X_{\leq T})$. From realizability condition it follows that true $f(x_{\leq T}) = \rho\Big(\sum_{j \leq T} \phi(x_j)\Big)$. We substitute the functional decomposition from Assumption 1 to get

$$\omega\Big(\sum_{j \leq T} \psi(x_j)\Big) = \rho\Big(\sum_{j \leq T} \phi(x_j)\Big). \tag{12}$$

$\omega$ and $\rho$ are both single layer perceptron with a bijective activation $\sigma$. We substitute the parametric form of $\omega$ and $\rho$ to obtain

$$\sigma\Big(A \sum_{j \leq T} \psi(x_j)\Big) = \sigma\Big(B \sum_{j \leq T} \phi(x_j)\Big) \implies A \sum_{j \leq T} \psi(x_j) = B \sum_{j \leq T} \phi(x_j). \tag{13}$$

The second equality in the above simplification follows from the fact that the activation $\sigma$ is bijective, the inputs to $\sigma$ are equal. We take the derivative of the expressions above w.r.t $x_r$ to get the following condition and equate them (follows from Lemma 1). For all $x_r \in \mathsf{supp}(X_r)$, i.e., $x_r \in [0, 1]^n$,

$$\nabla_{x_r}\Big(A \sum_{j \leq T} \psi(x_j)\Big) = \nabla_{x_r}\Big(B \sum_{j \leq T} \phi(x_j)\Big). \tag{14}$$

We drop the subscript $r$ to simplify the notation. Therefore, for all $x \in [0, 1]^n$

$$A \nabla_x \psi(x) = B \nabla_x \phi(x), \tag{15}$$

where $\nabla_x \psi(x)$ is the Jacobian of $\psi(x)$ w.r.t $x$ and $\nabla_x \phi(x)$ is the Jacobian of $\phi(x)$ w.r.t $x$. We now take the derivative w.r.t some component $x^k$ of vector $x = [x^1, \cdots, x^n]$. Denote the components other than $k$ as $x^{-k} = x \setminus x^k$. From the above condition, it follows that for all $x \in [0, 1]^n$

$$A \frac{\partial \psi(x)}{\partial x^k} = B \frac{\partial \phi(x)}{\partial x^k}. \tag{16}$$

Using fundamental theorem of calculus, we can integrate both sides for fixed $x^{-k}$ and obtain the following for all $x^k \in [0, 1]$,

$$A\psi(x^k, x^{-k}) = B\phi(x^k, x^{-k}) + C_k(x^{-k}) \implies A\psi(x) - B\phi(x) = C_k(x^{-k}). \tag{17}$$

The above condition is true of all $k \in \{1, \cdots, n\}$. Hence, we can deduce that for all $x \in [0, 1]^n$ and for $k \neq j$, where $j, k \in \{1, \cdots, d\}$,

$$A\psi(x) - B\phi(x) = C_k(x^{-k}) = C_j(x^{-j}). \tag{18}$$

Take the partial derivative of $C_k(x^{-k})$ and $C_j(x^{-j})$ w.r.t $x^j$ to obtain, for all $x^j \in [0, 1]$,

$$\frac{\partial C_k(x^{-k})}{\partial x^j} = \frac{\partial C_j(x^{-j})}{\partial x^j} = 0. \tag{19}$$

In the above simplification, we use the fact that $\forall x^j \in [0,1]$, $\frac{\partial C_j(x^{-j})}{\partial x^j} = 0$. Therefore, $C_k(x^{-k})$ cannot depend on $x^j$. We can apply the same condition on all $j \neq k$. As a result, $C_k(x^{-k})$ is a fixed constant vector denoted as $C$. We write this as

$$A\psi(x) = B\phi(x) + C. \tag{20}$$

Substitute the above into $A\sum_{j \leq T} \psi(x_j) = B\sum_{j \leq T} \phi(x_j)$ to obtain

$$B\sum_{j \leq T} \phi(x_j) + CT = B\sum_{j \leq T} \phi(x_j) \implies C = 0. \tag{21}$$

Therefore, we get

$$\forall x \in [0,1]^n, A\psi(x) = B\phi(x). \tag{22}$$

We now consider any sequence $x_{\leq \tilde{T}}$ from $[0,1]^{n\tilde{T}}$. The prediction made by $h$ is

$$h(x_{\leq \tilde{T}}) = \sigma\Big(A\sum_{j \leq \tilde{T}} \psi(x_j)\Big) = \sigma\Big(B\sum_{j \leq \tilde{T}} \phi(x_j)\Big) = f(x_{\leq \tilde{T}}). \tag{23}$$

We use equation 22 in the simplification above. From the above, we can conclude that $h$ continues to be optimal for distribution $\mathbb{P}_{X_{\leq \tilde{T}}}$.

$\square$

**Corollary 1.** *If $\mathcal{H}$ follows Assumption 1 with the condition that the output layer weight matrix is left invertible, the realizability condition holds, i.e., $f \in \mathcal{H}$, $\mathsf{supp}(X_j) = [0,1]^n$, $\forall j \geq 1$, and the regular closedness condition in Assumption 2 holds, then the model trained to minimize the risk in equation 1 with $\ell_2$ loss achieves linear identification. Further, under the stated conditions linear identification is necessary for compositional and length generalization.*

*Proof.* We follow the exact same steps as in the previous proof of Theorem 1 up to equation 22. We restate equation 22 below.

$$\begin{aligned} \forall x \in [0,1]^n, A\psi(x) &= B\phi(x) \\ \psi(x) &= A^{-1}B\phi(x) \end{aligned} \tag{24}$$

The above condition establishes linear identification, i.e., the learned model's representation for a token is a linear transform of the true model's representation. From the above, we can write that $x_{\leq \tilde{T}}$ from $[0,1]^{n\tilde{T}}$

$$\sum_{j \leq \tilde{T}} \psi(x_j) = A^{-1}B\sum_{j \leq \tilde{T}} \phi(x_j) \tag{25}$$

The above shows linear relationship holds for the entire sequence as well.

Now let us turn to the part on necessity. From the proof of previous theorem, we know that

$$\forall x_{\leq T} \in \mathsf{supp}(X_{\leq T}), \ \sigma\Big(A\sum_{j \leq T} \psi(x_j)\Big) = \sigma\Big(B\sum_{j \leq T} \phi(x_j)\Big) \implies \forall x \in [0,1]^n, A\psi(x) = B\phi(x) \tag{26}$$

Thus if $\forall x \in [0,1]^n, A\psi(x) = B\phi(x)$ is not true, then $\sigma\Big(A\sum_{j \leq T} \psi(x_j)\Big) = \sigma\Big(B\sum_{j \leq T} \phi(x_j)\Big)$ cannot be true either. Therefore, in the absence of linear identification neither length nor compositional generalization are achievable. $\square$

**Remarks** A few remarks and observations from the proof are in order. Firstly, observe that we do not require $\phi$ and $\psi$ to have the same output dimension for the above proof to go through. Secondly, in Theorem 1, we observe all the labels from $t = 1$ to $T$, i.e., $y_1$ to $y_T$. The result continues to hold if we only observe label at length $T$, i.e., $y_T$. Finally, we make an observation in this result, which would apply to all the subsequent theorems. The definition of compositional generalization requires generalization to the Cartesian product over sequences of length $T$, where $T$ is the training length. Since our model generalizes to the hypercube $[0, 1]^{nt}, \forall t$, we achieve compositional generalization even beyond the training lengths.

### C.1.1 EXTENDING THEOREM 1 TO DISCRETE TOKENS

In our discussion, we have focused on settings where the support of each token has a non-empty interior (Assumption 2). In practice of language modeling, we use discrete tokens and hence Assumption 2 does not hold anymore. In this section, we discuss the adaptation of results for deepsets to setting when the the support of tokens is a finite set.

**Assumption 9.** *The marginal support of token for all positions is the same and denoted as $\mathcal{X}$. The joint support of first and second token is $\mathcal{X} \times \mathcal{X}$.*

**Theorem 7.** *If $\mathcal{H}$ follows Assumption 1, the realizability condition holds, i.e., $f \in \mathcal{H}$, and Assumption 9 holds, then the model trained to minimize the risk in equation 1 with $\ell_2$ loss generalizes to all sequences in the hypercube $[0, 1]^{nt}$, $\forall t \geq 1$ and thus achieves length and compositional generalization.*

*Proof.* Consider any $h$ that solves equation 1. Since $\ell$ is $\ell_2$ loss and realizability condition holds, $f$ is one of the optimal solutions to equation 1. For all $x_{\leq T} \in \mathsf{supp}(X_{\leq T})$

$$h(x_{\leq T}) = f(x_{\leq T}). \tag{27}$$

The above follows from the fact that $h$ solves equation 1, i.e., $\mathbb{E}[\|h - f\|^2] = 0$ and the fact that tokens are discrete random vectors. From realizability condition it follows that true $f(x_{\leq T}) = \rho\Big( \sum_{j \leq T} \phi(x_j) \Big)$. We substitute the functional decomposition from Assumption 1 to get

$$\omega\Big( \sum_{j \leq T} \psi(x_j) \Big) = \rho\Big( \sum_{j \leq T} \phi(x_j) \Big). \tag{28}$$

$\omega$ and $\rho$ are both single layer perceptron with a bijective activation $\sigma$. We substitute the parametric form of $\omega$ and $\rho$ to obtain

$$\sigma\Big( A \sum_{j \leq T} \psi(x_j) \Big) = \sigma\Big( B \sum_{j \leq T} \phi(x_j) \Big) \implies A \sum_{j \leq T} \psi(x_j) = B \sum_{j \leq T} \phi(x_j). \tag{29}$$

The second equality in the above simplification follows from the fact that the activation $\sigma$ is bijective, the inputs to $\sigma$ are equal.

From Assumption 9, it follows that for all $x_1, x_2 \in \mathcal{X} \times \mathcal{X}$

$$A\phi(x_1) + A\phi(x_2) = B\psi(x_1) + B\psi(x_2) \tag{30}$$

Set $x_1 = x_2 = x$ (we can set this value due to Assumption 9) we get
$$\forall x \in \mathcal{X}, A\psi(x) = B\phi(x). \tag{31}$$

We now consider any sequence $x_{\leq \tilde{T}}$ from $\mathcal{X}^{\tilde{T}}$. The prediction made by $h$ is

$$h(x_{\leq \tilde{T}}) = \sigma\Big( A \sum_{j \leq \tilde{T}} \psi(x_j) \Big) = \sigma\Big( B \sum_{j \leq \tilde{T}} \phi(x_j) \Big) = f(x_{\leq \tilde{T}}). \tag{32}$$

We use equation 22 in the simplification above. From the above, we can conclude that $h$ continues to be optimal for distribution $\mathbb{P}_{X_{\leq \bar{T}}}$.

$\square$

### C.1.2 EXTENDING THEOREM 1 TO $\omega$ FROM $C^1$-DIFFEOMORPHISMS CLASS

**Assumption 10.** *Each function in $\mathcal{H}$ is expressed as $h(x_1, \cdots, x_i) = \omega(\sum_{j=1}^{i} \psi(x_j))$, where $\omega$ is a $C^1$-diffeomorphism.*

**Assumption 11.** *The joint support $\mathsf{supp}(X_{\leq i})$ is a regular closed set for all $i \leq T$. The support of all tokens is equal, i.e., $\mathsf{supp}(X_j) = [0,1]^n$, where $j \geq 1$. The support of $[\phi(X_1), \phi(X_2)]$ is $\mathbb{R}^{2m}$, where $\phi$ is the embedding function for the labeling function $f(\mathcal{X}) = \rho(\sum_{x \in \mathcal{X}} \phi(x))$.*

We provide a remark on the assumption and where it is used following the proof of the next theorem.

**Theorem 8.** *If $\mathcal{H}$ follows Assumption 10, the realizability condition holds, i.e., $f \in \mathcal{H}$, and a further assumption on the support (Assumption 11) holds, then the model trained to minimize the risk in equation 1 (with $T \geq 2$) with $\ell_2$ loss generalizes to all sequences in $[0,1]^{nt}, \forall t \geq 1$ and thus achieves length and compositional generalization.*

*Proof.* We start with the same steps as earlier proofs and equate the prediction of $h$ and $f$. We first use the fact $h(x_{\leq i}) = f(x_{\leq i})$ everywhere in the support. For all $x_{\leq i} \in \mathsf{supp}(X_{\leq i})$

$$\omega\Big(\sum_{j \leq i} \psi(x_j)\Big) = \rho\Big(\sum_{j \leq i} \phi(x_j)\Big) \implies \sum_{j \leq i} \psi(x_j) = \omega^{-1} \circ \rho\Big(\sum_{j \leq i} \phi(x_j)\Big) \implies$$
$$\sum_{j \leq i} \psi(x_j) = a\Big(\sum_{j \leq i} \phi(x_j)\Big), \tag{33}$$

where $a = \omega^{-1} \circ \rho$. In the above simplification, we used the parametric form for the true labeling function and the learned labeling function and use the invertibility of $\omega$. Let us consider the setting when $i = 1$. In that case summation involves only one term. Substitute $x_1 = x$. We obtain $\forall x \in [0,1]^n$,

$$\psi(x) = a(\phi(x)). \tag{34}$$

The above expression implies that $\psi$ bijectively identifies $\phi$. Let us consider the setting when $i = 2$. Substitute $x_1 = x$ and $x_2 = y$. We obtain

$$a(\phi(x)) + a(\phi(y)) = a\big(\phi(x) + \phi(y)\big). \tag{35}$$

We now use the that assumption $[\phi(x), \phi(y)]$ spans $\mathbb{R}^{2m}$, where $\phi(x)$ and $\phi(y)$ individually span $\mathbb{R}^m$. Substitute $\phi(x) = \alpha$ and $\phi(y) = \beta$. We obtain $\forall \alpha \in \mathbb{R}^m, \forall \beta \in \mathbb{R}^m$

$$a(\alpha) + a(\beta) = a\big(\alpha + \beta\big). \tag{36}$$

Observe that $a(0) = 0$ (substitute $\alpha = \beta = 0$ in the above).

We use equation 36 to show that $a$ is linear. To show that, we need to argue that $a(c\alpha) = ca(\alpha)$ as we already know $a$ satisfies additivity condition.

From the identity above, we want to show that equation 70 $a(p\alpha) = pa(\alpha)$, where $p$ is some integer.

Substitute $\beta = -\alpha$ in $a(\alpha + \beta) = a(\alpha) + a(\beta)$. We obtain $a(0) = a(\alpha) + a(-\alpha) \implies a(-\alpha) = -a(\alpha)$. Suppose $p$ is a positive integer. We simplify $a(p\alpha)$ as follows $a(\alpha + (p-1)\alpha) = a(\alpha) + a((p-1)\alpha)$. Repeating this simplification, we get $a(p\alpha) = pa(\alpha)$. Suppose $p$ is a negative integer. We can write $a(p\alpha) = a(-p \times -\alpha) = -pa(-\alpha)$. Since $a(-\alpha) = -a(\alpha)$, we get $a(p\alpha) = pa(\alpha)$.

Suppose $c$ is some rational number, i.e., $c = p/q$, where $p$ and $q$ are non-zero integers. We already know $a(p\alpha) = pa(\alpha)$. Further, we obtain

$a(q\frac{1}{q}\alpha) = qa(\frac{1}{q}\alpha) \implies a(\frac{1}{q}\alpha) = \frac{1}{q}a(\alpha)$, where $q$ is some integer.

Now combine these $a(p/q\alpha) = pa(1/q\alpha) = \frac{p}{q}a(\alpha)$. We have established the homogeneity condition for rationals.

We will now use the continuity of the function $a$ and density of rationals to extend the claim for irrationals. Suppose $c$ is some irrational. Define a sequence of rationals that approach $c$ (this follows from the fact that rationals are dense in $\mathbb{R}$).

$a(c\alpha) = a(\lim_{n\to\infty} q_n\alpha) = \lim_{n\to\infty} a(q_n\alpha)$.

In the second equality above, we use the definition of continuity ($a$ is continuous since composition of continuous functions is continuous). We can also use the property that we already showed for rationals to further simplify

$\lim_{n\to\infty} a(q_n\alpha) = a(\alpha)\lim_{n\to\infty} q_n = ca(\alpha)$.

Observe that $a : \mathbb{R}^m \to \mathbb{R}^m$ and for any $\alpha, \beta \in \mathbb{R}^m$ $a(\alpha + \beta) = a(\alpha) + a(\beta)$ and $a(c\alpha) = ca(\alpha)$. From the definition of a linear map it follows that $a$ is linear. As a result, we can write $\forall x \in [0, 1]^n$

$$\psi(x) = A(\phi(x)) \tag{37}$$

Observe that $a$ is invertible because both $\rho$ and $\omega$ are invertible. As a result, we know that $A$ is an invertible matrix. From this we get

$$\phi(x) = A^{-1}\psi(x) = C(\psi(x)) \tag{38}$$

For all $z \in \mathbb{R}^m$, we obtain

$$a(z) = \rho^{-1} \circ \omega(z) = Cz \implies \omega(z) = \rho(Cz)$$

Let us consider any sequence $x_{\leq \tilde{T}} \in [0, 1]^{n\tilde{T}}$. We use the above conditions

$$\omega\big(\sum_{j\leq\tilde{T}} \psi(x_j)\big) = \rho(C\sum_{j\leq\tilde{T}} \psi(x_j)) = \rho\big(\sum_{j\leq\tilde{T}} \phi(x_j)\big)$$

Thus we obtain length and compositional generalization.

$\square$

**Remark on Assumption 11**   In Assumption 11, we require that the support of $[\phi(X_1), \phi(X_2)]$ is $\mathbb{R}^{2m}$. This assumption is used in the proof in equation equation 36. We used this assumption to arrive at $a(\alpha+\beta) = a(\alpha)+a(\beta), \forall \alpha, \beta \in \mathbb{R}^m$. We then used continuity of $a$ to conclude $a$ is linear. Now suppose $[\phi(X_1), \phi(X_2)]$ is some subset $\mathcal{Z} \subseteq \mathbb{R}^{2m}$. We believe that it is possible to extend the result to more general $\mathcal{Z}$, it might still be possible to arrive at $a$ is linear. We leave this investigation to future work.

**Remark on expressivity under Assumption 10 and Assumption 11**   Assumption 11 requires $\omega$ is a $C^1$-diffeomorphism. Suppose the label is one dimensional, i.e., $m = 1$. From Assumption 11 output dimension of $\phi$ is restricted to be one dimensional. Consider the map $h(x_1, \cdots, x_i) = \rho(\sum_{j\leq i} \phi(x_j))$. The output dimension of $\phi$ is required to grow with sequence length to express all permutation invariant maps (See Theorem 7 in (Zaheer et al., 2017)). Thus by restricting the output dimension of $\phi$ to one, we cannot express all the permutation invariant maps.

**Multiplication operator** Consider the multiplication operator $y_i = \Pi_{j=1}^i x_i$, where each $x_i > 0$. Observe that we can rewrite this as $y_i = \exp(\sum_{j=1}^i \log(x_j))$. This operator is realizable on deep sets from hypothesis class described by Assumption 10 with $\omega = \exp$ and $\psi = \log$. In Assumption 11, we require the support of $[\phi(X_1), \phi(X_2)]$ to be $\mathbb{R}^2$. We let the support of $X_1$ and $X_2$ be $(0, \infty)$. In Assumption 11 we require that the support of each token was equal to $[0, 1]$. However, the proof of Theorem 8 still goes through even if support is $(0, \infty)$. Hence, we can use Theorem 8 to conclude that deep sets trained to predict the output of multiplication can multiply longer sequences and also multiply new token combinations.

## C.2 TRANSFORMERS

In this section, we provide the proofs for length and compositional generalization for transformers. We first provide the proof for Theorem 2, followed by Corollary 2, where we establish linear identification. We present an extention of Theorem 2 to incorporate positional encoding in Theorem 9. We then present the discrete tokens counterpart to Theorem 2 in Theorem 10. In the next part of this section, we extend Theorem 2 with $\omega$ from $C^1$-diffeomorphism in Theorem 3. Theorem 11 adapts Theorem 3 to incorporate positional encodings.

We restate the theorems from the main body for convenience of the reader. In what follows, we want to remind the reader we denote the labeling function $f(x_1, \cdots, x_i) = \rho(\sum_{j \leq i} \phi(x_i, x_j))$ and the function learned is denoted as $h(x_1, \cdots, x_i) = \omega(\sum_{j \leq i} \psi(x_i, x_j))$.

**Theorem 2.** *If $\mathcal{H}$ follows Assumption 3, the realizability condition holds, i.e., $f \in \mathcal{H}$, $\mathsf{supp}(X_i, X_j) = [0, 1]^{2n}$, $\forall i \neq j$ and the regular closedness condition in Assumption 2 holds, then the model trained to minimize the risk in equation 1 (with $T \geq 2$) with $\ell_2$ loss generalizes to all sequences in the hypercube $[0, 1]^{nt}$, $\forall t \geq 1$ and thus achieves length and compositional generalization.*

*Proof.* Consider any $h$ that solves equation 1. Since $\ell$ is $\ell_2$ loss and realizability condition holds, $f$ is one of the optimal solutions to equation 1. For all $i \leq T, x_{\leq i} \in \mathsf{supp}(X_{\leq i})$ except over a set of measure zero the following condition holds

$$h(x_{\leq i}) = f(x_{\leq i}). \tag{39}$$

The above follows from the fact that $h$ solves equation 1, i.e., $\mathbb{E}[\|h - f\|^2] = 0$ and from Theorem 1.6.6. (Ash & Doléans-Dade, 2000). Since $\mathsf{supp}(X_{\leq i})$ is regular closed, $f, h$ are both continuously differentiable, we can use Lemma 1, it follows that the above equality holds for all $x_{\leq i} \in \mathsf{supp}(X_{\leq i})$. From realizability condition it follows that true $f(x_{\leq i}) = \rho\Big(\sum_{k \leq i} \phi(x_i, x_k)\Big)$. We substitute the parametric forms from Assumption 3 to get

$$\omega\Big(\sum_{k \leq i} \frac{1}{i} \cdot \psi(x_i, x_k)\Big) = \rho\Big(\sum_{k \leq i} \frac{1}{i} \cdot \phi(x_i, x_k)\Big). \tag{40}$$

Since $\omega$ and $\rho$ are single layer perceptron with bijective activation $\sigma$. We substitute the parametric form of $\omega$ and $\rho$ to obtain the following condition. For all $x_{\leq i} \in \mathsf{supp}(X_{\leq i})$,

$$\sigma\Big(A \sum_{k \leq i} \frac{1}{i} \cdot \psi(x_i, x_k)\Big) = \sigma\Big(B \sum_{k \leq i} \frac{1}{i} \cdot \phi(x_i, x_k)\Big) \implies A \sum_{k \leq i} \psi(x_i, x_k) = B \sum_{k \leq i} \phi(x_i, x_k). \tag{41}$$

The second equality follows from the fact that the activation $\sigma$ is bijective and hence the inputs to $\sigma$ are equal. We take the derivative of the expressions above w.r.t $x_j$ to get the following (follows from Lemma 1). For $j < i$ (there exists a $j < i$ as $T \geq 2$ and we can set $i \geq 2$) and for all $x_j \in \mathsf{supp}(X_j)$, i.e., $x_j \in [0, 1]^n$,

$$\nabla_{x_j}\Big(A\sum_{k\leq i}\psi(x_i,x_k)\Big) = \nabla_{x_j}\Big(B\sum_{k\leq i}\phi(x_i,x_k)\Big) \implies$$
$$A\nabla_{x_j}\psi(x_i,x_j) = B\nabla_{x_j}\phi(x_i,x_j),$$

(42)

where $\nabla_{x_j}\psi(x_i,x_j), \nabla_{x_j}\phi(x_i,x_j)$ are the Jacobians of $\psi$ and $\phi$ w.r.t $x_j$ for a fixed $x_i$. Note that $A\nabla_{x_j}\psi(x_i,x_j) = B\nabla_{x_j}\phi(x_i,x_j)$ holds for all $x_i \in [0,1]^n, x_j \in [0,1]^n$ (here we use the fact that joint support of every pair of tokens spans $2n$ dimensional unit hypercube assumed in the Theorem 9). In this equality, we now consider the derivative w.r.t some component $x_j^k$ of $x_j$. Denote the remaining components as $x_j^{-k}$. From the above condition it follows that for all $x_i \in [0,1]^n, x_j \in [0,1]^n$,

$$A\frac{\partial \psi(x_i,x_j)}{\partial x_j^k} = B\frac{\partial \phi(x_i,x_j)}{\partial x_j^k}.$$

(43)

Using fundamental theorem of calculus, we can integrate both sides for fixed $x_j^{-k}$ and obtain the following for all $x_j^k \in [0,1]$,

$$A\psi\big(x_i, [x_j^k, x_j^{-k}]\big) = B\phi\big(x_i, [x_j^k, x_j^{-k}]\big) + C_k\big(x_i, x_j^{-k}\big) =$$
$$A\psi(x_i,x_j) = B\phi(x_i,x_j) + C_k(x_i, x_j^{-k}).$$

(44)

The same condition is true of all $k$. Hence, $\forall x_i \in [0,1]^d, \forall x_j \in [0,1]^d$ and for $k \neq q$, where $q,k \in \{1,\cdots,d\}$,

$$A\psi(x_i,x_j) - B\phi(x_i,x_j) = C_k(x_i, x_j^{-k}) = C_q(x_i, x_j^{-q}).$$

(45)

Take the partial derivative of both sides w.r.t $x_j^q$ to obtain, $\forall x_j^q \in [0,1]$,

$$\frac{\partial C_k(x_i, x_j^{-k})}{\partial x_j^q} = \frac{\partial C_q(x_i, x_j^{-q})}{\partial x_j^q} = 0.$$

(46)

Therefore, $C_k(x_i, x_j^{-k})$ cannot depend on $x_j^q$. We can apply the same condition on all $q \neq k$. As a result, $C_k(x_i, x_j^{-k})$ is only a function of $x_i$ denoted as $C(x_i)$. Therefore, for $j < i$ and for all $x_i \in [0,1]^n, x_j \in [0,1]^n$

$$A\psi(x_i,x_j) = B\phi(x_i,x_j) + C(x_i).$$

(47)

If we substitute $x_i = x_j = x$, then the above equality extends for $i = j$ and thus we get

$$A\psi(x_i,x_i) = B\phi(x_i,x_i) + C(x_i).$$

(48)

Substitute the above equation 47, equation 48 into $A\sum_{k\leq i}\psi(x_i,x_k) = B\sum_{k\leq i}\phi(x_i,x_k)$ to obtain

$$B\sum_{k\leq i}\phi(x_i,x_k) + (i)C(x_i) = B\sum_{k\leq i}\phi(x_i,x_k) \implies C(x_i) = 0.$$

(49)

Thus we obtain

$$\forall x_i \in [0,1]^n, x_j \in [0,1]^n \quad A\psi(x_i,x_j) = B\phi(x_i,x_j).$$

(50)

We now consider any sequence $x_{\leq \tilde{T}} \in [0,1]^{n\tilde{T}}$. The prediction made by $h$ is

$$h(x_{\leq \tilde{T}}) = \sigma\Big(A \sum_{j \leq \tilde{T}} \psi(x_{\tilde{T}}, x_j)\Big) = \sigma\Big(B \sum_{j \leq \tilde{T}} \phi(x_{\tilde{T}}, x_j)\Big) = f(x_{\leq \tilde{T}}) \tag{51}$$

We use equation 50 in the simplification above. From the above, we can conclude that $h$ continues to be optimal for all sequences in $[0, 1]^{n\tilde{T}}$.

$\square$

**Corollary 2.** *If $\mathcal{H}$ follows Assumption 3 with the condition that the output layer weight matrix is left invertible, the realizability condition holds, i.e., $f \in \mathcal{H}$, $\mathrm{supp}(X_i, X_j) = [0, 1]^{2n}$, $\forall i \neq j$ and the regular closedness condition in Assumption 2 holds, then the model trained to minimize the risk in equation 1 (with $T \geq 2$) with $\ell_2$ loss achieves linear identification. Further, linear identification is necessary for both length and compositional generalization.*

*Proof.* We follow the exact same steps as in the previous proof of Theorem 2 up to equation 50. We restate equation 50 below.

$$\forall x_i \in [0, 1]^n, x_j \in [0, 1]^n \ \ A\psi(x_i, x_j) = B\phi(x_i, x_j)$$
$$\psi(x_i, x_j) = A^{-1} B\phi(x_i, x_j) \tag{52}$$

In the second step above, we use left invertibility of $A$. The above condition establishes linear identification, i.e., the learned model's representation is a linear transform of the true model's representation. From this we obtain that for any sequence $x_{\leq \tilde{T}} \in [0, 1]^{n\tilde{T}}$

$$\sum_{j \leq \tilde{T}} \psi(x_{\tilde{T}}, x_j) = A^{-1} B\Big( \sum_{j \leq \tilde{T}} \psi(x_{\tilde{T}}, x_j)\Big) \tag{53}$$

The above establishes a linear relationship between the learned representation of the sequence and the representation of the sequence under the true model. Now let us turn to the part on necessity. From the proof of previous theorem, we know that

$$\omega\Big( \sum_{k \leq i} \frac{1}{i}\cdot\psi(x_i, x_k)\Big) = \rho\Big( \sum_{k \leq i} \frac{1}{i}\cdot\phi(x_i, x_k)\Big) \implies \forall x_i \in [0, 1]^n, x_j \in [0, 1]^n \ \ A\psi(x_i, x_j) = B\phi(x_i, x_j) \tag{54}$$

Thus from the above it follows that in the absence of linear identification neither length nor compositional generalization are achievable. $\square$

**On the absence of labels at all lengths from $t = 1$ to $t = T$** A few important remarks are to follow. In the proof above, we do not require to observe all the labels from $t = 1$ to $t = T$, where $T \geq 2$. The proof goes through provided we observe data at two different lengths.

### C.2.1 EXTENSION OF THEOREM 2 TO INCORPORATE POSITIONAL ENCODINGS

In what follows, we extend the above result (Theorem 2) to incorporate positional encoding. We start with extension of the hypothesis class to incorporate positional encoding.

**Assumption 12.** *Each function in the hypothesis class $\mathcal{H}$ used by the learner is given as $h(x_1, \cdots, x_i) = \omega\Big( \sum_{j \leq i} \frac{1}{i}\psi_{i-j}(x_i, x_j)\Big)$, where $\omega$ is a single layer perceptron with continuously differentiable bijective activation (e.g., sigmoid) and each $\psi_k$ is a map that is differentiable. Also, $\psi_k = 0$ for $k \geq T_{\max}$, i.e., two tokens that are sufficiently far apart do not interact.*

In the above assumption, we incorporate relative positional encodings by making the function $\psi_{i-j}$ depend on the relative positional difference between token $x_i$ and token $x_j$. We would like to emphasize the reasons why we assume that the tokens that are sufficiently far apart do not interact. Suppose $T_{\max} = \infty$, which implies tokens at all positions interact. As a result, during training since

we only see sequences of finite length $T$, we will not see the effect of interactions of tokens that are separated at a distance larger than $T$ on the data generation, which makes it impossible to learn anything about $\phi_{i-j}$, where $i - j \geq T - 1$.

In the theorem that follows, we show that we can achieve length and compositional generalization for the above hypothesis class.

**Theorem 9.** *If $\mathcal{H}$ follows Assumption 12, the realizability condition holds, i.e., $f \in \mathcal{H}$, $\mathsf{supp}(X_i, X_j) = [0, 1]^{2n}$, $\forall i \neq j \in \{1, \cdots, \infty\}$, the regular closedness condition in Assumption 2 holds and $T \geq T_{\max} \geq 2$, then the model trained to minimize the risk in equation 1 with $\ell_2$ loss generalizes to all sequences in the hypercube $[0, 1]^{nt}$, $\forall t$ and thus achieves length and compositional generalization.*

*Proof.* Consider any $h$ that solves equation 1. Since $\ell$ is $\ell_2$ loss and realizability condition holds, $f$ is one of the optimal solutions to equation 1. For all $i \leq T$ and for all $x_{\leq i} \in \mathsf{supp}(X_{\leq i})$ except over a set of measure zero the following condition holds

$$h(x_{\leq i}) = f(x_{\leq i}). \tag{55}$$

The above follows from the fact that $h$ solves equation 1, i.e., $\mathbb{E}[\|h - f\|^2] = 0$ and from Theorem 1.6.6. (Ash & Doléans-Dade, 2000). Since $\mathsf{supp}(X_{\leq i})$ is regular closed, $f, h$ are both continuously differentiable, we can use Lemma 1, it follows that the above equality holds for all $x_{\leq i} \in \mathsf{supp}(X_{\leq i})$. From realizability condition it follows that true $f(x_{\leq i}) = \rho\Big( \sum_{k \leq i} \phi_{i-k}(x_i, x_k) \Big)$. We substitute the parametric forms from Assumption 3 to get

$$\omega\Big( \sum_{k \leq i} \frac{1}{i} \cdot \psi_{i-k}(x_i, x_k) \Big) = \rho\Big( \sum_{k \leq i} \frac{1}{i} \cdot \phi_{i-k}(x_i, x_k) \Big). \tag{56}$$

Since $\omega$ and $\rho$ are single layer perceptron with bijective activation $\sigma$. We substitute the parametric form of $\omega$ and $\rho$ to obtain the following condition. For all $x_{\leq i} \in \mathsf{supp}(X_{\leq i})$,

$$\sigma\Big(A \sum_{k \leq i} \frac{1}{i} \cdot \psi_{i-k}(x_i, x_k) \Big) = \sigma\Big(B \sum_{k \leq i} \frac{1}{i} \cdot \phi_{i-k}(x_i, x_k) \Big) \implies$$

$$A \sum_{k \leq i} \psi_{i-k}(x_i, x_k) = B \sum_{k \leq i} \phi_{i-k}(x_i, x_k). \tag{57}$$

The second equality follows from the fact that the activation $\sigma$ is bijective and hence the inputs to $\sigma$ are equal. We take the derivative of the expressions above w.r.t $x_j$ to get the following (follows from Lemma 1). The equality holds true for all $i \leq T$.

From the above, we can use $i = 1$ and obtain

$$A\psi_0(x_1, x_1) = B\phi_0(x_1, x_1), \forall x_1 \in [0, 1]^n.$$

From $i = 2$, we obtain

$$A\psi_0(x_2, x_2) + A\psi_1(x_2, x_1) = B\phi_0(x_2, x_2) + B\phi_1(x_2, x_1), \forall x_1 \in [0, 1]^n, x_2 \in [0, 1]^n$$

Combining the two conditions we get

$$A\psi_1(x_2, x_1) = B\phi_1(x_2, x_1), \forall x_1 \in [0, 1]^n, x_2 \in [0, 1]^n.$$

We can use this argument and arrive at

$$A\psi_{i-1}(x_i, x_1) = B\phi_{i-1}(x_i, x_1), \forall x_i \in [0, 1]^n, x_1 \in [0, 1]^n, \forall i \leq T.$$

Thus we obtain

$$\forall i - j \leq T - 1, \forall x_i \in [0,1]^n, x_j \in [0,1]^n, \quad A\psi_{i-j}(x_i, x_j) = B\phi_{i-j}(x_i, x_j). \tag{58}$$

From Assumption 12 and $T \geq T_{\mathsf{max}}$, we already know that

$$\forall i - j \geq T, \forall x_i \in [0,1]^n, x_j \in [0,1]^n, \quad A\psi_{i-j}(x_i, x_j) = B\phi_{i-j}(x_i, x_j) = 0. \tag{59}$$

If $A$ is left invertible, then the above condition implies that linear representation identification is necessary for both compositional and length generalization.

We now consider any sequence $x_{\leq \tilde{T}} \in [0,1]^{n\tilde{T}}$. The prediction made by $h$ is

$$h(x_{\leq \tilde{T}}) = \sigma\Big(A \sum_{j \leq \tilde{T}} \psi_{\tilde{T}-j}(x_{\tilde{T}}, x_j)\Big) = \sigma\Big(B \sum_{j \leq \tilde{T}} \phi_{\tilde{T}-j}(x_{\tilde{T}}, x_{\hat{j}})\Big) = f(x_{\leq \tilde{T}}) \tag{60}$$

We use equation 50 in the simplification above. From the above, we can conclude that $h$ continues to be optimal for all sequences in $[0,1]^{n\tilde{T}}$.

$\square$

### C.2.2  EXTENDING THEOREM 2 TO DISCRETE TOKENS

In the above result we used Assumption 2. In practice of language modeling, we use discrete tokens and hence Assumption 2 does not hold anymore. In this section, we discuss the adaptation of results for transformers to setting when the the support of tokens is a finite set.

**Assumption 13.** *The marginal support of token for all positions is the same and denoted as $\mathcal{X}$. The joint support of first three tokens is $\mathcal{X} \times \mathcal{X} \times \mathcal{X}$.*

**Theorem 10.** *If $\mathcal{H}$ follows Assumption 3, the realizability condition holds, i.e., $f \in \mathcal{H}$, and Assumption 13 holds, then the model trained to minimize the risk in equation 1 (with $T \geq 2$) with $\ell_2$ loss generalizes to all sequences in the hypercube $[0,1]^{nt}$, $\forall t \geq 1$ and thus achieves length and compositional generalization.*

*Proof.* Consider any $h$ that solves equation 1. Since $\ell$ is $\ell_2$ loss and realizability condition holds, $f$ is one of the optimal solutions to equation 1. For all $i \leq T, x_{\leq i} \in \mathsf{supp}(X_{\leq i})$ the following condition holds

$$h(x_{\leq i}) = f(x_{\leq i}). \tag{61}$$

The above follows from the fact that $h$ solves equation 1, i.e., $\mathbb{E}[\|h - f\|^2] = 0$ and from the fact that the tokens are discrete random vectors. From realizability condition it follows that true $f(x_{\leq i}) = \rho\Big(\sum_{k \leq i} \phi(x_i, x_k)\Big)$. We substitute the parametric forms from Assumption 3 to get

$$\omega\Big(\sum_{k \leq i} \frac{1}{i} \cdot \psi(x_i, x_k)\Big) = \rho\Big(\sum_{k \leq i} \frac{1}{i} \cdot \phi(x_i, x_k)\Big). \tag{62}$$

Since $\omega$ and $\rho$ are single layer perceptron with bijective activation $\sigma$. We substitute the parametric form of $\omega$ and $\rho$ to obtain the following condition. For all $x_{\leq i} \in \mathsf{supp}(X_{\leq i})$,

$$\sigma\Big(A \sum_{k \leq i} \frac{1}{i} \cdot \psi(x_i, x_k)\Big) = \sigma\Big(B \sum_{k \leq i} \frac{1}{i} \cdot \phi(x_i, x_k)\Big) \implies A \sum_{k \leq i} \psi(x_i, x_k) = B \sum_{k \leq i} \phi(x_i, x_k). \tag{63}$$

The second equality follows from the fact that the activation $\sigma$ is bijective and hence the inputs to $\sigma$ are equal.

From Assumption 13, it follows that for all $x_1, x_2, x_3 \in \mathcal{X} \times \mathcal{X} \times \mathcal{X}$

$$A\psi(x_3, x_1) + A\psi(x_3, x_2) = B\phi(x_3, x_1) + B\phi(x_3, x_2) \tag{64}$$

Set $x_1 = x_2$ (we can do so owing to Assumption 13).

Thus we obtain

$$\forall x_i \in \mathcal{X}, x_j \in \mathcal{X} \quad A\psi(x_i, x_j) = B\phi(x_i, x_j). \tag{65}$$

We now consider any sequence $x_{\leq \tilde{T}} \in \mathcal{X}^{\tilde{T}}$. The prediction made by $h$ is

$$h(x_{\leq \tilde{T}}) = \sigma\Big(A \sum_{j \leq \tilde{T}} \psi(x_{\tilde{T}}, x_j)\Big) = \sigma\Big(B \sum_{j \leq \tilde{T}} \phi(x_{\tilde{T}}, x_j)\Big) = f(x_{\leq \tilde{T}}) \tag{66}$$

We use equation 65 in the simplification above. From the above, we can conclude that $h$ continues to be optimal for all sequences in $[0, 1]^{n\tilde{T}}$.

$\square$

### C.2.3 EXTENDING THEOREM 2 TO $\omega$ FROM $C^1$-DIFFEOMORPHISMS

**Theorem 3.** *If $\mathcal{H}$ follows Assumption 4, the realizability condition holds, i.e., $f \in \mathcal{H}$, and a further assumption on the support (Assumption 5) holds, then the model trained to minimize the risk in equation 1 (with $T \geq 3$) with $\ell_2$ loss generalizes to all sequences in $[0, 1]^{nt}, \forall t \geq 1$ and thus achieves length and compositional generalization.*

*Proof.* We start with the same steps as earlier proofs and equate the prediction of $h$ and $f$. We first use the fact $h(x_{\leq i}) = f(x_{\leq i}), \forall i \leq T$ almost everywhere in the support. We can use the continuity of $h, f$ and regular closedness of the support to extend the equality to all points in the support (follows from the first part of Lemma 1) to obtain the following. For all $x_{\leq i} \in \mathsf{supp}(X_{\leq i})$

$$\omega\Big(\sum_{j<i} \frac{1}{i-1} \cdot \psi(x_i, x_j)\Big) = \rho\Big(\sum_{j<i} \frac{1}{i-1} \cdot \phi(x_i, x_j)\Big) \implies$$

$$\sum_{j<i} \frac{1}{i-1}\psi(x_i, x_j) = \omega^{-1} \circ \rho\Big(\sum_{j<i} \frac{1}{i-1} \cdot \phi(x_i, x_j)\Big) \implies \tag{67}$$

$$\sum_{j<i} \frac{1}{i-1}\psi(x_i, x_j) = a\Big(\sum_{j<i} \frac{1}{i-1}\phi(x_i, x_j)\Big),$$

where $a = \omega^{-1} \circ \rho$. In the above simplification, we used the parametric form for the true labeling function and the learned labeling function and use the invertibility of $\omega$. Let us consider the setting when $i = 2$. In that case summation involves only one term. Substitute $x_1 = y$ and $x_2 = x$. We obtain $\forall x \in [0, 1]^n, y \in [0, 1]^n$,

$$\psi(x, y) = a(\phi(x, y)). \tag{68}$$

The above expression implies that $\psi$ bijectively identifies $\phi$. Let us consider the setting when $i = 3$ (this is possible since $T \geq 3$). We substitute $x_3 = x, x_2 = y, x_1 = z$ and obtain

$$\frac{1}{2}\Big[a(\phi(x, y)) + a(\phi(x, z))\Big] = a\Big(\frac{1}{2}\big(\phi(x, y) + \phi(x, z)\big)\Big). \tag{69}$$

Substitute $\phi(x, y) = \alpha$ and $\phi(x, z) = \beta$. In the simplifcation that follows, we use the that assumption $[\phi(x, y), \phi(x, z)]$ spans $\mathbb{R}^{2m}$, where $\phi(x, y)$ and $\phi(x, z)$ individually span $\mathbb{R}^m$.

$$\frac{1}{2}(a(\alpha) + a(\beta)) = a\left(\frac{1}{2}(\alpha + \beta)\right). \tag{70}$$

Observe that $a(0) = 0$ because $\omega^{-1} \circ \rho(0) = 0$ because $\omega^{-1}(0) = \rho(0) = 0$.

$$\frac{1}{2}(a(2\alpha) + a(0)) = a\left(\frac{1}{2}(2\alpha + 0)\right)$$
$$a(2\alpha) = 2a(\alpha) \tag{71}$$

Next, substitute $\alpha$ with $2\alpha$ and $\beta$ with $2\beta$ in equation 70 to obtain

$$\frac{1}{2}(a(2\alpha) + a(2\beta)) = a\left(\frac{1}{2}(2\alpha + 2\beta)\right)$$
$$a(\alpha + \beta) = a(\alpha) + a(\beta) \tag{72}$$

We use equation 72 to show that $a$ is linear. To show that, we need to argue that $a(c\alpha) = ca(\alpha)$ as we already know $a$ satisfies additivity condition.

Suppose $c$ is some rational number, i.e., $c = p/q$, where $p$ and $q$ are non-zero integers.

From the identity it is clear that $a(p\alpha) = pa(\alpha)$, where $p$ is some integer.

$a(q\frac{1}{q}\alpha) = qa(\frac{1}{q}\alpha) \implies a(\frac{1}{q}\alpha) = \frac{1}{q}a(\alpha)$, where $q$ is some integer.

Now combine these $a(p/q\alpha) = pa(1/q\alpha) = \frac{p}{q}a(\alpha)$. We have established the homogeneity condition for rationals.

We will now use the continuity of the function $a$ and density of rationals to extend the claim for irrationals. Suppose $c$ is some irrational. Define a sequence of rationals that approach $c$ (this follows from the fact that rationals are dense in $\mathbb{R}$).

$a(c\alpha) = a(\lim_{n\to\infty} q_n\alpha) = \lim_{n\to\infty} a(q_n\alpha)$.

In the second equality above, we use the definition of continuity ($a$ is continuous since composition of continuous functions is continuous). We can also use the property that we already showed for rationals to further simplify

$\lim_{n\to\infty} a(q_n\alpha) = a(\alpha)\lim_{n\to\infty} q_n = ca(\alpha)$.

Observe that $a : \mathbb{R}^m \to \mathbb{R}^m$ and for any $\alpha, \beta \in \mathbb{R}^m$ $a(\alpha + \beta) = a(\alpha) + a(\beta)$ and $a(c\alpha) = ca(\alpha)$. From the definition of a linear map it follows that $a$ is linear. As a result, we can write $\forall x \in [0,1]^n, y \in [0,1]^n$

$$\psi(x, y) = A(\phi(x, y)) \tag{73}$$

Observe that $a$ is invertible because both $\rho$ and $\omega$ are invertible. As a result, we know that $A$ is an invertible matrix. From this we get

$$\forall x \in [0,1]^n, y \in [0,1]^n, \phi(x, y) = A^{-1}\psi(x, y) = C(\psi(x, y)) \tag{74}$$

For all $z \in \mathbb{R}^m$, we obtain

$$a(z) = \rho^{-1} \circ \omega(z) = Cz \implies \omega(z) = \rho(Cz)$$

Let us consider any sequence $x_{\leq \tilde{T}} \in [0,1]^{n\tilde{T}}$. We use the above conditions

$$\omega\left(\sum_{j<\tilde{T}} \psi(x_{\tilde{T}}, x_j)\right) = \rho\left(C\sum_{j<\tilde{T}} \psi(x_{\tilde{T}}, x_j)\right) = \rho\left(\sum_{j<\tilde{T}} \phi(x_{\tilde{T}}, x_j)\right).$$

Thus we obtain length and compositional generalization.

$\square$

**Corollary 3.** *If $\mathcal{H}$ follows Assumption 4, the realizability condition holds, i.e., $f \in \mathcal{H}$, and a further assumption on the support (Assumption 5) holds, then the model trained to minimize the risk in equation 1 (with $T \geq 3$) with $\ell_2$ loss achieves linear identification. Further, under the stated conditions linear identification is necessary for both length and compositional generalization.*

*Proof.* We follow the exact same steps as in the previous proof of Theorem 3 up to equation 74. We restate equation 74 below.

$$\forall x \in [0,1]^n, y \in [0,1]^n, \phi(x,y) = C(\psi(x,y)) \tag{75}$$

The above condition directly implies linear identification. We can use this to obtain that for any sequence $x_{\leq \tilde{T}} \in [0,1]^{n\tilde{T}}$

$$\sum_{j < \tilde{T}} \psi(x_{\tilde{T}}, x_j) = \sum_{j < \tilde{T}} \phi(x_{\tilde{T}}, x_j) \tag{76}$$

To show necessity of linear identification, from the proof of Theorem 3 observe that

$$\forall i \leq T, \forall x_{\leq i} \in \mathsf{supp}(X_{\leq i}), \; \omega\Big(\sum_{j < i} \frac{1}{i-1} \cdot \psi(x_i, x_j)\Big) = \rho\Big(\sum_{j < i} \frac{1}{i-1} \cdot \phi(x_i, x_j)\Big) \implies$$
$$\forall x \in [0,1]^n, y \in [0,1]^n, \phi(x,y) = C(\psi(x,y)) \tag{77}$$

Thus from the above it follows that in the absence of linear identification neither length nor compositional generalization are achievable. $\square$

**On absence of labels at all lengths from $1$ to $T$**  We argue that the above proof can be adapted to the setting where we do not observe labels at all lengths from $1$ to $T$. Suppose we only observe label at length $T$. Take equation equation 67 and substitute $x_i = x$ and $x_j = y$ for all $j < i$ to obtain the same condition as equation equation 68. Suppose $T$ is odd and larger than or equal to 3. Fix $x_i = x, x_{2j-1} = y, \forall j \in \{1, \cdots, (T-1)/2\}, x_{2j} = z, \forall j \in \{1, \cdots, (T-1)/2\}$. We obtain the same condition as equation equation 69. Rest of the proof can be adapted using a similar line of reasoning.

**Remark on Assumption 4**  We require that the support of $[\phi(X_1, X_2), \phi(X_1, X_3)]$ is $\mathbb{R}^{2m}$. This assumption is used in the proof in equation equation 72. We used this assumption to arrive at $a(\alpha + \beta) = a(\alpha) + a(\beta), \forall \alpha, \beta \in \mathbb{R}^m$. We then used continuity of $a$ to conclude $a$ is linear. Now suppose $[\phi(X_1, X_2), \phi(X_1, X_3)]$ is some subset $\mathcal{Z} \subseteq \mathbb{R}^{2m}$. We believe that it is possible to extend the result to more general $\mathcal{Z}$, it might still be possible to arrive at $a$ is linear. We leave this investigation to future work.

### C.2.4   EXTENDING THEOREM 3 TO INCORPORATE POSITIONAL ENCODINGS

We next present the result when $\omega$ is continuously differentiable and invertible.

**Assumption 14.** *Each function in the hypothesis class $\mathcal{H}$ used by the learner is given as $h(x_1, \cdots, x_i) = \omega\Big(\sum_{j \leq i} \psi_{i-j}(x_i, x_j)\Big)$, where $\omega$ is a $C^1$-diffeomorphism. Also, $\psi_{i-j} = 0$ for $i - j > T_{\mathsf{max}} - 1$, i.e., two tokens that are sufficiently far apart do not interact. For all $k \leq T_{\mathsf{max}} - 1$ each $x \in [0,1]^n, \exists y \in [0,1]^n$ we $\psi_k(x,y) = 0$.*

In the theorem that follows, we require the support of training distribution under consideration is already sufficiently diverse and hence we only seek to prove length generalization guarantees.

**Assumption 15.** *The joint support $\mathsf{supp}(X_{\leq T}) = [0,1]^T$. The support of $[\phi_1(X_1, X_2), \phi_2(X_1, X_3)]$ is $\mathbb{R}^{2k}$, where $\phi_{i-j}$ is the embedding function for the labeling function $\rho(\sum_{j \leq i} \phi_{i-j}(x_i, x_j))$.*

**Theorem 11.** *If $\mathcal{H}$ follows Assumption 14, the realizability condition holds, i.e., $f \in \mathcal{H}$, Assumption 15 holds and $T \geq T_{\max}$, then the model trained to minimize the risk in equation 1 (with $T \geq 2$) with $\ell_2$ loss achieves length generalization.*

*Proof.* We start with the same steps as earlier proofs and equate the prediction of $h$ and $f$. We first use the fact $h(x_{\leq i}) = f(x_{\leq i})$ almost everywhere in the support. We can use the continuity of $h, f$ and regular closedness of the support to extend the equality to all points in the support (follows from the first part of Lemma 1) to obtain the following. For all $x_{\leq i} \in \mathsf{supp}(X_{\leq i})$

$$
\omega\Big(\sum_{j<i} \frac{1}{i-1}\psi_{i-j}(x_i, x_j)\Big) = \rho\Big(\sum_{j<i} \frac{1}{i-1}\phi_{i-j}(x_i, x_j)\Big),
$$
$$
\sum_{j<i} \frac{1}{i-1}\psi_{i-j}(x_i, x_j) = \omega^{-1} \circ \rho\Big(\sum_{j<i} \frac{1}{i-1}\phi_{i-j}(x_i, x_j)\Big), \tag{78}
$$
$$
\sum_{j<i} \frac{1}{i-1}\psi_{i-j}(x_i, x_j) = a\Big(\sum_{j<i} \frac{1}{i-1}\phi_{i-j}(x_i, x_j)\Big),
$$

where $a = \omega^{-1} \circ \rho$. In the above simplification, we used the parametric form for the true labeling function and the learned labeling function. We also used the invertibility of $\rho$. Let us consider the setting when $i = 2$. In that case summation involves only one term. Substitute $x_1 = y$ and $x_2 = x$. We obtain $\forall x \in [0, 1]^n, y \in [0, 1]^n$,

$$
\psi_1(x, y) = a(\phi_1(x, y)). \tag{79}
$$

For $i = 3$, substitute $x_1 = x$, $x_3 = z$ and set $x_2 = y$ in such a way that $\phi_1(x, y) = 0$ (follows from Assumption 14). Thus we obtain

$$
\psi_2(x, y) = a(\phi_2(x, y)). \tag{80}
$$

Similarly, we can obtain the following. For all $k \leq T_{\max}$

$$
\psi_k(x, y) = a(\phi_k(x, y)). \tag{81}
$$

The above expression implies that $\psi$ bijectively identifies $\phi$. Let us consider the setting when $i = 3$ (this is possible since $T \geq 3$). We substitute $x_3 = x$, $x_2 = y$, $x_1 = z$ to give

$$
\frac{1}{2}\big(a(\phi_1(x, y)) + a(\phi_2(x, z))\big) = a\big(\frac{1}{2}(\phi_1(x, y) + \phi_2(x, z))\big). \tag{82}
$$

We now use the that assumption $[\phi_1(x, y), \phi_2(x, z)]$ spans $\mathbb{R}^{2k}$ and substitute $\phi_1(x, y) = \alpha$ and $\phi_2(x, z) = \beta$

$$
\frac{1}{2}\big(a(\alpha) + a(\beta)\big) = a\big(\frac{1}{2}(\alpha + \beta)\big). \tag{83}
$$

Rest of the proof follows the same strategy as proof of Theorem 3. □

### C.2.5 EXTENDING THEOREM 3 TO INCORPORATE MULTIPLE ATTENTION HEADS

Our choice of the archictecture did not invoke multiple attention heads. If we include multiple attention heads, then also we can arrive at the same length generalization guarantees. The model class with two attention heads $\psi_1, \psi_2$ can be stated as follows $\omega\Big(\sum_{j<i} A[\psi_1(x_i, x_j), \psi_2(x_i, x_j)]^\top\Big)$, where $A$ combines the outputs of the attention heads linearly. Following the same steps of proof of Theorem 3, we obtain the following.

$$\omega\Big(\sum_{j<i} A[\psi_1(x_i, x_j), \psi_2(x_i, x_j)]^\top\Big) = \rho\Big(\sum_{j<i} B[\phi_1(x_i, x_j), \phi_2(x_i, x_j)]^\top\Big),$$

$$\omega\Big(\sum_{j<i} \tilde{\psi}(x_i, x_j)\Big) = \rho\Big(\sum_{j<i} \tilde{\phi}(x_i, x_j)\Big), \tag{84}$$

$$\sum_{j<i} \tilde{\psi}(x_i, x_j) = a\Big(\sum_{j<i} \tilde{\phi}(x_i, x_j)\Big),$$

where $a = \omega^{-1} \circ \rho$. In the above simplification, the RHS shows the labeling function and the RHS is the function that is learned. We can follow the same strategy as the proof of Theorem 3 for the rest of the proof. We set $i = 2$ and obtain a condition similar to equation 68 and for $i = 3$ we obtain a condition similar to equation 69. Following a similar proof technique, we obtain $a$ is linear and the proof extends to multiple attention heads.

## C.3 STATE SPACE MODELS

In this section, we first provide the proof to Theorem 4. We then provide Corollary 4, where we describe how the learned representations linearly identify the true representations. In Theorem 12, we present the discrete tokens counterpart to Theorem 4.

**Theorem 4.** *If $\mathcal{H}$ follows Assumption 6, and the realizability condition holds, i.e., $f \in \mathcal{H}$, and a further condition on the support, i.e., Assumption 7, holds, then the model trained to minimize the risk in equation 1 with $\ell_2$ loss ($T \geq 2$) achieves length and compositional generalization.*

*Proof.* We start with the same steps as earlier proofs and equate the prediction of $h$ and $f$. We first use the fact $h(x_{\leq i}) = f(x_{\leq i}), \forall i \leq T$ almost everywhere in the support. We can use the continuity of $h, f$ and regular closedness of the support to extend the equality to all points in the support (from first part of Lemma 1) to obtain the following. For all $x_{\leq i} \in \mathsf{supp}(X_{\leq i})$.

$$f(x_{\leq i}) = h(x_{\leq i}) =$$
$$\rho(\sum_{j=0}^{i-1} \Lambda^j B x_{i-j}) = \omega(\sum_{j=0}^{i-1} \tilde{\Lambda}^j \tilde{B} x_{i-j}) \implies$$
$$\omega^{-1} \circ \rho(\sum_{j=0}^{i-1} \Lambda^j B x_{i-j}) = \sum_{j=0}^{i-1} \tilde{\Lambda}^j \tilde{B} x_{i-j} = \tag{85}$$
$$c(\sum_{j=0}^{i-1} \Lambda^j B x_{i-j}) = \sum_{j=0}^{i-1} \tilde{\Lambda}^j \tilde{B} x_{i-j}$$

For $i = 1, \forall x_1 \in \mathbb{R}^n, c(Bx_1) = \tilde{B}x_1$. Substitute $Bx_1 = x$, we obtain $\forall x \in \mathbb{R}^n, c(x) = \tilde{B}B^{-1}x = Cx$, where we use the fact that $Bx_1$ spans $\mathbb{R}^n$ as $B$ is invertible.

From linearity of $c$, we obtain

$$\omega^{-1} \circ \rho(z) = Cz \implies \rho(z) = \omega(Cz), \forall z \in \mathbb{R}^n \tag{86}$$

We use this linearity of $c$ to simplify

$$c(\sum_{j=0}^{i-1} \Lambda^j B x_{i-j}) = \sum_{j=0}^{i-1} \tilde{\Lambda}^j \tilde{B} x_{i-j} \implies$$

$$C(\sum_{j=0}^{i-1} \Lambda^j B x_{i-j}) = \sum_{j=0}^{i-1} \tilde{\Lambda}^j \tilde{B} x_{i-j} \implies$$

$$[CB, C\Lambda B, C\Lambda^2 B, \cdots, C\Lambda^{i-1} B] \begin{bmatrix} x_i \\ x_{i-2} \\ \vdots \\ x_1 \end{bmatrix} - [\tilde{B}, \tilde{\Lambda}\tilde{B}, \tilde{\Lambda}^2 \tilde{B}, \cdots, \tilde{\Lambda}^{i-1}\tilde{B}] \begin{bmatrix} x_i \\ x_{i-2} \\ \vdots \\ x_1 \end{bmatrix} = 0 \implies$$

$$\Big[ [CB, C\Lambda B, C\Lambda^2 B, \cdots, C\Lambda^{i-1} B] - [\tilde{B}, \tilde{\Lambda}\tilde{B}, \tilde{\Lambda}^2 \tilde{B}, \cdots, \tilde{\Lambda}^{i-1}\tilde{B}] \Big] \boldsymbol{X} = 0,$$

$$(87)$$

where $\boldsymbol{X} = \begin{bmatrix} x_i \\ x_{i-2} \\ \vdots \\ x_1 \end{bmatrix}$.

Denote $R = \Big[ [CB, C\Lambda B, C\Lambda^2 B, \cdots, C\Lambda^{i-1} B] - [\tilde{B}, \tilde{\Lambda}\tilde{B}, \tilde{\Lambda}^2 \tilde{B}, \cdots, \tilde{\Lambda}^{i-1}\tilde{B}] \Big]$. We collect a set of points $\boldsymbol{X}^+ = [\boldsymbol{X}^{(1)}, \cdots, \boldsymbol{X}^{(l)}]$ where $l \geq ni$ and rank of $\boldsymbol{X}^+ = ni$ (from Assumption 7). Since the matrix $\boldsymbol{X}^+$ is full rank, we have

$$R\boldsymbol{X}^+ = 0 \implies R = 0.$$

This yields

$$CB = \tilde{B}, C\Lambda B = \tilde{\Lambda}\tilde{B}, \cdots, C\Lambda^i B = \tilde{\Lambda}^i \tilde{B}.$$

$$(88)$$

Observe that from the second equality, we get $\tilde{\Lambda} = C\Lambda C^{-1}$. Given the parameters $(\Lambda, B)$, the set of parameters $(\tilde{\Lambda}, \tilde{B})$ that solve the first two equalities are $-$ {$\tilde{B}$ is an arbitrary invertible matrix, $\tilde{\Lambda} = C\Lambda C^{-1}$, where $C = \tilde{B}B^{-1}$}.

Take any solution of the first two equalities and compute

$$\tilde{\Lambda}^i \tilde{B} = C\Lambda^i C^{-1} \tilde{B} = C\Lambda^i B, \forall i \geq 1$$

$$(89)$$

From equation 89 and equation 86, we obtain that for all $x_{\leq i} \in \mathbb{R}^{ni}$

$$h(x_{\leq i}) = \omega(\sum_{j=0}^{i-1} \tilde{\Lambda}^j \tilde{B} x_{i-j}) = \omega(C \sum_{j=0}^{i-1} \Lambda^j B x_{i-j}) = \rho(\sum_{j=0}^{i-1} \Lambda^j B x_{i-j}) = f(x_{\leq i}) \quad (90)$$

This establishes both compositional and length generalization.

$\square$

**Corollary 4.** *If $\mathcal{H}$ follows Assumption 6, and the realizability condition holds, i.e., $f \in \mathcal{H}$, and a further condition on the support, i.e., Assumption 7, holds, then the model trained to minimize the risk in equation 1 with $\ell_2$ loss ($T \geq 2$) achieves linear identification. Further, under the stated conditions linear identification is necessary for both length and compositional generalization.*

*Proof.* We follow the same steps as proof of Theorem 4 up to equation 89. From that we obtain that for all $x_{\leq i} \in \mathbb{R}^{ni}$

$$\sum_{j=0}^{i-1} \tilde{\Lambda}^j \tilde{B} x_{i-j} = C(\sum_{j=0}^{i-1} \Lambda^j B x_{i-j})$$

$$(91)$$

Recall that $\sum_{j=0}^{i-1} \tilde{\Lambda}^j \tilde{B} x_{i-j} = \tilde{h}_j$ and $\sum_{j=0}^{i-1} \Lambda^j B x_{i-j} = h_j$. From this it follows that $\tilde{h}_j = C h_j$, which proves that learned hidden state are a linear transform of the hidden state underlying the labeling function. This establishes linear identification.

To show the necessity of linear identification, from the proof of Theorem 4 it follows that

$$
\forall i \leq T, \forall x_{\leq i} \in \mathsf{supp}(X_{\leq i}), \rho(\sum_{j=0}^{i-1} \Lambda^j B x_{i-j}) = \omega(\sum_{j=0}^{i-1} \tilde{\Lambda}^j \tilde{B} x_{i-j}) \implies
$$
$$
\forall i \geq 1, \forall x_{\leq i} \in \mathbb{R}^{ni}, \tilde{h}_j = C h_j \tag{92}
$$

If the latter conditon in the above implication does not hold, then the former condition cannot hold. Hence, linear identification is necessary. $\qquad\square$

### C.3.1 EXTENDING THEOREM 4 TO DISCRETE TOKENS

In our discussion, we have focused on settings where the support of each token has a non-empty interior (Assumption 2). In practice of language modeling, we use discrete tokens and hence Assumption 2 does not hold anymore. In this section, we discuss the adaptation of results for SSMs to setting when the the support of tokens is a finite set.

**Assumption 16.** *Each function in the hypothesis class $\mathcal{H}$ takes a sequence $\{x_1, \cdots, x_i\}$ as input and outputs $h(x_1, \cdots, x_i) = \omega\left(\sum_{j=0}^{i-1} \Lambda^j B x_{i-j}\right)$, where $\omega : \mathbb{R}^k \to \mathbb{R}^m$ is a single layer perceptron denoted as $\sigma \circ A$. $A$, $B$ and $\Lambda$ are square invertible. As a result, $k = m = n$.*

**Assumption 17.** *For some length $2 \leq i \leq T$ an there exists in sequences $x_{\leq i}$ such that their concatenation forms a $in \times in$ matrix of rank $in$.*

**Theorem 12.** *If $\mathcal{H}$ follows Assumption 16, and the realizability condition holds, i.e., $f \in \mathcal{H}$, and a further condition on the support, i.e., Assumption 17, holds, then the model trained to minimize the risk in equation 1 with $\ell_2$ loss $(T \geq 2)$ achieves length and compositional generalization.*

*Proof.* We start with the same steps as earlier proofs and equate the prediction of $h$ and $f$. We first use the fact $h(x_{\leq i}) = f(x_{\leq i}), \forall i \leq T$ almost everywhere in the support. We can use the continuity of $h, f$ and regular closedness of the support to extend the equality to all points in the support (from first part of Lemma 1) to obtain the following. For all $x_{\leq i} \in \mathsf{supp}(X_{\leq i})$.

$$
f(x_{\leq i}) = h(x_{\leq i}) =
$$
$$
\rho(\sum_{j=0}^{i-1} \Lambda^j B x_{i-j}) = \omega(\sum_{j=0}^{i-1} \tilde{\Lambda}^j \tilde{B} x_{i-j}) \implies
$$
$$
\sigma(A \sum_{j=0}^{i-1} \Lambda^j B x_{i-j}) = \sigma(\tilde{A} \sum_{j=0}^{i-1} \tilde{\Lambda}^j \tilde{B} x_{i-j}) = \tag{93}
$$
$$
C(\sum_{j=0}^{i-1} \Lambda^j B x_{i-j}) = \sum_{j=0}^{i-1} \tilde{\Lambda}^j \tilde{B} x_{i-j},
$$

where $C = \tilde{A}^{-1} A$.

We simplify the last identity in the above further.

$$C(\sum_{j=0}^{i-1} \Lambda^j B x_{i-j}) = \sum_{j=0}^{i-1} \tilde{\Lambda}^j \tilde{B} x_{i-j} \implies$$

$$[CB, C\Lambda B, C\Lambda^2 B, \cdots, C\Lambda^{i-1} B] \begin{bmatrix} x_i \\ x_{i-2} \\ \vdots \\ x_1 \end{bmatrix} - [\tilde{B}, \tilde{\Lambda}\tilde{B}, \tilde{\Lambda}^2\tilde{B}, \cdots, \tilde{\Lambda}^{i-1}\tilde{B}] \begin{bmatrix} x_i \\ x_{i-2} \\ \vdots \\ x_1 \end{bmatrix} = 0 \implies$$

$$\Big[ [CB, C\Lambda B, C\Lambda^2 B, \cdots, C\Lambda^{i-1} B] - [\tilde{B}, \tilde{\Lambda}\tilde{B}, \tilde{\Lambda}^2\tilde{B}, \cdots, \tilde{\Lambda}^{i-1}\tilde{B}] \Big] \boldsymbol{X} = 0,$$

$$(94)$$

where $\boldsymbol{X} = \begin{bmatrix} x_i \\ x_{i-2} \\ \vdots \\ x_1 \end{bmatrix}$.

Denote $R = \Big[ [CB, C\Lambda B, C\Lambda^2 B, \cdots, C\Lambda^{i-1} B] - [\tilde{B}, \tilde{\Lambda}\tilde{B}, \tilde{\Lambda}^2\tilde{B}, \cdots, \tilde{\Lambda}^{i-1}\tilde{B}] \Big]$. We collect a set of points $\boldsymbol{X}^+ = [\boldsymbol{X}^{(1)}, \cdots, \boldsymbol{X}^{(l)}]$ where $l \geq ni$ and rank of $\boldsymbol{X}^+ = ni$ (from Assumption 7). Since the matrix $\boldsymbol{X}^+$ is full rank, we have

$$R\boldsymbol{X}^+ = 0 \implies R = 0.$$

This yields

$$CB = \tilde{B}, C\Lambda B = \tilde{\Lambda}\tilde{B}, \cdots, C\Lambda^i B = \tilde{\Lambda}^i \tilde{B}. \tag{95}$$

Observe that from the second equality, we get $\tilde{\Lambda} = C\Lambda C^{-1}$. Given the parameters $(\Lambda, B)$, the set of parameters $(\tilde{\Lambda}, \tilde{B})$ that solve the first two equalities are – $\{\tilde{B}$ is an arbitrary invertible matrix, $\tilde{\Lambda} = C\Lambda C^{-1}$, where $C = \tilde{B}B^{-1}\}$.

Take any solution of the first two equalities and compute

$$\tilde{\Lambda}^i \tilde{B} = C\Lambda^i C^{-1} \tilde{B} = C\Lambda^i B, \forall i \geq 1 \tag{96}$$

From equation 96, we obtain that for all $x_{\leq i} \in \mathbb{R}^{ni}$

$$h(x_{\leq i}) = \omega(\sum_{j=0}^{i-1} \tilde{\Lambda}^j \tilde{B} x_{i-j}) = \omega(C\sum_{j=0}^{i-1} \Lambda^j B x_{i-j}) = \rho(\sum_{j=0}^{i-1} \Lambda^j B x_{i-j}) = f(x_{\leq i}) \tag{97}$$

This establishes both compositional and length generalization.

$$\square$$

### C.4 VANILLA RNNs

In this section, we discuss RNNs and present the proof of Theorem 5. We first build some lemmas in the form of Lemma 2 and 3 that are used to prove Theorem 5. In Corollary 5, we explain the learned hidden state are a permutation transform of the true hidden state and also show that its a necessary condition for length and compositional generalization. Finally, in Theorem 13, we present the discrete token counterpart to Theorem 5.

**Lemma 2.** *The $k^{th}$ derivative of sigmoid function denoted $\frac{\partial^k \sigma(s)}{\partial s^k}$ is not zero identically.*

*Proof.* The first derivative of the sigmoid function $\frac{\partial \sigma(s)}{\partial s} = \sigma(s)(1-\sigma(s))$. We argue that the $\frac{\partial^k \sigma(s)}{\partial s^k}$ is a polynomial in $\sigma(s)$ with degree $k+1$. Consider the base case of $k = 1$. This condition is true

as $\frac{\partial \sigma(s)}{\partial s} = \sigma(s)(1 - \sigma(s))$. Now let us assume that $\frac{\partial^k \sigma(s)}{\partial s^k}$ is a polynomial of degree at most $k + 1$ denoted as $P_{k+1}(\sigma(s))$. We simplify

$$\frac{\partial^k \sigma(s)}{\partial s^k} = P_{k+1}(\sigma(s)) = \sum_{j=1}^{k+1} a_j (\sigma(s))^j$$

We take another derivative of the term above as follows.

$$\frac{\partial^{k+1} \sigma(s)}{\partial s^{k+1}} = \frac{\partial P_{k+1}(\sigma(s))}{\partial s} = \sum_{j=1}^{k+1} a_j \frac{\partial (\sigma(s))^j}{\partial s} = \sum_{j=1}^{k+1} a_j j \sigma(s)^{j-1} (\sigma(s)(1 - \sigma(s)))$$

Observe that the $\frac{\partial^{k+1} \sigma(s)}{\partial s^{k+1}}$ is also a polynomial in $\sigma(s)$. Observe that the degree $k + 2$ term has one term with coefficient $-a_{k+1} \cdot (k+1)$. Since $a_{k+1} \neq 0$, the coefficient of degree $k+2$, $-a_{k+1} \cdot (k+1)$, is also non-zero. Since $\frac{\partial^k \sigma(s)}{\partial s^k}$ is a polynomial in $\sigma(s)$ with degree $k + 1$ and hence, it cannot be zero identically.

$\square$

**Lemma 3.** *Let $x \in \mathbb{R}^n$ and $A \in \mathbb{R}^{n \times n}$. Suppose $Ax = 0, \forall x \in \mathcal{X}$, where $\mathcal{X}$ has a non-empty interior. Under these conditions $A = 0$.*

*Proof.* Since $\mathcal{X}$ has a non-empty interior, we can construct a $\ell_\infty$ ball centered on $\theta$, defined as follows $- \tilde{\mathcal{X}} = \{\theta + \sum_{j=1}^n \alpha_j e_j \ |\|\alpha\|_\infty \leq \alpha_{\max} \}$, where $e_j$ is a vector that is zero in all components and one on the $j^{th}$ component. Suppose $A$ was non-zero. One of the columns say $a_j$ is non-zero. Consider two points in the ball $\tilde{\mathcal{X}}$ such that $j^{th}$ coefficients are non-zero but rest of the coefficients are zero. We denote the $j^{th}$ components for the two components as $\alpha_j$ and $\tilde{\alpha}_j$, where $\alpha_j \neq \tilde{\alpha}_j$. We now plug these two points into the condition that $Ax = 0$

$$\begin{aligned} A(\theta + \alpha_j e_j) = 0 &\implies A\theta = \alpha_j a_j, \\ A(\theta + \tilde{\alpha}_j e_j) = 0 &\implies A\theta = \tilde{\alpha}_j a_j, \end{aligned} \tag{98}$$

We take a difference of the two steps above and obtain

$$(\alpha_j - \tilde{\alpha}_j) a_j = 0 \implies a_j = 0$$

This is a contradiction. Hence, $A = 0$. $\square$

**Theorem 5.** *If $\mathcal{H}$ follows Assumption 8, and the realizability condition holds, i.e., $f \in \mathcal{H}$ and regular closedness condition in Assumption 2 holds, then the model trained to minimize the risk in equation 1 with $\ell_2$ loss (with $T \geq 2$) achieves length and compositional generalization.*

*Proof.* We start with the same steps as earlier proofs and equate the prediction of $h$ and $f$ everywhere in the support of the training distribution (using first part of Lemma 1). We start with equating label at length 1, i.e., $y_1$. For all $x_1 \in \text{supp}(X_1)$

$$\begin{aligned} \sigma(A\sigma(Bx_1)) = \sigma(\tilde{A}\sigma(\tilde{B}x_1)) &\implies A\sigma(Bx_1) = \tilde{A}\sigma(\tilde{B}x_1) \implies \\ \sigma(B\tilde{B}^{-1}\tilde{B}x_1) &= A^{-1}\tilde{A}\sigma(\tilde{B}x_1) \end{aligned} \tag{99}$$

Say $y = \tilde{B}x_1$, $A^{-1}\tilde{A} = U$, $B\tilde{B}^{-1} = V$. We substitute these expressions in the simplificaction below. We pick a $y$ in the interior of $\tilde{B} \cdot \text{supp}(X_1)$.

$$\sigma(Vy) = U\sigma(y) \tag{100}$$

Take the first row of $V$ and $U$ as $v^\top$ and $u^\top$ to obtain

$$\sigma(v^\top y) = u^\top \sigma(y) \tag{101}$$

Suppose there is some non-zero component of $v$ say $i$ but the corresponding component is zero in $u$.

$$\frac{\partial \sigma(v_i y_i + v_{-i} y_{-i})}{\partial y_i} = \sigma^{'}(v_i y_i + v_{-i} y_{-i}) v_i = \frac{\partial u_{-i}^{\top} \sigma(y_{-i})}{\partial y_i} = 0 \tag{102}$$

From the above we get $\sigma^{'}(v^{\top} y) = 0$. But sigmoid is strictly monotonic on $\mathbb{R}$, $\sigma^{'}(x) > 0, \forall x \in \mathbb{R}$ and $v^{\top} y \in \mathbb{R}$. Hence, $\sigma^{'}(v^{\top} y) = 0$ is not possible. Similarly, suppose some component is non-zero in $u$ and zero in $v$.

$$\frac{\partial \sigma(v_{-i}^{\top} y_{-i})}{\partial y_i} = 0 = \frac{\partial (u_i \sigma(y_i) + u_{-i}^{\top} \sigma(y_{-i}))}{\partial y_i} = u_i \sigma^{'}(y_i) \tag{103}$$

Since the derivative of $\sigma$ cannot be zero, the above condition cannot be true.

From the above, we can deduce that both $u$ and $v$ have same non-zero components.

Let us start with the case where $p \geq 2$ components of $u, v$ are non-zero. Below we equate the partial derivative w.r.t all components of $y$ that have non-zero component in $u$ (since $y$ is in the interior of the image of $\tilde{B} x_1$, we can equate these derivatives).

$$\sigma(v^{\top} y) = u^{\top} \sigma(y),$$
$$\frac{\partial^p \sigma(s)}{\partial s^p}\bigg|_{s=v^{\top} y} \left( \Pi_{v_i \neq 0} v_i \right) = 0 \implies \frac{\partial^p \sigma(s)}{\partial s^p} = 0. \tag{104}$$

Since support $X_1$ has a non-empty interior, the set of values $v^{\top} y$ takes also has a non-empty interior in $\mathbb{R}$. Hence, the above equality is true over a set of values $s$, which have a non-empty interior. Since $\sigma(s)$ is analytic, $\frac{\partial^p \sigma(s)}{\partial s^p}$ is also analytic. From (Mityagin, 2015), it follows that $\frac{\partial^p \sigma(s)}{\partial s^p} = 0$ everywhere. From Lemma 2, we know this condition cannot be true.

We are left with the case where $u$ and $v$ have one non-zero component each.

$$\frac{1}{1 + e^{-vy}} = \frac{u}{1 + e^{-y}} \implies 1 + e^{-y} = u + u e^{-vy}$$

In the simplification above, we take derivative w.r.t $y$ to obtain $e^{-(v-1)y} = 1/uv$. We now again take derivative again w.r.t $y$ to get $v = 1$ and substitute it back to get $u = 1$. Note that no other row of $U$ or $V$ can have same non-zero element because that would make matrix non invertible. From this we deduce that $U$ and $V$ are permutation matrices. From $\sigma(Vy) = U\sigma(y)$ it follows that $U = V = \Pi$. Thus $B = \Pi \tilde{B}$ and $\tilde{A} = A\Pi$.

Next, we equate predictions for $y_2$ to the ground truth (label $y_2$ exists as $T \geq 2$). For all $x_1 \in \text{supp}(X_1)$

$$\sigma(A\sigma(\Lambda \sigma(Bx_1) + Bx_2)) = \sigma(\tilde{A}\sigma(\tilde{\Lambda}\sigma(\tilde{B}x_1) + \tilde{B}x_2)) \implies$$
$$A\sigma(\Lambda \sigma(Bx_1) + Bx_2) = \tilde{A}\sigma(\tilde{\Lambda}\sigma(\tilde{B}x_1) + \tilde{B}x_2) \implies \tag{105}$$
$$\tilde{A}\sigma(\tilde{\Lambda}\sigma(\tilde{B}x_1) + \tilde{B}x_2) = A\Pi\sigma(\tilde{\Lambda}\Pi^{\top}\sigma(Bx_1) + \Pi^{\top}Bx_2) = A\sigma(\Pi\tilde{\Lambda}\Pi^{\top}\sigma(Bx_1) + Bx_2).$$

We use the simplification in the second step to equate to LHS in the first step as follows.

$$A\sigma(\Pi\tilde{\Lambda}\Pi^{\top}\sigma(Bx_1) + Bx_2) = A\sigma(\Lambda \sigma(Bx_1) + Bx_2)$$
$$\implies (\Pi\tilde{\Lambda}\Pi^{\top} - \Lambda)\sigma(Bx_1) = 0. \tag{106}$$

Since $\sigma(Bx_1)$ spans a set that has a non-empty interior, we get that $\tilde{\Lambda} = \Pi^{\top}\Lambda\Pi$ (from Lemma 3).

From the above conditions, we have arrived at $\tilde{\Lambda} = \Pi^{\top}\Lambda\Pi, \tilde{B} = \Pi^{\top}B, \tilde{A} = A\Pi$.

We want to show that for all $k \geq 1$

$$h_k = \Pi \tilde{h}_k, \tag{107}$$

where $h_k = \sigma(\Lambda h_{k-1} + B x_k)$ and $\tilde{h}_k = \sigma(\tilde{\Lambda} \tilde{h}_{k-1} + \tilde{B} x_k)$ and $h_0 = \tilde{h}_0 = 0$. In other words, we define $T_k$ as a mapping that takes $x_{\leq k}$ as input and outputs $h_k$, i.e., $T_k(x_{\leq k}) = h_k$. Similarly, we write $\tilde{T}_k(x_{\leq k}) = \tilde{h}_k$. We want to show

$$T_k = \Pi \tilde{T}_k, \forall k \tag{108}$$

We show the above by principle of induction. Let us consider the base case below. For all $x_1 \in \mathbb{R}^n$

$$\tilde{A} \sigma(\tilde{B} x_1) = A \Pi \sigma(\Pi^\top B x_1) = A \sigma(B x_1) = A h_1 \implies h_1 = \Pi \tilde{h}_1 \implies T_1(x_1) = \Pi \tilde{T}_1(x_1) \tag{109}$$

Suppose $\forall j \leq k, T_j = \Pi \tilde{T}_j$.

Having shown the base case and assumed the condition for $j \leq k$, we now consider the mapping $\tilde{T}_{k+1}$

$$\Pi \tilde{T}_{k+1}(x_{\leq k+1}) = \Pi \sigma(\tilde{\Lambda} \tilde{h}_k + \tilde{B} x_{k+1}) = \Pi \sigma(\Pi^\top \Lambda \Pi \tilde{h}_k + \Pi^\top B x_k) = \sigma(\Lambda h_k + B x_k) = T_{k+1}(x_{\leq k+1}). \tag{110}$$

The prediction from the model $(\tilde{A}, \tilde{\Lambda}, \tilde{B})$ at a time step $k$ is denoted as $\tilde{y}_k$ and it relates to $\tilde{h}_k$ as follows $\tilde{y}_k = \sigma(\tilde{A} \tilde{h}_k)$. We use the above condition in equation equation 126 to arrive at the following result. For all $x_{\leq k} \in \mathbb{R}^{nk}$

$$\tilde{y}_k = \sigma(\tilde{A} \tilde{h}_k) = \sigma(\tilde{A} \tilde{T}(x_{\leq k})) = \sigma(A \Pi \tilde{T}(x_{\leq k})) = \sigma(A T(x_{\leq k})) = y_k$$

This completes the proof. $\qquad \square$

**Corollary 5.** *If $\mathcal{H}$ follows Assumption 8, and the realizability condition holds, i.e., $f \in \mathcal{H}$ and regular closedness condition in Assumption 2 holds, then the model trained to minimize the risk in equation 1 with $\ell_2$ loss (with $T \geq 2$) achieves permutation identification. Further, under the stated conditions permutation identification is necessary for both length and compositional generalization.*

*Proof.* We follow the exact steps from the proof of Theorem 5 up to equation 128. From equation 128 it follows that for all $x_{\leq k} \in \mathbb{R}^{nk}$

$$T_k(x_{\leq k}) = \Pi \tilde{T}(x_{\leq k}) \implies h_k = \Pi \tilde{h}_k \tag{111}$$

The above implies permutation identification. To show the necessity of permutation identification, from the proof of Theorem 5 observe that

$$\forall i \leq T, \forall x_{\leq i} \in \mathsf{supp}(X_{\leq i}), \; \sigma(A \sigma(B x_i + \Lambda h_{i-1})) = \sigma(\tilde{A} \sigma(\tilde{B} x_i + \Lambda \tilde{h}_{i-1})) \implies T_k(x_{\leq k}) = \Pi \tilde{T}(x_{\leq k}) \tag{112}$$

The latter condition implies permutation identification. If it does not hold, then the condition in LHS cannot hold and hence neither length nor compositional generalization can be achieved. $\qquad \square$

### C.4.1 EXTENDING THEOREM 5 TO DISCRETE TOKENS

In our discussion, we have focused on settings where the support of each token has a non-empty interior (Assumption 2). In practice of language modeling, we use discrete tokens and hence Assumption 2 does not hold anymore. In this section, we discuss the adaptation of results for vanilla RNNs to setting when the the support of tokens is a finite set.

Define $\mathcal{S} = \{y = B x \mid x \in \mathcal{X}\}$, where $\mathcal{X}$ is the marginal support of each token.

**Assumption 18.** *a) For each component $i$ of $y$, $\mathcal{S}$ contains two pairs where the first coordinate differs by the same amount. Mathematically stated, the two pairs are $\Big((y_i, y_{-i}), (y_i + \delta, y_{-i}))\Big)$ and $\Big((y_i^{'}, y_{-i}^{'}), (y_i^{'} + \delta, y_{-i}^{'}))\Big)$.*

*b) For every pair of components $i, j$ of $y$, $\mathcal{S}$ contains a point $y$ that satisfies the following. There exists three points in $\mathcal{S}$ such that they only differ in $y_i, y_j$, and form a rectangle, $(y_i, y_j), (y_i^{'}, y_j)$, $(y_i, y_j^{'}), (y_i^{'}, y_j^{'})$. Similarly, there exists another set of points where $y_i^{'} < y_i$ and $y_j^{'} < y_j$.*

**Theorem 13.** *If $\mathcal{H}$ follows Assumption 8, and the realizability condition holds, i.e., $f \in \mathcal{H}$ and regular closedness condition in Assumption 2 holds, then the model trained to minimize the risk in equation 1 with $\ell_2$ loss (with $T \geq 2$) achieves length and compositional generalization.*

*Proof.* We start with the same steps as earlier proofs and equate the prediction of $h$ and $f$ everywhere in the support of the training distribution. We start with equating label at length 1, i.e., $y_1$. For all $x_1 \in \mathsf{supp}(X_1)$

$$\sigma(A\sigma(Bx_1)) = \sigma(\tilde{A}\sigma(\tilde{B}x_1)) \implies A\sigma(Bx_1) = \tilde{A}\sigma(\tilde{B}x_1) \implies$$
$$\tilde{A}^{-1}A\sigma(Bx_1) = \sigma(\tilde{B}B^{-1}Bx_1) \tag{113}$$

Say $y = Bx_1$, $\tilde{A}^{-1}A = U$, $\tilde{B}B^{-1} = V$. We substitute these expressions in the simplificaction below. We pick a $y$ in the interior of $\tilde{B} \cdot \mathsf{supp}(X_1)$.

$$\sigma(Vy) = U\sigma(y) \tag{114}$$

Take the first row of $V$ and $U$ as $v^{\top}$ and $u^{\top}$ to obtain

$$\sigma(v^{\top}y) = u^{\top}\sigma(y) \tag{115}$$

Say $v_i \neq 0$ and $u_i = 0$. We consider a $(y_i, y_{-i})$ and $(y_i^{'}, y_{-i})$ satisfying Assumption 18 a. We substitute these points in equation 115 and take the difference of the LHS and RHS in equation 115 to obtain.

$$\sigma(v_i y_i^{'} + v_{-i}y_{-i}) - \sigma(v_i y_i + v_{-i}y_{-i}) = 0 \tag{116}$$

$\sigma$ is strictly monotonic and thus the above cannot be true. Similarly, we can rule out the case when $u_i \neq 0$ and $v_i = 0$. Thus we can deduce that both $u$ and $v$ have same non-zero components.

Let us start with the case where $p \geq 2$ components of $u, v$ are non-zero. Without loss of generality say the first two components are among coordinates that are non-zero. Pick a $y \in \mathcal{S}$ that satisfies Assumption 18 b. Suppose $v^{\top}y \geq 0$. We select the neighbors of $y$ that form the rectangle such that each coordinate is greater than $y$. We substitute these points in equation 115 and the simplification procedure works as follows. Let

$$s_1 = v_1 y_1^{'} + v_2 y_2^{'} + \cdots v_n y_n, \quad s_3 = v_1 y_1^{'} + v_2 y_2 + \cdots v_n y_n$$
$$s_2 = v_1 y_1 + v_2 y_2^{'} + \cdots v_n y_n, \quad s_4 = v_1 y_1 + v_2 y_2 + \cdots v_n y_n \tag{117}$$

Observe that $s_1 > s_2 > s_4$ and $s_1 > s_3 > s_4$. It is possible that $s_2 \geq s_3$ or $s_3 > s_2$. Suppose $s_2 \geq s_3$.

We can write

$$\sigma(s_1) = u_1\sigma(y_1^{'}) + u_2\sigma(y_2^{'}) + \cdots + u_n\sigma(y_n), \sigma(s_2) = u_1\sigma(y_1) + u_2\sigma(y_2^{'}) + \cdots + u_n\sigma(y_n)$$
$$\sigma(s_3) = u_1\sigma(y_1^{'}) + u_2\sigma(y_2) + \cdots + u_n\sigma(y_n), \sigma(s_4) = u_1\sigma(y_1) + u_2\sigma(y_2) + \cdots + u_n\sigma(y_n) \tag{118}$$

We take a difference of the first two and the latter two, and subtract these differences to get

$$\Big(\sigma(s_1) - \sigma(s_2)\Big) - \Big(\sigma(s_3) - \sigma(s_4)\Big) = 0 \tag{119}$$

From mean value theorem, we get that $\sigma^{'}(\tilde{s}) = \sigma^{'}(s^{\dagger})$, where $\sigma^{'}$ is the derivative of $\sigma$, $\tilde{s}$ is a value between $s_1$ and $s_2$, and $s^{\dagger}$ is a value between $s_3$ and $s_4$. Since $s_1 > s_2 > s_3 > s_4 > 0$, $\tilde{s} > s^{\dagger} > 0$. Since $\sigma^{'}$ strictly decreases on positive values, the above equality $\sigma^{'}(\tilde{s}) = \sigma^{'}(s^{\dagger})$ is not possible. Similarly, we can tackle the case $v^{\top}y < 0$.

We are left with the case where $u$ and $v$ have one non-zero component each. From Assumption 18a, we select two pairs that differe exactly in the non-zero component. We can resort to dealing with scalars as follows. We start with first pair $(y, y + \delta)$.

$$\sigma(vy) = u\sigma(y) \implies \frac{1}{1 + e^{-vy}} = \frac{u}{1 + e^{-y}} \implies 1 - u = ue^{-vy} - e^{-y}$$
$$\sigma(v(y + \delta)) = u\sigma(y + \delta) \implies 1 - u = ue^{-v(y+\delta)} - e^{-(y+\delta)} \tag{120}$$

By equating the RHS in the above, we obtain

$$\frac{1 - e^{-\delta}}{1 - e^{-v\delta}} = ue^{-(v-1)y} \tag{121}$$

For the second pair $(y^{'}, y^{'} + \delta)$, we obtain

$$\frac{1 - e^{-\delta}}{1 - e^{-v\delta}} = ue^{-(v-1)y^{'}} \tag{122}$$

If we compare the RHS of equation 121 and equation 122, we obtain $ue^{-(v-1)y} = ue^{-(v-1)y^{'}}$. Since $u$ is non-zero, we obtain that $v = 1$. Substituting this into $\sigma(vy) = u\sigma(y)$, we also obtain $u = 1$.

Note that no other row of $U$ or $V$ can have same non-zero element because that would make matrix non invertible. From this we deduce that $U$ and $V$ are permutation matrices. From $\sigma(Vy) = U\sigma(y)$ it follows that $U = V = \Pi$. Thus $B = \Pi\tilde{B}$ and $\tilde{A} = A\Pi$.

Next, we equate predictions for $y_2$ to the ground truth (label $y_2$ exists as $T \geq 2$). For all $x_1 \in \text{supp}(X_1)$

$$\sigma(A\sigma(\Lambda\sigma(Bx_1) + Bx_2)) = \sigma(\tilde{A}\sigma(\tilde{\Lambda}\sigma(\tilde{B}x_1) + \tilde{B}x_2)) \implies$$
$$A\sigma(\Lambda\sigma(Bx_1) + Bx_2) = \tilde{A}\sigma(\tilde{\Lambda}\sigma(\tilde{B}x_1) + \tilde{B}x_2) \implies \tag{123}$$
$$\tilde{A}\sigma(\tilde{\Lambda}\sigma(\tilde{B}x_1) + \tilde{B}x_2) = A\Pi\sigma(\tilde{\Lambda}\Pi^{\top}\sigma(Bx_1) + \Pi^{\top}Bx_2) = A\sigma(\Pi\tilde{\Lambda}\Pi^{\top}\sigma(Bx_1) + Bx_2).$$

We use the simplification in the second step to equate to LHS in the first step as follows.

$$A\sigma(\Pi\tilde{\Lambda}\Pi^{\top}\sigma(Bx_1) + Bx_2) = A\sigma(\Lambda\sigma(Bx_1) + Bx_2)$$
$$\implies (\Pi\tilde{\Lambda}\Pi^{\top} - \Lambda)\sigma(Bx_1) = 0. \tag{124}$$

Since $\sigma(Bx_1)$ spans a set that has a non-empty interior, we get that $\tilde{\Lambda} = \Pi^{\top}\Lambda\Pi$ (from Lemma 3).

From the above conditions, we have arrived at $\tilde{\Lambda} = \Pi^{\top}\Lambda\Pi, \tilde{B} = \Pi^{\top}B, \tilde{A} = A\Pi$.

We want to show that for all $k \geq 1$

$$h_k = \Pi\tilde{h}_k, \tag{125}$$

where $h_k = \sigma(\Lambda h_{k-1} + Bx_k)$ and $\tilde{h}_k = \sigma(\tilde{\Lambda}\tilde{h}_{k-1} + \tilde{B}x_k)$ and $h_0 = \tilde{h}_0 = 0$. In other words, we define $T_k$ as a mapping that takes $x_{\leq k}$ as input and outputs $h_k$, i.e., $T_k(x_{\leq k}) = h_k$. Similarly, we write $\tilde{T}_k(x_{\leq k}) = \tilde{h}_k$. We want to show

$$T_k = \Pi\tilde{T}_k, \forall k \tag{126}$$

We show the above by principle of induction. Let us consider the base case below. For all $x_1 \in \mathbb{R}^n$

$$\tilde{A}\sigma(\tilde{B}x_1) = A\Pi\sigma(\Pi^\top Bx_1) = A\sigma(Bx_1) = Ah_1 \implies h_1 = \Pi\tilde{h}_1 \implies T_1(x_1) = \Pi\tilde{T}_1(x_1) \tag{127}$$

Suppose $\forall j \leq k, T_j = \Pi\tilde{T}_j$.

Having shown the base case and assumed the condition for $j \leq k$, we now consider the mapping $\tilde{T}_{k+1}$

$$\Pi\tilde{T}_{k+1}(x_{\leq k+1}) = \Pi\sigma(\tilde{\Lambda}\tilde{h}_k + \tilde{B}x_{k+1}) = \Pi\sigma(\Pi^\top\Lambda\Pi\tilde{h}_k + \Pi^\top Bx_k) = \sigma(\Lambda h_k + Bx_k) = T_{k+1}(x_{\leq k+1}). \tag{128}$$

The prediction from the model $(\tilde{A}, \tilde{\Lambda}, \tilde{B})$ at a time step $k$ is denoted as $\tilde{y}_k$ and it relates to $\tilde{h}_k$ as follows $\tilde{y}_k = \sigma(\tilde{A}\tilde{h}_k)$. We use the above condition in equation equation 126 to arrive at the following result. For all $x_{\leq k} \in \mathcal{X}^k$

$$\tilde{y}_k = \sigma(\tilde{A}\tilde{h}_k) = \sigma(\tilde{A}\tilde{T}(x_{\leq k})) = \sigma(A\Pi\tilde{T}(x_{\leq k})) = \sigma(AT(x_{\leq k})) = y_k$$

This completes the proof. $\qquad\square$

## C.5 FINITE HYPOTHESIS CLASS

Before we present the proof of Theorem 6, we revisit some basics of convergence of sets. Consider a sequence of sets $(A_n)$ which are a subset of $\Omega$, i.e., $A_n \subseteq \Omega$. We define the lim inf first and then the lim sup.

$$\liminf_{n\to\infty} A_n = \bigcup_{n\geq 1}\bigcap_{j\geq n} A_j$$

$$\limsup_{n\to\infty} A_n = \bigcap_{n\geq 1}\bigcup_{j\geq n} A_j$$

The limit of this sequence of sets exists provided the lim inf and lim sup are equal, i.e.,

$$\lim_{n\to\infty} A_n = \bigcup_{n\geq 1}\bigcap_{j\geq n} A_j = \bigcap_{n\geq 1}\bigcup_{j\geq n} A_j$$

If the sequence comprises of non-increasing sets, i.e., $A_{n+1} \subseteq A_n$, then the limit exists. For this non-increasing sequence observe that

$$\bigcap_{j\geq n} A_j = \bigcap_{j\geq 1} A_j$$

$$\bigcup_{j\geq n} A_j = A_n$$

We combine the above two observations to see both lim inf and lim sup are equal and thus the limit of non-increasing sets exists. There is another way to define the limit of sets using indicator functions that goes as follows. $\mathbf{1}_A(\cdot)$ is the indicator function that checks if input belongs to the set

or not and takes the value of one if the input is in the set and zero otherwise. We define the limit using indicator functions as follows.

$$\lim_{n\to\infty} A_n = \{\omega \in \Omega, \lim_{n\to\infty} \mathbf{1}_{A_n}(\omega) = 1\},$$

where $\mathbf{1}_{A_n}(\omega)$ is one if $\omega \in A_n$ and zero otherwise. The limit of sequence of sets $A_n$ exists if and only if $\lim_{n\to\infty} \mathbf{1}_{A_n}(\omega)$ exists for all $\omega \in \Omega$.

**Theorem 6.** *If $\mathcal{H}$ is a finite hypothesis class, the realizability condition holds, i.e., $f \in \mathcal{H}$, then $\exists\, T_0 < \infty$ such that the model trained to minimize the risk in equation 1 with $\ell_2$ loss and $T > T_0$ achieves length generalization.*

*Proof.* Let $\mathcal{H}_T$ be the set of solutions to equation 1, where $T$ is the maximum length of the sequence in the training distribution. Observe that each $\mathcal{H}_T$ can take one of the possible values in the power set $2^{\mathcal{H}}$, i.e., the set of all the possible subsets of $\mathcal{H}$. Since the objective at length $T$ in equation 1 evaluates the model at all lengths up to length $T$ we obtain that $\mathcal{H}_{T+1} \subseteq \mathcal{H}_T$. Since the sequence $\mathcal{H}_T$ indexed by $T$ is non-increasing, the limit of the above sequence exists and is denoted as $\mathcal{H}^\star$. From the indicator function definition of the limit, we can write $\mathcal{H}^\star$ as

$$\mathcal{H}^\star = \{h \in \mathcal{H}, \lim_{T\to\infty} \mathbf{1}_{\mathcal{H}_T}(h) = 1\}$$

Since the limit of the sequence $\mathcal{H}_T$ exists, for each $h \in \mathcal{H}$, the limit $\lim_{T\to\infty} \mathbf{1}_{\mathcal{H}_T}(h)$ exists denoted as $p(h)$. Each element of this sequence $\mathbf{1}_{\mathcal{H}_T}(h)$ indexed by $T$ takes a value of one or zero. From the standard definition of limit, we know that for each $\epsilon$, there exists $T(h, \epsilon)$ such that $T > T(h, \epsilon)$, $|\mathbf{1}_{\mathcal{H}_T}(h) - p(h)| < \epsilon$. Both $\mathbf{1}_{\mathcal{H}_T}(h)$ and $p(h)$ can only take a value of 0 or 1 (for $p(h)$ if there is any other value it takes, then the distance of sequence terms $\mathbf{1}_{\mathcal{H}_T}(h)$ from $p(h)$ will be bounded away from zero, which is not possible). If $\epsilon < 1$, then for all $T > T(h, \epsilon)$, $\mathbf{1}_{\mathcal{H}_T}(h) = p(h)$.

Define $T_0 = \sup_{h\in\mathcal{H}} T(h, \epsilon)$. Since $\mathcal{H}$ is finite, $T_0 < \infty$.

We can write the set $\mathcal{H}_T$ as

$$\mathcal{H}_T = \{h \in \mathcal{H}, \mathbf{1}_{\mathcal{H}_T}(h) = 1\}$$

If $T > T_0$, then

$$\mathcal{H}_T = \{h \in \mathcal{H}, p(h) = 1\} = \{h \in \mathcal{H}, \lim_{T\to\infty} \mathbf{1}_{\mathcal{H}_T}(h) = 1\} = \mathcal{H}^\star$$

We now argue that $\mathcal{H}^\star$ contains all length generalizing solutions. Since $f \in \mathcal{H}_t$ for all $t \geq 1$, $f \in \mathcal{H}^\star$. Now let us suppose that there is a $g \in \mathcal{H}^\star$, which does not length generalize. In other words, this $g$ leads to a non-zero error for some finite length $\tilde{T}$. Thus $g$ cannot be in $\mathcal{H}_{\tilde{T}}$. From the definition of limit, it follows that $\mathcal{H}^\star \subseteq \mathcal{H}_t$ for all $t$. This leads to contradiction. Hence, such a $g$ cannot exist. Thus all the solutions in $\mathcal{H}^\star$ the set length generalize, which proves the claim.

$\square$

# D  EXPERIMENTS

Here we provide additional experimental results as well as the training details.

**Model Architecture** In all the architectures, there are two types of non-linearities, $\omega$ that generates the target label, $\psi$ that operates on inputs (used in deep sets and transformers). We use MLPs to implement these non-linearities. We instantiate MLPs with $l$ hidden layers, and the input, output, and hidden dimensions are all the same $m = n = k$. Recall that under the realizability assumption $f \in \mathcal{H}$. Therefore, we need to select the labeling function from $\mathcal{H}$. To do so, the weights of MLP

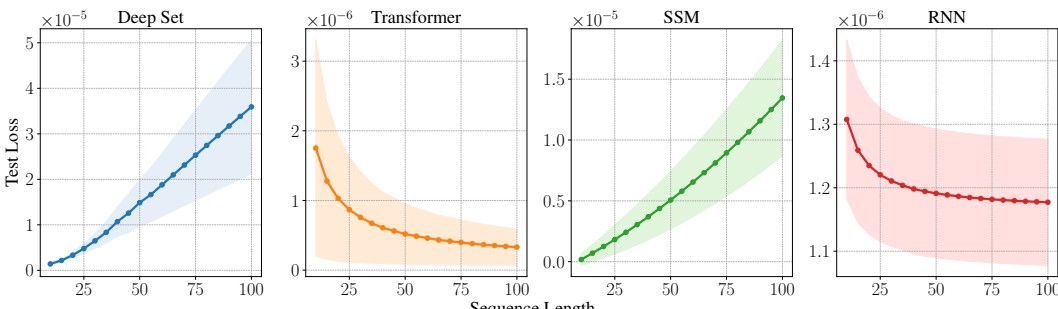

Figure 5: Length generalization: Test $\ell_2$ loss on sequences of different lengths. The models are trained only on sequences of length up to $T = 10$. All models achieve small error values $\approx 10^{-5} - 10^{-6}$ at all sequence lengths and thus length generalize. Since the error values are already quite small, the increasing or decreasing trends are not numerically significant.

are initialized according to $\mathcal{N}(\mu, \sigma^2)$, where $\mu = 0.0, \sigma = 0.6$. For RNNs and SSMs, $A, B, \Lambda$ are initialized separately for the learner and true generating process as orthogonal matrices. All hidden layers, as well as the output layer are followed by a sigmoidal activation function.

**Training Details and Hyperparameter Selection** We train all models with AdamW optimizer (Loshchilov & Hutter, 2019) with a learning rate of $10^{-3}$, weight decay of $0.01$, $\epsilon = 10^{-8}, \beta_1 = 0.9, \beta_2 = 0.95$. We reduce the learning rate by a factor of $0.8$ if the validation loss is not improved more than $10^{-6}$ for 1 epoch. This drop is followed by a cool-down period of 1 epoch, and the learning rate cannot decrease to lower than $10^{-7}$. For all datasets we use a streaming dataset where each epoch contains 100 batches of size 256 sampled online from the specified training and test distributions, and we train all models for 100 epochs. Therefore, the size of the training dataset is $256 \times 10^4$ and the size of the testing dataset is $256 \times 10^2$. Since our models are generally small, running the experiments is rather inexpensive, and we carried out each experiment on 4 CPU cores using 20 GB of RAM. For inference, specially for SSM and RNN with very long sequences, we use RTX8000 GPUs.

### D.1    LENGTH GENERALIZATION

In Figure 5, we present additional findings for length generalization capability of all architectures when both the learner and the generating process MLPs all consist of one hidden layers with input, output, and hidden size matching $n = m = k = 20$.

To complement Figure 3, in Figures 6, 7 we present the prediction behaviour of SSM and RNN architectures with two hidden layer MLPs for $\omega$ trained on sequences output by two hidden layer MLPs for $\rho$.

Figures 8, 9, 10, 11 present the prediction behaviour of deep set, Transformer with softmax attention, SSM, and RNN architectures with one hidden layer in $\rho$ (and one hidden layer MLPs for the learner $\omega$). Training procedure remains the same. We can observe that all models length generalize.

Additionally, to support the theory on other types of attention, Figures 12, 13 demonstrate the loss and prediction of a Transformer with ReLU attention and one hidden layer MLPs for $\omega, \psi$ trained on output sequences of a Transformer with ReLU attention and one hidden layer MLP for $\rho, \phi$. Similarly, all these models were trained to predict sequences of length up to $T = 10$ output by a true labeling function $f$ in their respective hypothesis classes $\mathcal{H}$, and were tested with sequences of length up to 100. As a reminder, the output tokens $y_i \in \mathbb{R}^m$, where $m = 20$, and the figures below show only one representative dimension for illustration. All models demonstrate strong length generalization capacity.

**Discrete Tokens** In Table 3 we present the results for successful length generalization of the different architectures when the inputs are discrete. We sample all components from $[0, 1]$ interval and discretize the values to one of the 5 levels in $[0.0, 0.2, 0.4, 0.6, 0.8]$. Note that the small scale of values of loss at longer lengths indicate successful generalization. For a visual depiction of results,

| Model | Test Loss $\times 10^6$ ($t = 10$) | Test Loss $\times 10^6$ ($t = 90$) |
|---|---|---|
| Deep set | $3.48 \pm 0.15$ | $52.7 \pm 0.88$ |
| Transformer | $1.72 \pm 0.27$ | $48.8 \pm 2.44$ |
| SSM | $0.2 \pm 0.0$ | $4.06 \pm 0.0$ |
| RNN | $0.22 \pm 0.0$ | $1.3 \pm 0.0$ |

Table 3: Length generalization of different architectures when the input tokens are discrete. Models are trained in sequences of length up to $T = 10$ and show successful generalization on much longer sequences.

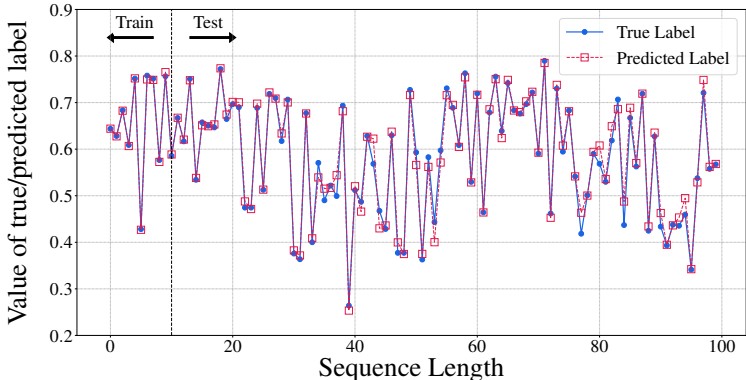

Figure 6: A SSM model with *two* hidden layer MLP for $\omega$ trained on sequences of length up to $T = 10$ length generalizes to sequences of length up to 100.

please see Fig. 14. Also note that in all architectures $\rho$ in $f$ and $\omega$ in $h$ are comprised of two hidden layers.

## D.2 COMPOSITIONAL GENERALIZATION

Here we present the prediction behavior of different architectures on the test sequences that consist of unseen token combinations during training. This helps us better interpret qualitatively how the model actually performs in following the true labels. Figures 16- 19 show the prediction trajectories for different architectures. We can observe that not only do these models perform quite well on unseen sequences of length up to $T = 10$, but they also length generalize and continue to remain consistent with the true labels on unseen combinations at longer lengths than the training. Table 4 presents the test loss and $R^2$ on the test set when the model is only trained on the red region in

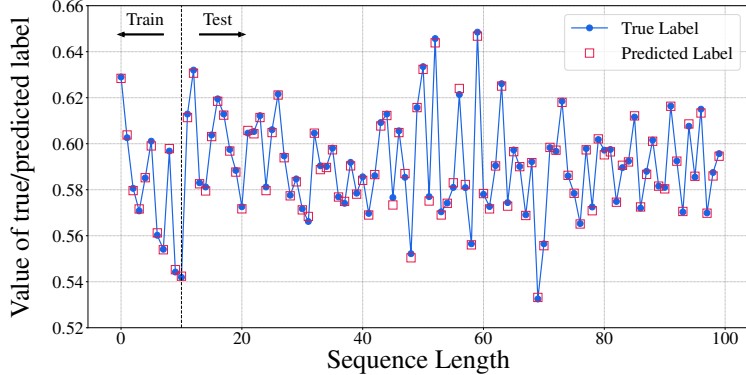

Figure 7: A RNN model with *two* hidden layer MLP for $\omega$ trained on sequences of length up to $T = 10$ length generalizes to sequences of length up to 100.

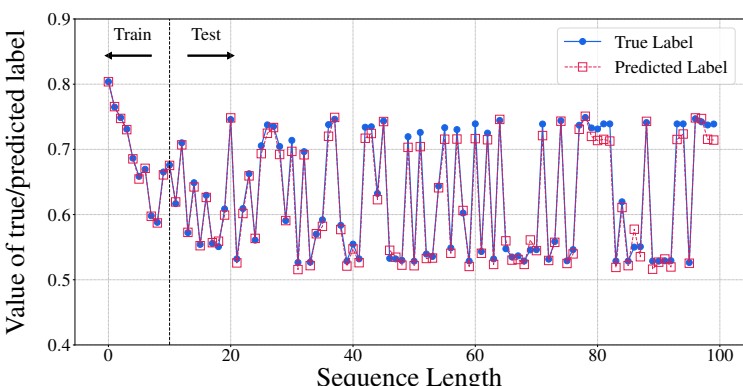

Figure 8: A deep set model with one hidden layer MLP for $\psi, \omega$ trained on sequences of length up to $T = 10$ shows perfect generalization to sequences of length up to 100.

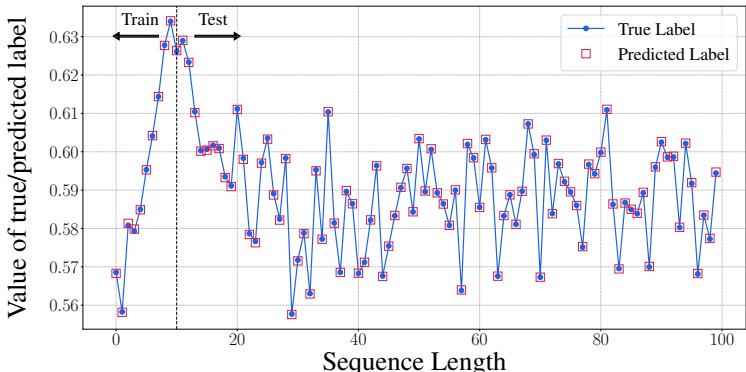

Figure 9: A Transformer model with softmax attention and one hidden layer MLP trained on sequences of length up to $T = 10$ shows perfect generalization to sequences of length up to 100.

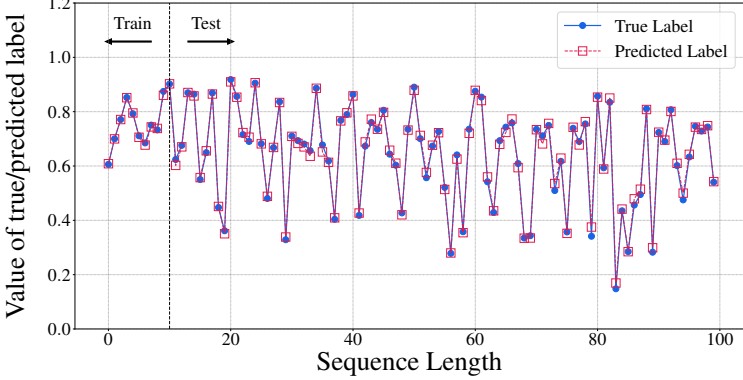

Figure 10: A SSM model with one hidden layer MLP for $\omega$ trained on sequences of length up to $T = 10$ length generalizes to sequences of length up to 100.

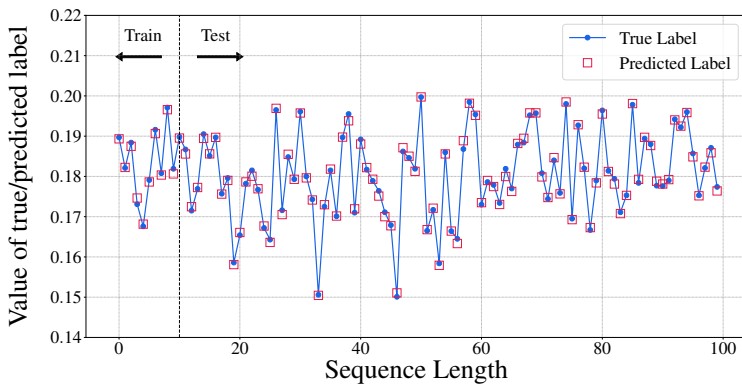

Figure 11: A RNN model with one hidden layer MLP for $\omega$ trained on sequences of length up to $T = 10$ length generalizes to sequences of length up to 100.

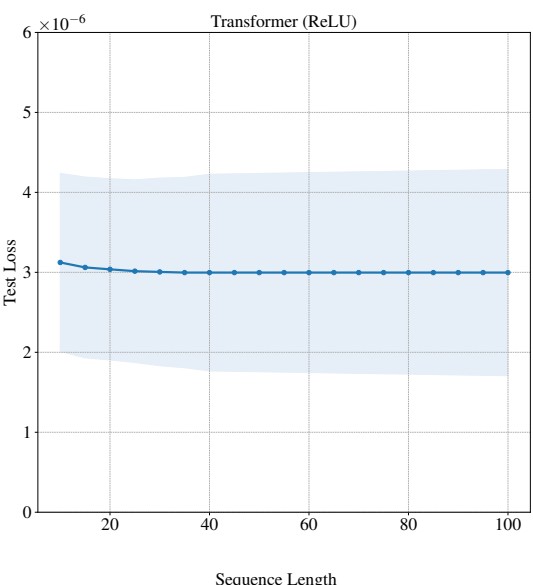

Figure 12: Test loss of a transformer model with ReLU attention and one hidden layer MLP for $\omega, \psi$ trained on sequences of length up to $T = 10$ length generalizes to sequences of length up to 100. The results are averaged over five seeds.

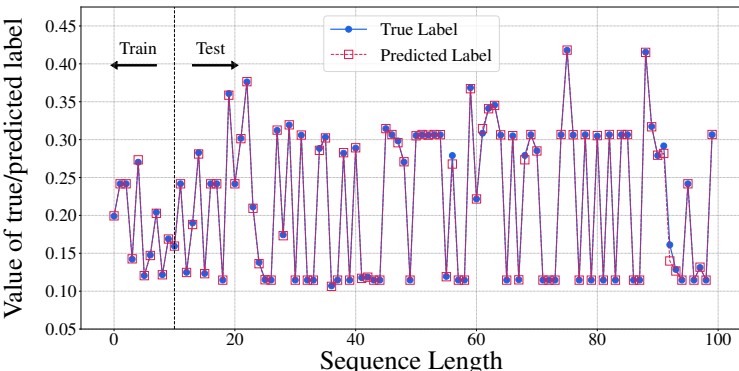

Figure 13: A transformer model with ReLU attention and one hidden layer MLP for $\omega, \psi$ trained on sequences of length up to $T = 10$ length generalizes to sequences of length up to 100.

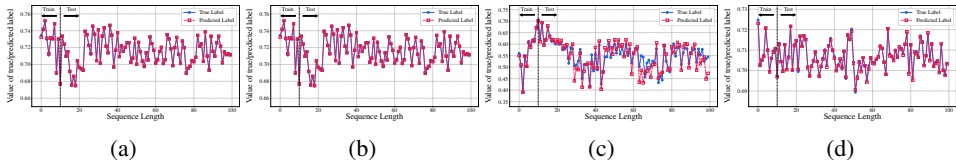

(a)  (b)  (c)  (d)

Figure 14: Successful length generalization of different architectures (with 2 hidden layers for $\rho$) when the input is discrete. From the left: Deep set, Transformer, SSM, RNN.

| Model | Test Loss $\times 10^6$ | $R^2$ |
|---|---|---|
| Deep set | $1.27 \pm 0.24$ | $0.96 \pm 0.01$ |
| Transformer | $4.50 \pm 3.28$ | $1.00 \pm 0.00$ |
| SSM | $11.00 \pm 10.92$ | $1.00 \pm 0.00$ |
| RNN | $1.22 \pm 0.12$ | $0.99 \pm 0.00$ |

Table 4: Compositional generalization: Test $\ell_2$ loss and $R^2$ score for models with one hidden layers on sequences of length $T = 10$. A strong linear relationship is observed for all models for new sequences made of unseen token combinations.

Figure 15. All models generalize to unseen combination of tokens and the learned representations linearly identify the true hidden representations.

Figures 20, 21, 22, 23 present the prediction behaviour of deep set, transformer with softmax attention, SSM, and RNN architectures with two hidden layers in $\rho$ (and two hidden layer MLPs for the learner $\omega$) when trained on sequences of length up to $T = 10$ sampled from the red region in Figure 15. We can observe that all models continue to generalize to unseen combinations beyond their training length.

**Discrete Tokens** Evaluating compositional generalization with discrete tokens introduces additional challenges. This is because we have to sample the training and test distribution according to Fig. 1 (and 15). There are multiple ways to achieve this but they become infeasible with long sequences of interest in practice:

- We could continuously sample from the regions in Fig. 1 and 15 and then round up or down the components to one of the predefined set of values. However, with longer sequences this translates to sampling and then rounding values in a high-dimensional hyper-diamond where points are increasingly spread out toward the boundaries. Rounding up results in corners becoming part of the training samples, corrupting the test set. Rounding down will result in a training set in which the support of tokens no longer follows Fig. 1 (i.e., does not cover the discrete set of values predefined in $[0, 1]$).

- We could instead sample continuously and then discretize based on finding the nearest neighbour of each point to the points in a discrete grid of values in $\mathbb{R}^T$. Having as few as 5 discrete levels renders this sampling procedure impossible for long sequences due to the complexity of finding nearest neighbours.

- Lastly, one could construct the set of discrete points in $\mathbb{R}^T$ that satisfy the constraints in Fig. 1 and then sample from this set, however, this enumeration also proves infeasible as the search space grows exponentially.

Therefore, evaluating compositional generalization in the discrete case is not straightforward beyond very short sequences.

**Practical Considerations** For training and evaluating compositional aspect of generalization, we follow the sampling procedure described in Figure 1 with a slight modification that allows for faster sampling and easier training. This procedure is illustrated in Figure 15, and results in a more difficult testing strategy, as the test set spans a smaller area than the complement of the training set.

We opted for such a procedure because rejection sampling from the complement of the training set given in Figure 1 is extremely slow. In particular, given our batch size of 256, token dimension $n =$

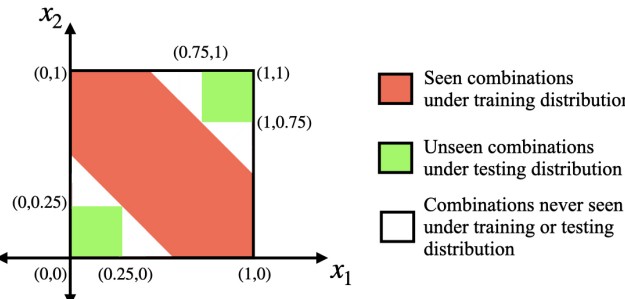

Figure 15: Illustrating the modified support of train vs test distribution for compositional generalization. This enables speed up in the sampling procedure, while keeping the challenging aspect of generalization to the corners.

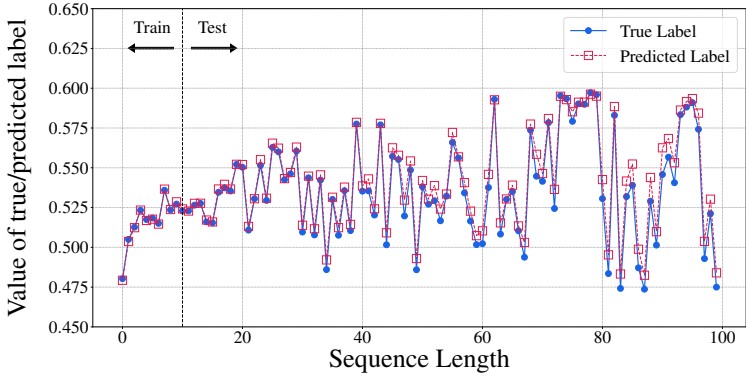

Figure 16: A deep set model with one hidden layer MLP for $\omega, \psi$ trained on sequences of length up to $T = 10$ sampled according to Figure 15 can generalize to unseen test sequences (Figure 15). Additionally, the compositional generalization holds even beyond the training length.

20, and having 100 batches per epoch, constructing the full test set requires finding $256 \times 100 \times 20$ sequences of length $t \leq T$ that are rejected by the original constraints. This becomes quite inefficient and slow specially in higher dimensions as the sum of the sequence along each component tends to concentrate more around $t/2$, therefore it becomes harder to find such sequences (the sum follows Irwin-Hall distribution since the components come from the Uniform distribution). To improve the speed of sampling the test dataset, we sample token dimensions $x_i^k$ from the smaller corners shown in Figure 15 which allows for parallel sampling. These corners correspond to sampling $x_i^k \sim$ Uniform$[0, 1/2T]$ or $x_i^k \sim$ Uniform$[1/2 + 1/2T, 1]$. This way we can sample token components independently and in parallel without having to reject any samples, since by construction no test sequence coincides with the training set. This procedure leaves a gap (see Figure 15) that will not be sampled neither during training nor testing.

### D.3 FAILURE CASES

Although most of our focus has been on the success scenarios for length and compositional generalization, here we provide examples to show how a model might fail.

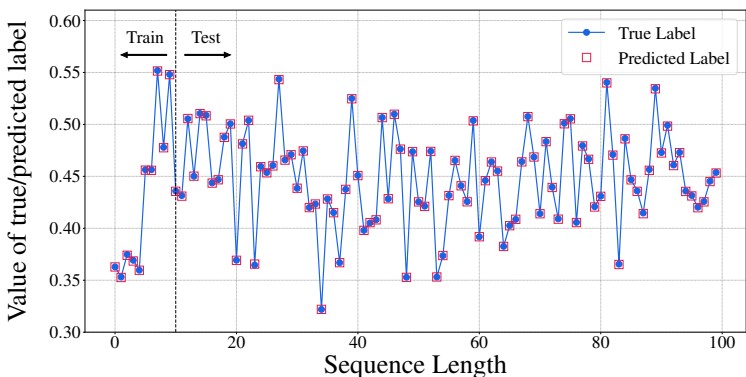

Figure 17: A Transformer model with softmax attention and one hidden layer MLP for $\omega$ trained on sequences of length up to $T = 10$ sampled according to Figure 15 can generalize to unseen test sequences (Figure 15). Additionally, the compositional generalization holds even beyond the training length.

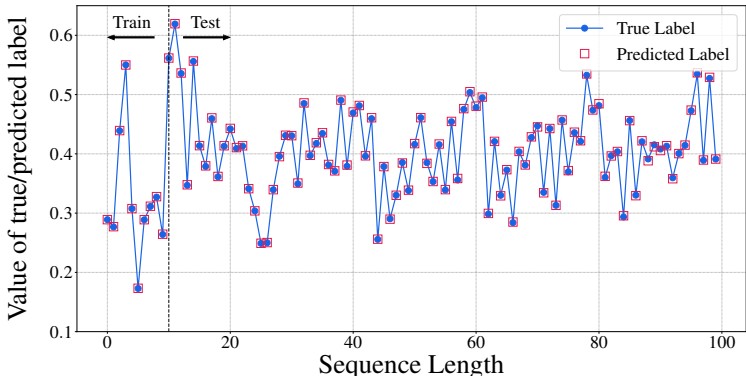

Figure 18: A SSM model with one hidden layer MLP for $\omega$ trained on sequences of length up to $T = 10$ sampled according to Figure 15 can generalize to unseen test sequences (Figure 15). Additionally, the compositional generalization holds even beyond the training length.

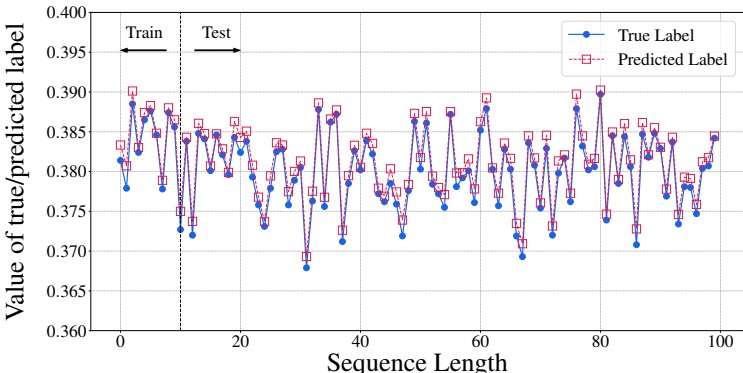

Figure 19: A RNN model with one hidden layer MLP for $\omega$ trained on sequences of length up to $T = 10$ sampled according to Figure 15 can generalize to unseen test sequences (Figure 15). Additionally, the compositional generalization holds even beyond the training length.

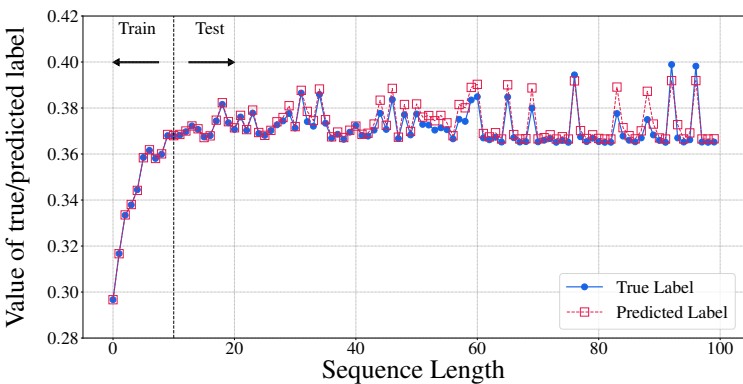

Figure 20: A deep set model with *two* hidden layer MLP for $\omega, \psi$ trained on sequences of length up to $T = 10$ sampled according to Figure 15 can generalize to unseen test sequences. Additionally, the compositional generalization holds even beyond the training length.

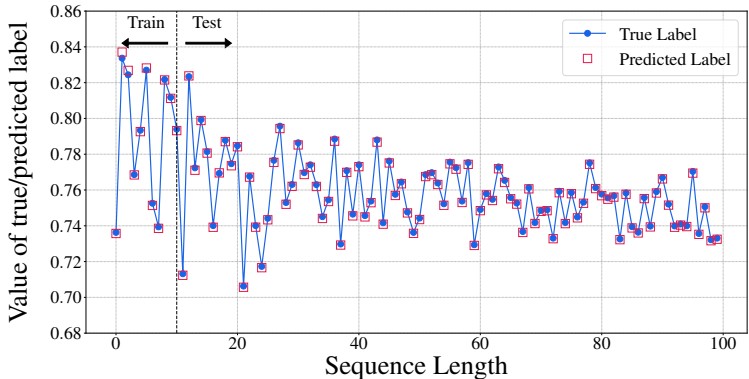

Figure 21: A Transformer model with softmax attention and *two* hidden layer MLP for $\omega$ trained on sequences of length up to $T = 10$ sampled according to Figure 15 can generalize to unseen test sequences. Additionally, the compositional generalization holds even beyond the training length.

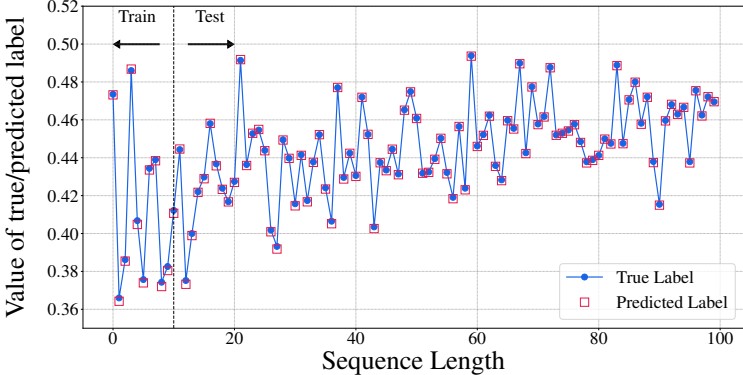

Figure 22: A SSM model with *two* hidden layer MLP for $\omega$ trained on sequences of length up to $T = 10$ sampled according to Figure 15 can generalize to unseen test sequences. Additionally, the compositional generalization holds even beyond the training length.

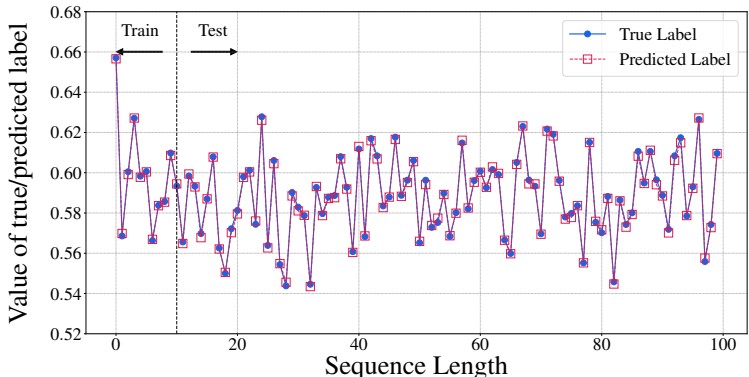
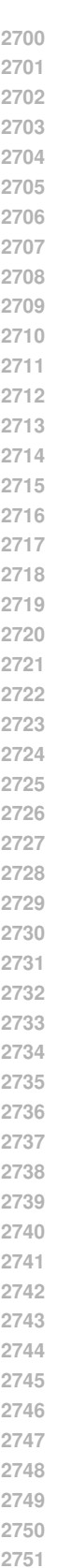

Figure 23: A RNN model with *two* hidden layer MLP for $\omega$ trained on sequences of length up to $T = 10$ sampled according to Figure 15 can generalize to unseen test sequences. Additionally, the compositional generalization holds even beyond the training length.

**$f$ is not realizable in $\mathcal{H}$** In Figures 25, 26, 27, 28, we present the predictions of learned models from different architectures initialized in their respective $\mathcal{H}$ that does not contain the true $f$. In particular, we have the following for the different architectures:

- Deep set: The true labeling function $f$ is a deep set with one hidden layer MLPs for $\rho, \phi$, but the learner uses $h$, a deep set model for which the MLPs $\psi, \omega$ have no hidden layers.

- Transformer: The true labeling function $f$ is a Transformer with 1 hidden layer in $\rho$, but the learner uses $h$, an RNN with 1 hidden layer in $\omega$.

- SSM: The true labeling function $f$ is an SSM with 1 hidden layer in $\rho$, but the learner uses $h$, an RNN without any hidden layers in $\omega$.

- RNN: The true labeling function $f$ is an RNN with 1 hidden layer in $\rho$, but the learner uses $h$, a Transformer with 1 hidden layer in $\omega$.

In each case, the learner is trained on sequences of length up to $T = 10$ and its performance on the test set at longer lengths indicates whether generalization is possible or not. For a visual illustration of such failures beyond the training length, see Figures 25, 26, 27, 28. We can observe that the model can predict the test sequence well up to the length it has learned during training, but starts to diverge from the true labels beyond that. This demonstrates a failure case in which the realizability condition is violated.

**$f$ is realizable in a high capacity $\mathcal{H}$** For a given $\mathcal{H}$, if all solutions to 1 achieve length generalization or compositional generalization, then we can guarantee length or compositional generalization regardless of the training procedure. When the capacity of $\mathcal{H}$ becomes very large, it continues to contain the right solutions but it starts to contain many incorrect solutions that match the true solution only on the support of training distribution. In such a case, there is no reason to presume that our learning procedure picks the right solution to 1 that also achieves length and compositional generalization. Figure 24 show experiments illustrating the above. We experiment with the following scenarios for deep sets and transformers:

- Deep set: We use the labeling function that takes the following form $f = \rho(\sum_{i \leq t} \phi(x_i))$ for $t \leq T$ and $f = \rho(\sum_{i \leq t} \phi(x_i)) + c$ for $t > T$ with $c = 0.2, T = 5$. We use 1 hidden layer MLPs for $\rho, \phi$ (with no activation on the output of $\rho$). We use 2 hidden layer MLPs for $\omega, \psi$ for $h$ so that it can express the above labeling function. The input, hidden, and output dimensions are all equal $m = n = k = 20$ for $f, h$. We train on sequences of length longer than $T$ to demonstrate this expressivity claim. When the model is trained on sequences of length less than $T$, due to the simplicity bias of the training procedure model learns $\rho(\sum_{i \leq t} \phi(x_i))$ and uses it on longer sequences and hence fails.

- Transformer: We use the labeling function that takes the following form $f = \rho(\sum_{j=1}^{i} \frac{1}{i} \phi(x_i, x_j))$ for $t \leq T$ and $f = \rho(\sum_{j=1}^{i} \frac{1}{i} \phi(x_i, x_j)) + c$ for $t > T$ with

| Deep set | Loss ($t < T_0$) | Loss ($t \geq T_0$) |
|---|---|---|
| Fig 2-a | $0.001 \pm 10^{-4}$ | $0.002 \pm 3 \times 10^{-4}$ |
| Fig 2-b | $0.0007 \pm 10^{-4}$ | $0.007 \pm 0.001$ |
| Transformer | Loss ($t < T_0$) | Loss ($t \geq T_0$) |
| Fig 2-c | $0.0006 \pm 10^{-4}$ | $0.006 \pm 3 \times 10^{-3}$ |
| Fig 2-d | $10^{-5} \pm 10^{-6}$ | $0.01 \pm 0.003$ |

Table 5: Length generalization of different architectures when the hypothesis class $\mathcal{H}$ is highly expressive. For further details see Fig. 24

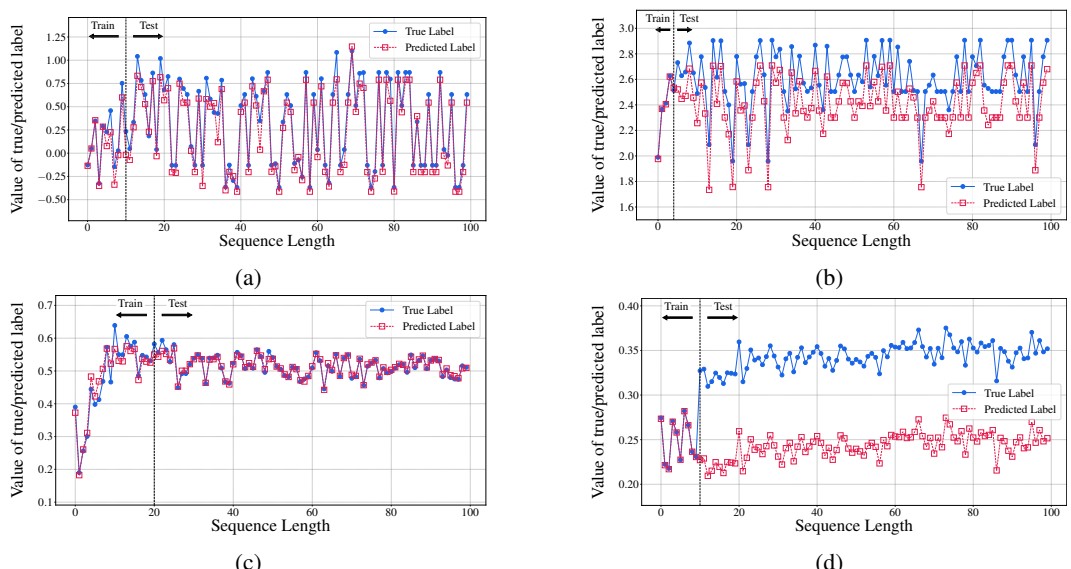

Figure 24: A failure case of length generalization under arbitrary expressive generative model with (a,b) Deep sets, (c,d) and Transformer. The generative function on both cases introduces an offset to sequences longer than some critical length ($T_0$). The learner is once trained on sequences longer than $T_0$ and successfully generalizes (a,c), and once is trained only on sequences shorter than $T_0$ where the offset never appears, and hence fails to generalize beyond that.

$c = 0.1, T = 10$. We use 1 hidden layer MLPs for $\rho$ (with no activation on the output of $\rho$). We use a Transformer with 3 hidden layer MLPs for $\omega$ so that it can express the above labeling function. The input, hidden, and output dimensions are all equal $m = n = k = 20$ for $f, h$. We train on sequences of length longer than $T$ to demonstrate this expressivity claim. When the model is trained on sequences of length less than $T$, due to the simplicity bias of the training procedure model learns $f = \rho(\sum_{j=1}^{i} \frac{1}{i}\phi(x_i, x_j))$ and uses it on longer sequences and hence fails.

The failures of such degenerate solutions can be visualized in Figure 24 (right), where the predictions diverge from the true values when the model is only trained on sequences shorter than $T_0$. Figure 24 (left) shows that when the model is trained on sequences longer than $T_0$, it can successfully generalize to longer lengths. Table 5 further validates this observation numerically. It presents the test loss of each model at lengths shorter and longer than $T_0$ under the two training schemes: a) When trained only on sequences of length shorter than $T_0$ (rows corresponding to Fig 2-b and 2-d which result in failure due to degenerate solution), b) when trained on sequences of length longer than $T_0$ (rows corresponding to Fig 2-a and 2-c which result in successful generalization).

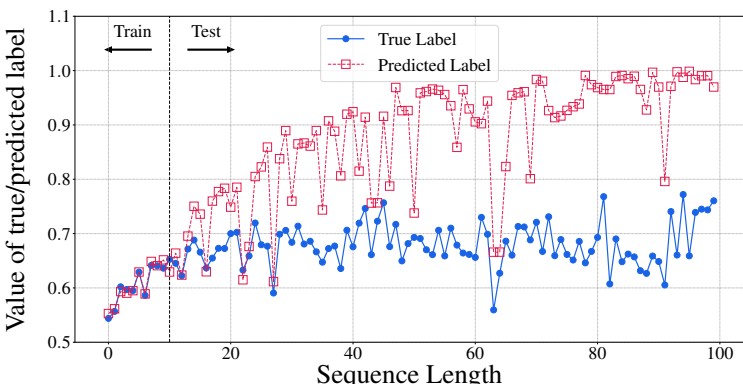

Figure 25: A failure case of length generalization in the unrealizble setting: The predictions come from a deep set with linear layers for $\psi, \omega$ trained to predict the sequences (of length up to $T$) output by a deep set with 1 hidden layer MLPs for $\phi, \rho$. In this case the realizability condition does not hold, and the learner fails to length generalize.

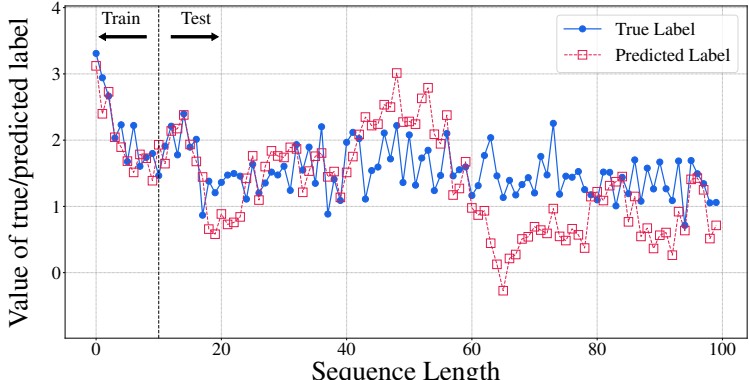

Figure 26: A failure case of length generalization in the unrealizble setting: The predictions come from an RNN with 1 hidden layer in $\omega$ trained to predict the sequences (of length up to $T = 10$) output by a Transformer with 1 hidden layer in $\rho$. In this case the realizability condition does not hold, and the learner fails to length generalize.

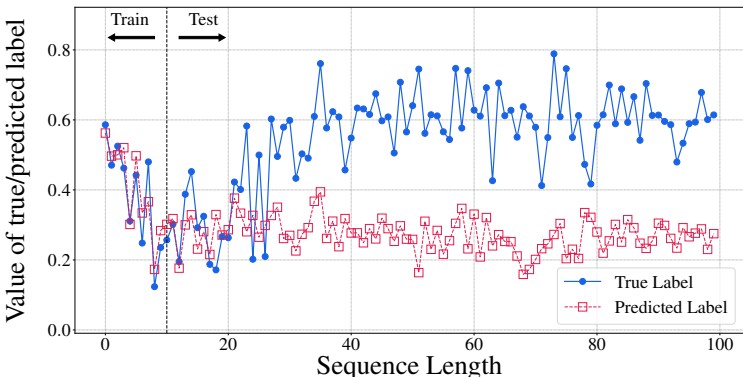

Figure 27: A failure case of length generalization: The predictions come from an RNN without any hidden layers in $\omega$ trained to predict the sequences (of length up to $T = 10$) output by an SSM with 1 hidden layer in $\rho$. In this case the realizability condition does not hold, and the learner fails to length generalize.

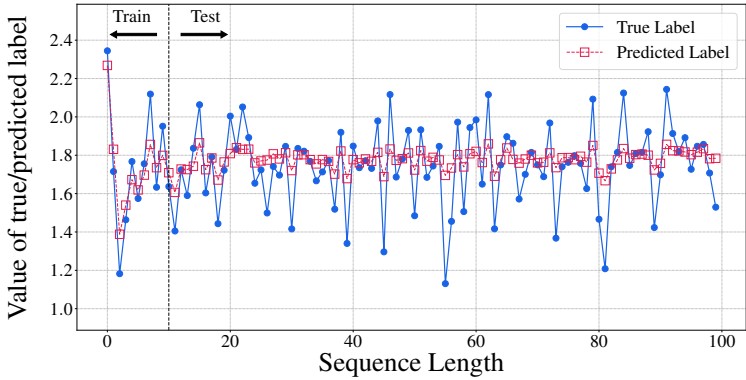

Figure 28: A failure case of length generalization: The predictions come from a Transformer with 1 hidden layer in $\omega$ trained to predict the sequences (of length up to $T = 10$) output by an RNN with 1 hidden layer in $\rho$. In this case the realizability condition does not hold, and the learner fails to length generalize.

