# OpenReview forum: "On Provable Length and Compositional Generalization"
_ICLR.cc/2025/Conference — Submitted to ICLR 2025_

### Official Review · Reviewer_MsSC · 2024-10-30

**Soundness:** 3
**Presentation:** 3
**Contribution:** 3
**Rating:** 6
**Confidence:** 4

**Summary:**

The paper theoretically studies length generalization and compositional generalization in various simplified models: deep sets, simplified one-layer transformers and simple RNNs and state space models. All results are given in the realizable case, i.e. when the ground-truth labels are generated by a model from the hypothesis class, and assuming some capacity restriction on the class. The authors define length generalization to be a setting where the model minimizes the expected risk up to length T and achieves zero generalization error on longer lengths. Similarly, they define compositional generalization of a model minimizing the expected risk on some input distribution and generalizing to the Cartesian product of the support of the different variables. Then, the authors show that the simplified models achieve length and compositional generalization, under assumptions on the capacity of the models. Additionally, the authors complement their results with experiments.

**Strengths:**

To my knowledge, this is the first work defining and analyzing length generalization, and one of the first works to theoretically study compositional generalization. The authors study several different architectures for learning over sequences and sets that are used in practice and show positive results on length and compositional generalization under reasonable assumptions. Additionally, the authors show that without restricting the capacity of the models, length generalization and compositional generalization are not possible.

**Weaknesses:**

-	While the paper shows some novel theoretical results, the setting analyzed is somewhat idealized. Primarily, to my understanding, the authors assume that the model achieves zero error on the training distribution. Typically, however, theory of learning focuses on achieving (arbitrarily) small error and using a finite sample. It is not clear how these results extend to a setting where the learner achieves small, but non-zero, error on the training distribution.
-	The result on transformers extends only to positional encodings where tokens far enough from one another do not affect each other, which does not apply to positional encodings used in practice.
-	The authors can be a bit more rigorous in stating the definitions and setting. For example, in Definition 1, it is not clear what “zero generalization error” means. Does this mean zero error on the target distribution? Additionally, when assuming “a model trained on sequences of length up to train T”, does this mean that the model achieves zero error on the distribution P(T)?
-	Assumption 15, used in Theorem 4, is deferred to the appendix, which makes reading and understanding the result harder. This assumption should be stated in the main paper.
-	It is not clear what is the takeaway from the experimental section. For example, the authors plot the error of different sequence models displaying different behaviors when varying the sequence length, but then admit that all errors are very low so the conclusion from these plots is not clear. What are the questions or hypotheses that these experiments are suppose to prove or validate?

**Questions:**

-	The models studied in the paper are simplified, one layer, models. Can the results shown in the paper be extended to deep, multi-layer, models?
-	The discussion on high-capacity models, specifically the “negative results” on length and compositional generalization, assumes no constraints at all on the model, which may be too permissive, in contrast to the “positive results” which analyze a specific family of functions. Can these results be extended to general classes of functions with some more generic capacity control (e.g., finite classes or classes with bounded VC dimension)?

---

> ### Author Response · Authors · 2024-11-16
> **Response to Reviewer MsSC**
>
> We thank the reviewer for their encouraging review and brilliant questions. Please see our responses below. We look forward to hearing back from you.
>
>    1. **Regarding finite sample issues:** We want to emphasize that the challenge of out-of-distribution generalization persists even when the training data consists of infinite samples from the training distribution. Due to this most works that study out-of-distribution generalization (such as Wiedemer et al., Lachapelle et al., Arjovsky et al., Dong et al.), assume that there is infinite data from training distribution. In the experiments, we operate with finite data.
>
>
>
> 2. **About experiments:** Our theory predicts that solutions to expected risk minimization under stated conditions achieve length and compositional generalization. Our theory also predicts that learned representations are linear transforms of true representations. When working with experiments, we do not have access to infinite data and the training procedure relies on gradient descent that is not guaranteed to reach the exact minima. Hence, the experiments section is meant to validate the theory under non-ideal conditions. In Figure 2, we observe small numbers for errors, which confirms that the model achieves length generalization and validates the theory. In Table 1, we verify the linearity of representations and in Table 2, we verify the claim of compositionality.
>
>
> 3. **Regarding finite hypothesis classes:** You asked a brilliant question, i.e., if the theory extends to finite hypothesis classes or bounded VC dimensions. We have added a new result (Theorem 6), which proves this claim for finite hypothesis classes. For a finite hypothesis class, we show that if the training length is sufficiently large, then the solutions to expected risk minimization achieve length generalization.  Finite hypothesis classes contain arbitrarily deep transformers with weights constrained to a finite set of values as a special case.
>
> 4. The assumption we make on positional encoding is similar to the Hard-Alibi positional encoding scheme used in Hou et al., Jelassi et al.. Jelassi et al. also constructed the positional encoding for theoretical purposes but found it to be effective as well.
>
> 5. We have made changes to the definition of length generalization to remove any confusions you raised.  We have moved the Assumption 15 to the main body.
>
>
>
>
>
> **References**
>
>  [Wiedemer et al.] "Compositional generalization from first principles." Advances in Neural Information Processing Systems 36 (2024)
>
>    [Lachapelle et al.] "Additive decoders for latent variables identification and cartesian-product extrapolation." Advances in Neural Information Processing Systems 36 (2024).
>
>    [Arjovsky et al.] "Invariant risk minimization." arXiv preprint arXiv:1907.02893 (2019).
>
>    [Dong et al.] "First steps toward understanding the extrapolation of nonlinear models to unseen domains." arXiv preprint arXiv:2211.11719 (2022).
>
>    [Jelassi et al]. "Repeat after me: Transformers are better than state space models at copying." arXiv preprint arXiv:2402.01032 (2024).
>
>    [Hou et al.] "Universal length generalization with turing programs." arXiv preprint arXiv:2407.03310 (2024).

---

> > ### Comment · Reviewer_MsSC · 2024-11-25
> >
> > Thank you for your response. I will keep my original score.

---

### Official Review · Reviewer_n34f · 2024-11-02

**Soundness:** 3
**Presentation:** 2
**Contribution:** 2
**Rating:** 5
**Confidence:** 3

**Summary:**

This paper considers the problems of length/compositional generalization from a functional expressivity perspective. It considers a variety of similar architectures under assumptions of realizability, some intricate assumptions about the support of continuous tokens, and strongly limited capacity (limiting the interaction between tokens). In this setting it considers the ERM solutions and shows that they must recover the ground truth features up to a linear transformation which allows them to generalize. There are negative results presented for all of the architectures considered if the capacity is increased.

**Strengths:**

1. The paper proposes a (to my knowledge) novel method to study length/compositional generalization.

2. The paper does a good job of extending results to a family of related toy models of different types of architectures. This generality is a strength of the approach.

3. The paper does a good job of presenting both negative and positive results. In particular the negative results actually seem to show that once the restrictive limits on the capacity in the architectures are weakened, then generalization can easily fail.

**Weaknesses:**

1. The perspective of the paper on the prospects of length/compositional generalization seems overly optimistic. In particular, in practice achieving length/compositional has proven very difficult. The papers cited on length generalization struggle to even achieve 2x length generalization despite using all sorts of unnatural tricks. This paper seems to approach the problem as "length generalization happens and we want to explain it", where the real situation appears to be "length generalization is very difficult to achieve if even possible generally". Even in the context of the paper, perhaps the more realistic results are the negative results for each architecture showing that if you allow for (still restrictive) higher capacity, the architectures are able to represent solutions that achieve zero in-distribution error but fail to generalize.

2. It is not clear why we would expect the simplified models presented in the paper to bear any relation to things we may see in practice. Of course in theory we want to study simplified models, but the ultimate goal is for those models to bring insights to practical settings. The negative results indicate that as soon as we expand the very tight capacity restrictions, things can Simplified architectures seem to have no chance of applying to real settings.

3. There is no consideration for statistical issues. Essentially the ERM approach is assuming infinite data. This is combined with the strong assumptions of full support over continuous variables to pin down the learned function to be similar to the realizable ground-truth solution. It is not clear if the arguments presented can apply at all to the case of finite datasets, even as those datasets become very large.

4. The coverage assumptions and continuous support seem to be quite strong. It seems that the tools are very reliant on having full support over continuous variables. For transformers the assumption is even stronger than having full support over the marginals to full support over the joint distributions of each pair of tokens which is even stronger. It is not clear how these assumptions could be relaxed.

5. The paper does not study standard attention, but instead a simplified version. This should be made more clear in the abstract/intro.

6. The numerical experiments basically only verify the theory. Moreover the plots showing large trends in increasing or decreasing loss are a somewhat confusing way to claim that length generalization is always happening (because the y-axes are very small). Perhaps adding some baselines/alternatives that do not length generalize could make it more clear whether the ability of these architectures to generalize is interesting.

**Questions:**

See weaknesses

---

> ### Author Response · Authors · 2024-11-16
> **Response to Reviewer n34f**
>
> We thank the reviewer for their time and effort in their useful review. Please see our responses below. We look forward to hearing back from you.
>
>    1. We agree that length generalization has been quite difficult to achieve in general. However, there have been empirical works that have put forth a conjecture -- RASP-L conjecture that states when such generalization is achievable https://arxiv.org/pdf/2310.16028. Even if the empirical evidence indicates generalization up to 2x length, taking first steps towards theoretical explanations is important and our work is an endeavor in that direction. For the models we considered in our experiments, we already show length generalization up to 10x training length.
>
>
>    2. **On insights from simplified models**:
>    a) Firstly, we believe that our results are the first on the theory of length generalization and we should be credited for that. Secondly, we also add a new result on finite hypothesis classes (described below), which we hope helps you see our work in a more positive light. Thirdly, we believe that there are many theoretical papers that make simplifying assumptions and it is not always possible to have immediate practical insights from them. Take for example the whole line of work on PAC learning theory, or analysis of convergence in neural networks (often done with very shallow depth two (Du et al.), or the recent analysis of in-context learning (Oswald et al.) or analysis of out-of-distribution generalization (Dong et al., Wiedemer et al., Lachapelle et al.). We believe all these works are very valuable but may not give any immediate insights that change practice. However, in the big picture they contribute valuable knowledgable which slowly turns into mental models and useful practical insights.
>
>       b) We introduce a new result (Theorem 6) that assumes the hypothesis class is finite and impose no other constraints on the model. For a finite hypothesis class, we show that if the training length is sufficiently large, then the solutions to expected risk minimization achieve length generalization. To consider example of finite hypothesis classes, imagine we have a transformer network with many layers with softmax attention or an RNN. Instead of operating with the property that each scalar value of the weight can take all possible real values, we constrain the weights to only take discrete set of values.
>    3. **On statistical issues:** We want to emphasize that the challenge of out-of-distribution generalization persists even when the training data consists of infinite samples from the training distribution. Due to this most works that study out-of-distribution generalization (such as Wiedemer et al., Lachapelle et al., Arjovsky et al.), assume that there is infinite data from training distribution. In the experiments, we operate with finite data.
>    4. **Full support over marginals**: Full support over marginals (or pairs) is a far weaker assumption than any assumption on the joint support. Look at Figure 1 and Figure 4 for illustration to see that the joint support occupies a much smaller portion of the region. Such assumptions have also been made in other works such as (Wiedemer et al., Dong et al., and Lachapelle et al.). Also, observe that our results for RNN and our new result (Theorem 6) above with finite hypothesis classes does not invoke such an assumption.
>    5. To supplement Figure 2, we also have Figures 6-11, 13,14, 16-23 in the Appendix that track the predicted sequence against the true sequence for lengths beyond the training length.
>
>
> **References**
>
>  [Wiedemer et al.] "Compositional generalization from first principles." Advances in Neural Information Processing Systems 36 (2024)
>
>    [Lachapelle et al.] "Additive decoders for latent variables identification and cartesian-product extrapolation." Advances in Neural Information Processing Systems 36 (2024).
>
>    [Arjovsky et al.] "Invariant risk minimization." arXiv preprint arXiv:1907.02893 (2019).
>
>    [Dong et al.] "First steps toward understanding the extrapolation of nonlinear models to unseen domains." arXiv preprint arXiv:2211.11719 (2022).
>
>    [Du et al.]  "Gradient descent provably optimizes over-parameterized neural networks." arXiv preprint arXiv:1810.02054 (2018).
>
>    [Oswald et al.] "Transformers learn in-context by gradient descent." International Conference on Machine Learning. PMLR, 2023.

---

> > ### Author Response · Authors · 2024-11-21
> >
> > Dear Reviewer
> >
> > We believe that our responses and new additions to the paper should sufficiently address your concerns. Please let us know if you have further questions. We hope that you consider increasing the score.
> >
> > Thank you!

---

> > > ### Comment · Reviewer_n34f · 2024-11-25
> > >
> > > Thanks for the response.
> > >
> > > I do appreciate the potential significance of simplified models and of the novelty of the methods introduced. However, I think that these models are useful when they speak towards the actual underlying phenomena. What I am skeptical about here is that the techniques introduced are not actually capturing the root of what is going on because they are so sensitive to things like full coverage and certain types of capacity restriction that are not related to what we see in practice. That said, I appreciate the response and the added theorem.
> > >
> > > I do want to reiterate that I think the exposition and the abstract/intro are overly optimistic about the actual results. The negative results in fact seem to be more practically realistic and length generalization is actually quite hard to make arise.
> > >
> > > I will increase my score to a 5.

---

> ### Author Response · Authors · 2024-11-25
>
> We thank the reviewer for their response and raising the score. Please take our responses below into consideration.
>
>
> 1. **Coverage-based assumptions:** Our coverage-based assumptions are inspired by Wiedemer et al. These assumptions seem necessary to even be able to meaningfully define the notion of compositionality.
>
> 2. **Abstract and introduction:** Our abstract and introduction explicitly state that our results are developed for simple, limited-capacity models. We would appreciate it if you could specify which part of the abstract and introduction you found overly optimistic.
>
> 3. **Simplifying assumptions in theoretical frameworks:** Most theoretical frameworks in deep learning often make simplifying assumptions. For instance, analysis of shallow networks is often used to draw stylized insights for the general phenomenon at hand. Our theory is inspired by the RASP-L conjecture proposed in Zhou et al., which identifies conditions under which transformers empirically generalize on real algorithmic tasks, such as copy, mode, addition, and sort. The RASP-L conjecture identifies three conditions - realizability, simplicity, and data diversity - but did not provide a formal proof. Our theory captures these three conditions and proves results for a range of architectures and a range of tasks (e.g., multiplication, addition, tasks expressed by finite automata through our result on finite hypothesis classes).
>
> We hope our above response changes your perspective on our work.
>
> **References**
>
> Wiedemer, et al. "Compositional generalization from first principles." Advances in Neural Information Processing Systems 36 (2024).
>
> Zhou et al. "What algorithms can transformers learn? a study in length generalization." arXiv preprint arXiv:2310.16028 (2023).

---

### Official Review · Reviewer_JxtZ · 2024-11-02

**Soundness:** 3
**Presentation:** 4
**Contribution:** 3
**Rating:** 5
**Confidence:** 4

**Summary:**

This paper analyzes length generalization and compositional generalization in deep sets, transformers, SSMs, and RNNs, proving that these models achieve both types of generalization under certain assumptions. Additionally, the paper demonstrates that the representations learned by each model in these settings are linearly related to the true labeling function, indicating linear identification. Experimental results further confirm the occurrence of linear identification in real models.

**Strengths:**

- This paper provides a clear definition of length generalization and compositional generalization in sequence-to-sequence models and demonstrates valuable guidance on how to analyze these properties. This serves as a strong foundation for future work in this area.
- The figures effectively illustrate the concepts of length generalization and compositional generalization, and the rationale behind focusing on simple limited capacity models is explained in a convincing manner.
- The paper not only examines a range of models, but also extends its theorems to cover various scenarios, such as $C^1$-diffeomorphisms and discrete tokens.

**Weaknesses:**

- **Strong assumption in hypothesis selection**: As acknowledged by the authors in the discussion and limitations section, the assumption, stated as equation 1, that a hypothesis is chosen to minimize expected risk may be a bit strong.
- **Repetitive proofs**: Some sections of the proofs appear to rely heavily on copy-pasting, leading to redundancy and potential errors. For example:
  + On page 28, line 1483, the reference to “equation 31 of equation 30” seems incorrect and should likely be “equation 67 of equation 66.”
  + At the beginning of Appendix C.3.1, “Assumption 2” appears, which may be intended to refer to “Assumption 15.”

  If possible, reducing redundancy by grouping similar arguments into a single lemma could help minimize errors and improve readability.
- **Extension to discrete tokens**: While the extension of the theorems to discrete tokens is particularly intriguing, some aspects of the proof raise questions:
  + This overlaps slightly with the second point. The phrase “regular closedness of the support” appears in Theorems 6,9,11, and 12, but for discrete tokens, the support is not regular closed, and the “equality almost everywhere -> equality everywhere” arguments may not be necessary after all.
  + The reasoning from equation 30 to 33 is somewhat unclear; could you refine this explanation a bit? Is it not possible to derive equation 33 directly by setting $x_1 = x_2$ in equation 30?
- **Presentation issues**:
  + In the statements of Theorems 3 and 4, Assumptions 12 and 15 are referenced before being defined. It would be helpful to include these assumptions in the main text, or alternatively, to provide informal statements of Theorems 3 and 4 in the main text and formalize them in the Appendix.
  + In equation 66, $\psi$ and $\phi$ are swapped.
  + In the second line of equation 108, the left-hand side would be more readable if written as $\left.\frac{\partial^p \sigma(s)}{\partial s^p}\right|_{s=v^\top y}$.
  + The formula on page 37, line 1987, should have line breaks before and after it, with an added explanation to improve readability.

**Questions:**

- This question applies to the proofs of nearly all the theorems; as an example, let’s consider the case of deep sets: if we are considering a hypothesis that minimizes the expected risk $R(h;T) = \sum_{t=1}^T \mathbb{E}\left[\ell(h(x_{\leq t},y_t)\right]$, why is it not possible to directly derive $A\psi(x) = B\phi(x)$ for all $x \in [0,1]^n$ from the equality $\omega\left(\psi(x)\right) = \rho\left(\phi(x)\right)$ for all $x \in [0,1]^n$ in the case of $t=1$?
- How realistic is Assumption 18? It would be helpful to see concrete examples that illustrate this assumption.

---

> ### Author Response · Authors · 2024-11-16
> **Response to Reviewer JxtZ**
>
> We thank the reviewer for their efforts in a dilligent review and constructive feedback. We especially appreciate reviewer's eye for detail. Please see our responses below. We look forward to hearing back from you.
>
>    1. **Regarding equation 1**: It is not clear what part of equation 1 you are objecting to. We presume you are objecting to the realizability condition, i.e., the labeling function is in the hypothesis class. Firstly, the realizability assumption is a part of the RASP-L conjecture (Zhou et al.) that motivated our work. Secondly, the assumption of realizability is made in most works studying different forms of out-of-distribution generalization (Wiedemer et al., Lachapelle et al., Arjovsky et al., Dong et al.). It is tricky to study generalization beyond the training distribution in the absence of such an assumption.
>
>    2. **Repetitive proofs and other fixes**: We kept some of the proofs separate as some steps change between similar proofs and may not be obvious to the reader. We have fixed the equation referencing errors. We have removed the statements on regular closedness from the discrete token proofs. We have shortened the description from equation 30-33 and used your suggestion.  We have included the assumptions that were referred to from th Appendix in the main body. We have corrected the psi and phi notation swaping error.  We have made other presentation changes that you pointed to.
>
>
>    3. **On equating predictions at $t=1$:** Your comment intends to show that we can obtain $A\psi = B\phi$ from $t=1$. Firstly, your comment is applicable to only Theorem 1 and Theorem 2 and there is an important extension that I discuss in a bit. The comment is not applicable to more general extensions for deep sets and transformers presented in Theorem 8 and Theorem 3, and also not to any of the results in SSMs and RNNs. Further, in the Appendix, below Theorem 1 and 2, we explain that our results can also be extended to setting where we do not observe data for t=1 and only observe data at some length T>1. In this case, when we do not observe data at t=1 we cannot apply your observation.
>
>
>    7. **Regarding Assumption 18:** Assumption 18 is a form of diversity assumption. Suppose change along each dimension of the token captures some concept (e.g., gender). The first part of the assumption requires that we observe at least two instances of the concept change appear -- ("Man", "Woman"), ("Boy", "Girl"). The second part of the assumption requires that we observe concept change independently for each dimension -- ("Man, Boy"), ("Woman, Boy"), ("Man, Girl"), ("Woman, Girl"). In this example, along each of the two dimensions, we observe gender change.
>
>
>
>   **References**
>
>    [Orvieto et al.] "On the universality of linear recurrences followed by nonlinear projections." arXiv preprint arXiv:2307.11888 (2023).
>
>    [Wiedemer et al.] "Compositional generalization from first principles." Advances in Neural Information Processing Systems 36 (2024)
>
>    [Lachapelle et al.] "Additive decoders for latent variables identification and cartesian-product extrapolation." Advances in Neural Information Processing Systems 36 (2024).
>
>    [Arjovsky et al.] "Invariant risk minimization." arXiv preprint arXiv:1907.02893 (2019).
>
>    [Dong et al.] "First steps toward understanding the extrapolation of nonlinear models to unseen domains." arXiv preprint arXiv:2211.11719 (2022).
>
>    [Zhou et al.] "What algorithms can transformers learn? a study in length generalization." arXiv preprint arXiv:2310.16028 (2023).

---

> > ### Author Response · Authors · 2024-11-21
> >
> > Dear Reviewer
> >
> > We believe that our responses and changes to the paper should sufficiently address your concerns. Please let us know if you have further questions. We hope that you consider increasing the score.
> >
> > Thank you!

---

> > > ### Author Response · Authors · 2024-11-26
> > >
> > > Dear Reviewer
> > >
> > > We have greatly valued your feedback and incorporated it all in our revisions. We hope that you would consider increasing your score if you are happy with our responses. If not, please let us know your concerns.
> > >
> > > Thank you

---

> > > ### Comment · Reviewer_JxtZ · 2024-11-26
> > >
> > > Thank you for your reply. Below are my comments on your rebuttal.
> > > 1. What I am referring to is that length/compositional generalization is considered with respect to the hypothesis that minimizes equation (1), and not the realizability condition itself, which I do not particularly object to.
> > > The assumption that a true risk minimizer can be obtained is much stronger than the assumptions typically made in Empirical Risk Minimization in PAC learning. I read your response to reviewer 4sDj, and if this assumption is indeed standard when studying compositional generalization, it would be helpful to clarify this by adding a note either under equation (1) or in the explanation following Definition 2, where you mention, "This definition of compositionality above is based on...", because most readers encountering this field for the first time may find this assumption rather stringent.
> > > 2. Thank you for the revisions.
> > > 3. Beyond Theorems 1 and 2, for instance, in Theorem 7, wouldn’t Equation (31) follow directly from Equation (29) by setting $T=1$? Similarly, in Theorem 10, wouldn’t Equation (65) be directly obtained from Equation (63) by setting $i = 2$?
> > > For SSMs, the situation is slightly different. However, it seems that the relationship between $B$ and $\tilde{B}$ can still be derived from the case where $t=1$, and the relationship between $\Lambda$ and $\tilde{\Lambda}$ can be obtained from the case where $t=2$.
> > > That said, this reasoning cannot be applied to Theorems 3 and 8, as demonstrating the linearity of $\omega^{-1} \circ \rho$ becomes crucial in those cases.
> > > 4. Thank you for the clear explanation. I now understand that the first part of Assumption 18 means that for every concept, the support contains two instances which differ only in terms of that specific concept. However, the assumption that the concept is represented separately in each coordinate and that for each concept the two instances differ by the same amount feels somewhat strong.

---

> ### Author Response · Authors · 2024-11-26
>
> We thank the reviewer for their responses.
>
> 1. **Adding further clarification on true risk minimizer:** Yes, we would be happy to add that clarification. In particular, we can refer to other works on compositionality that analyze true risk minimizers like us.
>
> 2.  **Regarding linear identification from T=1:** Yes, you are right in Theorem 7 and Theorem 10, we could have used T=1 but there is an important point that we mention below. We were trying to keep the setting similar to Theorem 1 and Theorem 2 as they are the extensions of Theorem 1 and Theorem 2 respectively. Similar to Theorem 1 and Theorem 2, we can show that there is an extension of Theorem 7 and Theorem 10 to settings where we do not observe data at T=1. Suppose we observe data at some length T>1. In that case, we would require that the support for that length T to be the product $\mathcal{X} \times \cdots  \text{T times} \; \mathcal{X}$ so that we can set $x_1=x_2...x_T=x$. We agree with your observations for the other Theorems.
>
> We hope that our explanations are satisfactory and help you view our paper more positively.

---

### Official Review · Reviewer_4sDj · 2024-11-04

**Soundness:** 3
**Presentation:** 3
**Contribution:** 2
**Rating:** 5
**Confidence:** 3

**Summary:**

This paper addresses two key challenges in out-of-distribution generalization for seq-to-seq models: length generalization and compositional generalization. While previous works have primarily focused on empirical approaches to these challenges, this paper aims to provide a theoretical, provable guarantee for achieving both types of generalization. For various architectures (deep sets, Transformers, SSMs, RNNs), the authors provide conditions on models for achieving length and compositional generalization, under the expected risk minimization setup and the realizable assumption. The paper validates their theoretical findings by presenting the experimental results on toy examples.

**Strengths:**

- The paper addresses important challenges in seq-to-seq generalization, length generalization and compositional generalization, across multiple architectures.
- The paper is well-structured and easy to follow.
- This paper is the first work in the literature that presents provable guarantees for length and compositional generalization for seq-to-seq models. Also, the theoretical framework introduced in the paper is a pioneering contribution to the ML community.
- I briefly read the proof, and the proof seems correct and solid.

**Weaknesses:**

1. The models investigated in the paper are highly simplified. For example, the authors describe the decoder-only Transformer model as $w(\sum_{j=1}^i \frac{1}{i} \psi(x_i, x_j))$, where $w$ is a single-layer perception with a bijective activation. This model can not capture softmax operation (as explained in the paper), and apply only to single-layer Transformers. I understand that models with arbitrary capacity (no constraints on $w, \psi$ ) will not exhibit any generalization (as explained in the paper), but the current restrictions are too strong.
- Continuing from the first point, although the theoretical framework is novel, the derived results are not particularly surprising. Given that $w$ is a bijective single-layer perceptron, one can immediately derive $A\sum_{j\le T} \psi(x_j) = B \sum_{j \le T} \phi(x_j)$ (where the right-hand side corresponds to the realizable solution). While the paper presents rigorous proofs involving mathematical analysis techniques to establish the length and compositional generalization from this equation, the results feel somewhat expected, and I think this predictability diminishes the impact and novelty of the findings.
2. The theorems are built under the expected risk minimization setup, which deviates from the practical scenario where we can only access a finite number of samples.
3. While the paper validates their findings by providing experimental results, they are conducted on simplified models. Including experiments on more realistic scenarios would significantly strengthen the paper’s applicability.

**Questions:**

1. Would it be possible to incorporate the optimization process into the theoretical framework?
2. The key point of the proof relies on showing that the Jacobians of the realizable solution f and an arbitrary expected risk minimization solution g are identical. On a related note, it seems that the approach in [1] also involves the use of Jacobians. Could you briefly explain the main differences between your proof and that of [1]? I ask this as I am not familiar with theoretical works on compositional generalization.

[1] Wiedemer, Thaddäus, et al. "Compositional generalization from first principles." *Advances in Neural Information Processing Systems* 36 (2024).

---

> ### Author Response · Authors · 2024-11-16
> **Response to Reviewer 4sDj**
>
> We thank the reviewer for their efforts in giving an insightful review. Please see our responses below. We look forward to hearing back from you.
>
> 1. **Regarding simplified models**
>
>    a) The simplicity of Theorems 1 and 2 is a feature and not a bug.  We present these primarily to build intuition and progressively go towards more complex results in subsequent theorems (Theorem 3-6 and Theorem 8). In Theorem 3 for instance, we move beyond single layer perceptron to C1-diffeomorphisms. In Theorem 4, we consider the model for SSMs used in Orvieto et al. In Theorem 5, we use standard RNN except we only impose restriction on the dimension of the matrices parametrizing the model.
>
>
>
>    b) We also introduce a new result (Theorem 6) on finite hypothesis classes and make no other constraints on the model. For a finite hypothesis class, we show that if the training length is sufficiently large, then the solutions to expected risk minimization achieve length generalization. To illustrate finite hypothesis classes, consider transformer network with many layers with softmax attention. Instead of operating with the property that each scalar value of the weight can take all possible real values, we constrain the weights to only take discrete set of values.
>
>
>
>
> 2. **Regarding expected risk minimization** We want to emphasize that the challenge of out-of-distribution generalization persists even when the training data consists of infinite samples from the training distribution. Due to this most works that study out-of-distribution generalization (such as Wiedemer et al., Lachapelle et al., Arjovsky et al.), assume that there is infinite data from training distribution. In the experiments, we operate with finite data.
>
>
> 3. **On incorporating optimization process:** There are many important and interesting challenges to overcome before a general framework can be built to incorporate optimization process. Due to this we mentioned this problem in the discussions and limitations section as a conjecture. Even in many non-sequence to sequence models studying out-of-distribution generalization, it is common to not include the optimization process (Wiedemer et al., Lachapelle et al., Arjovsky et al, Dong et al.).
>
> 4. **Comparisons with Wiedemer et al.:** Our proof of Theorem 1 has similarity in the Jacobian step but rest of the proof is quite different from Wiedemer et al.. Other theorems --  Theorem 3 for transformers, Theorem 4 for SSMs, Theorem 5 RNNs and Theorem 6 for finite hypothesis classes -- each use different proof technique . We elaborate more below but before that we want to say that  Wiedemer is a very nice work. We view our contributions as complementary to them. They make a strong assumption, which is not practical for our case and hence we have different proofs. They assume that part of the mixing function that generates the data is completely known.
>
>
>      i) The proofs in [Wiedemer et al.] use the fact that  function $C$ is known. The remaining unknowns are $\phi$, which are solutions to a set of PDEs. They show that with the help of the knowledge of value of true $\phi$  at one point, one can reduce the solution space of PDEs to that of ODEs. These ODEs under standard regularity assumptions have a unique solution, which leads to exact identification of true $\phi$. In contrast, we do not need to exactly identify and linear identification suffices.
>
>     ii) The proof for transformers carefully uses properties of diffeomorphisms and spanning conditions to show that attention heads are linearly identified. We first arrive at the condition $a(\frac{\alpha + \beta}{2}) = \frac{1}{2}(a(\alpha) + a(\beta))$, where $a$  dictates the relationship between true and learned attenteion head, i.e., $\phi = a \circ \psi$. Note that this condition immediately does not imply that  $a$ is linear. Standard definition of a linear transform requires that $a(\alpha+\beta) = a(\alpha) + a(\beta)$ and that $a$ is also homogeneous. We use the fact that $a$ is continuous to prove homogeneity.
>
>     iii) For RNNs, we show that the set of all the solutions to ERM in (1) are all permutations of one another in the weight space. We first show a permutation relationship between the matrices that act on the input directly $B$ and $\tilde{B}$. We use this at the subsequent length to show a permutation like relationship between the matrices that operate on the hidden state. To arrive at these results, we prove some important properties of higher order derivatives of sigmoid activation (how it can be viewed as a polynomial) and use properties of zeros of an analytic function. Once this is established we use principle of induction to show that these permutation relationships suffice to match the true labels at all the lengths.

---

> > ### Author Response · Authors · 2024-11-16
> > **Response continued**
> >
> > iv) For SSMs, the proof is again broken down into steps. We first show that learned non-linearity $\omega$ and true one $\rho$ are related to one another by a linear transform, i.e., output of one is a linear transform of the others' output. We then use this linear relationship and use the assumption that the set of points in the support are diverse, i.e., satisfy a certain rank condition, which if met ensures that matrices are all related to each other via a linear transform
> >
> >    v) For finite hypothesis class, we use i) property of monotone convergence of sets, ii) property that a sequence of sets where sets take finitely many possible values has a finite convergence time, iii) the converged set cannot contain a non-length generalizing solution otherwise the set violates monotononicity of the sequence.
> >
> >
> >    **References**
> >
> >    [Orvieto et al.] "On the universality of linear recurrences followed by nonlinear projections." arXiv preprint arXiv:2307.11888 (2023).
> >
> >    [Wiedemer et al.] "Compositional generalization from first principles." Advances in Neural Information Processing Systems 36 (2024)
> >
> >    [Lachapelle et al.] "Additive decoders for latent variables identification and cartesian-product extrapolation." Advances in Neural Information Processing Systems 36 (2024).
> >
> >    [Arjovsky et al.] "Invariant risk minimization." arXiv preprint arXiv:1907.02893 (2019).
> >
> >    [Dong et al.] "First steps toward understanding the extrapolation of nonlinear models to unseen domains." arXiv preprint arXiv:2211.11719 (2022).

---

> > > ### Author Response · Authors · 2024-11-21
> > >
> > > Dear Reviewer
> > >
> > > We believe that our responses and new additions to the paper should sufficiently address your concerns. Please let us know if you have further questions. We hope that you consider increasing the score.
> > >
> > > Thank you!

---

> > > > ### Comment · Reviewer_4sDj · 2024-11-25
> > > >
> > > > Sorry for the late reply.
> > > >
> > > > I thank the authors for their detailed responses. However, I am still not sure about the significant of this paper. The main reason that makes me hesitate to leave a positive opinion on this paper is its limited applicability to real-world scenarios. The proofs and equations presented in the paper rely on strong assumptions, and as a result, I believe these results offer less intuition and insight into real-world applications.

---

> > > > > ### Author Response · Authors · 2024-11-25
> > > > >
> > > > > We thank the reviewer for their response. Please take our response below into consideration.
> > > > >
> > > > > 1. **Regarding real-world application of our theory:** Our theory is inspired by the RASP-L conjecture proposed in Zhou et al., which identifies conditions under which transformers empirically generalize on real algorithmic tasks, such as copy, mode, addition, and sort. The RASP-L conjecture identifies three conditions - realizability, simplicity, and data diversity - but did not provide a formal proof. Our theory captures these three conditions and proves results for a range of architectures and a range of tasks (e.g., multiplication, addition, tasks expressed by finite automata through our result on finite hypothesis classes).
> > > > >
> > > > > 2. **Theoretical research and real world insights:** We believe that there are many theoretical papers that make simplifying and stylized assumptions and their insights may not immediately change practice. Take for example the whole line of work on PAC learning theory, or analysis of convergence in neural networks (often done with very shallow depth two (Du et al.)), or the recent analysis of in-context learning (Oswald et al.) or the analysis of out-of-distribution generalization (Dong et al., Wiedemer et al., Lachapelle et al.), which is quite closely related to ours. We believe all these works are very valuable but may not give any immediate insights that change practice. However, in the big picture they contribute valuable knowledgable which slowly turns into mental models and useful practical insights.
> > > > >
> > > > >
> > > > > Finally, we would appreciate if you can be more concrete on what type of real insights you expect to see that we are missing.
> > > > >
> > > > > **References**
> > > > >
> > > > > [Wiedemer et al.] "Compositional generalization from first principles." Advances in Neural Information Processing Systems 36 (2024)
> > > > >
> > > > > [Lachapelle et al.] "Additive decoders for latent variables identification and cartesian-product extrapolation." Advances in Neural Information Processing Systems 36 (2024).
> > > > >
> > > > >
> > > > >
> > > > > [Dong et al.] "First steps toward understanding the extrapolation of nonlinear models to unseen domains." arXiv preprint arXiv:2211.11719 (2022).
> > > > >
> > > > > [Du et al.] "Gradient descent provably optimizes over-parameterized neural networks." arXiv preprint arXiv:1810.02054 (2018).
> > > > >
> > > > > [Oswald et al.] "Transformers learn in-context by gradient descent." International Conference on Machine Learning. PMLR, 2023.
> > > > >
> > > > > [Zhou et al.] "What algorithms can transformers learn? a study in length generalization." arXiv preprint arXiv:2310.16028 (2023).

---

### Author Response · Authors · 2024-11-16
**Overall Response**

We thank all the reviewers for their efforts and insightful feedbacks. We thank the reviewers for encouraging words such as "serves as a strong foundation for future work in this area", "first work in the literature", "pioneering contribution to the community". We have revised the manuscript to address the concerns that were raised (with major changes marked in teal color). Below we start with the description of a new result followed by a brief summary of responses to key concerns from each reviewer.

1. **New result (Theorem 6) on finite hypothesis classes**  We assume that the hypothesis class is finite and make no other constraints on the model. For a finite hypothesis class, we show that if the training length is sufficiently large, then the solutions to expected risk minimization achieve length generalization.  This captures as a special case settings where we have a transformer network with many layers with standard softmax attention, where each weight vector is constrained to only take a finite of values.


1. **Reviewer 4sDj**:  Theorem 1 and 2 are indeed on the simpler side and are presented to build intuition. Theorem 3-6 and 8, which include the new result described above, apply to more complex model classes than the ones used in Theorem 1 and 2. We believe there is a fair bit of surprise in Theorem 3-6 and 8, and each proof involves several interesting elements.

2. **Reviewer JxtZ:** In the Appendix, following the proofs to the Theorem 1 and 2, we have remarks that state that the theorems continue to hold even if we do not observe data at t=1. We have fixed all the presentation issues you raised. We have also corrected the facts you pointed to in the proofs of theorems concerning discrete tokens.


3. **Reviewer n34f:** We strongly believe there is value in doing theoretical research even if the results do not immediately translate into practical insights. Seminal works on PAC learning theory or convergence analysis (mostly about shallow neural networks) are valuable theoretical contributions but have not found practical insights. Our work is the first to develop theoretical guarantees for the hard problem of length generalization.  We also present a new result (Theorem 6), which applies to arbitrary models (e.g., transformers with many layers) with controlled capacity.


4. **Reviewer MsSC:** We have included a new result on finite hypothesis classes that addresses your brilliant question. Finite hypothesis classes contain as a special case deeper transformers with discretized weights and hard-coded positional encodings. We corrected other presentation issues (e.g., moving assumption to the main body) and changed the definitions as you stated.

---

### Meta-Review · Area_Chair_5GXs · 2024-12-23

**Metareview:**

This submission aims to provide theoretical guarantees on length and compositional generalization for various simplified sequence-to-sequence architectures (deep sets, single-layer transformer variants, simplified RNNs, and SSMs) under the realizability assumption and certain restrictive conditions on capacity and data support. The authors’ motivation is commendable: length and compositional generalization are important challenges, and providing the first theoretical framework to study them is a noteworthy effort. The paper takes initial steps toward understanding these complex phenomena.

**Additional Comments On Reviewer Discussion:**

However, after careful consideration of the reviewers’ comments and the subsequent author responses, the consensus leans toward rejection. Several critical points raised by the reviewers remain insufficiently resolved:
	1.	Practical Relevance and Overly Simplified Assumptions:
The models and assumptions analyzed are highly stylized and differ substantially from architectures and conditions encountered in practice. While simplified settings can provide theoretical insight, the conditions here—such as requiring infinite data from the training distribution, perfect realizability, and highly restricted model capacities or finite hypothesis classes—are seen as too strong and not evidently leading to concrete insights that generalize to real scenarios. One reviewer noted that these assumptions do not illuminate why length or compositional generalization is often so elusive in real-world tasks.
	2.	Connection to Empirical Phenomena:
Although the authors reference the RASP-L conjecture and try to position the theory as a response to empirical findings, the general feeling among the reviewers is that the results remain detached from practical insights. The negative results, which show that slight relaxations of the capacity constraints can break generalization, might better reflect the challenges in real settings. Reviewers suggest that the positive theoretical results, while mathematically correct, might be capturing a phenomenon too fragile and artificial to offer meaningful guidance.
	3.	Presentation and Clarifications:
While the authors made attempts to address typos, redundancy, and referencing errors, some organizational and presentation issues persist. Important assumptions were initially placed in the appendix, and the narrative remains somewhat optimistic about the final applicability of the results. Reviewers found the framing and positioning in the introduction and abstract not sufficiently highlighting the extreme restrictive nature of the assumptions.
	4.	Finite Sample and Statistical Considerations:
The theory does not address finite sample complexity or approximate realizations. The assumption that we have a true risk minimizer with zero training error under infinite data scenarios is substantially stronger than standard settings in learning theory. Given that compositional and length generalization are challenging precisely because of limited data, the lack of statistical considerations limits the relevance of these results.

In summary, while the paper takes a pioneering theoretical step and contributes a novel formalism, the reviewers are not convinced that the current results provide substantial insight into practical length and compositional generalization. The negative results seem more realistic and underscore how delicate the phenomena are, but they do not substantially move the needle on our understanding of how to achieve such generalization in practice. The meta-reviewer thus recommends rejection at this time, encouraging the authors to further refine their theory, relax their assumptions, and improve the connection to practical models and finite sample considerations for future work.

---

### Decision · Program_Chairs · 2025-01-22

Reject